



# Brief communication: Everest South Col Glacier did not thin during the last three decades

Fanny Brun[1], Owen King[2], Marion Réveillet[1], Charles Amory[1], Anton Planchot[1,3], Etienne Berthier[4], Amaury Dehecq[1], Tobias Bolch[2], Kévin Fourteau[5], Julien Brondex[5], Marie Dumont[5], Christoph Mayer[6], and Patrick Wagnon[1]

[1]Univ. Grenoble Alpes, CNRS, IRD, Grenoble INP, IGE, Grenoble, France
[2]Department of Geography and Sustainable Development, University of St Andrews, St Andrews, Scotland, UK
[3]Geosciences Department, École Normale Supérieure - PSL University, Paris, France
[4]Université de Toulouse, CNES, CNRS, IRD, UPS, Toulouse, France
[5]Univ. Grenoble Alpes, Université de Toulouse, Météo-France, CNRS, CNRM, Centre d'Études de la Neige, Grenoble, France
[6]Bavarian Academy of Sciences and Humanities, Geodesy and Glaciology, Munich, Germany

**Correspondence:** Fanny Brun (fanny.brun@univ-grenoble-alpes.fr)

**Abstract.** The South Col Glacier is an iconic small body of ice and snow (approx. 0.2 km$^2$), located on the southern ridge of Mt. Everest. A recent study proposed that South Col Glacier is rapidly losing mass. This seems in contradiction with our comparison of two digital elevation models derived from aerial photographs taken in 1984 and a stereo Pléiades satellite acquisition from 2017, from which we measure a mean elevation change of $0.01 \pm 0.07$ m a$^{-1}$. To reconcile these results we investigate wind erosion and surface energy and mass balance, and find that melt is unlikely a dominant process, contrary to previous findings.

## 1 Introduction

Glaciers and ice caps are losing mass at an accelerated rate in almost all regions on Earth and are icons of climate change (IPCC, 2021; Zemp et al., 2019; Hugonnet et al., 2021). It is generally observed that the lowest parts of glaciers thin at large rates, often exceeding 1 m a$^{-1}$, with variability between regions and glaciers (Hugonnet et al., 2021). Upper accumulation areas are more stable, with rates of elevation changes often close to zero, even for some regions where the warming of the firn column is documented (Vincent et al., 2020).

In the Everest region, glaciers have lost mass at a continuously-accelerating rate since 1962, reaching a regionally-averaged mass loss rate of -0.38 $\pm$ 0.13 m w.e a$^{-1}$ for the period 2009-2018 (King et al., 2020). Thinning for the period 1984-2018 was observed up to elevations of approximately 6000 m a.s.l., and was close to zero at higher elevations (King et al., 2020). Within this context a recent study surprisingly suggested substantial thinning of the South Col Glacier, located at 8020 m a.s.l. (Potocki et al., 2022). Based on the interpretation of an ice core and surface energy balance modeling, Potocki et al. (2022) estimated that contemporary thinning rates — or ablation rates, because the authors did not consider ice dynamics — were





approaching 2 m a$^{-1}$ over the 1990-2019 period.

The South Col Glacier is a small body of snow and ice, covering an area of approximately 0.2 km$^2$. It is located on the ascent route of Mt. Everest from Nepal. Despite this iconic location, it is largely without prior scientific investigation, due to the logistical challenge to conduct scientific fieldwork in this very high altitude environment. It is also a difficult target
for remote sensing techniques, being small and subject both to bright snow conditions and persistent cloud coverage during monsoon (occurring in JJAS). Processes governing the surface energy balance of the ice and snow in the extreme high altitude weather remain poorly constrained, despite the installation of an Automatic Weather Station (AWS) in May 2019 on a rock outcrop close to the South Col (Matthews et al., 2020).

Interpreting a shallow ice core (approximately 10 m long) drilled in May 2019 and modeling the surface energy balance, Potocki et al. (2022) concluded that the glacier surface state likely transitioned from a snow-dominated surface to an ice dominated surface in the 1990s, leading to large amounts of melt (averaging to 1.5 m a$^{-1}$ over the considered period), which was the main cause for the disappearance of ~55 m of ice over a 30 year period covering 1990 to present. Based on these observations, Potocki et al. (2022) suggest that Himalayan glaciers at or above 8000 m a.s.l. may not survive beyond the middle of the 21st
century.

In this brief communication, we explore the elevation change of South Col Glacier for the period 1984-2017, based on digital elevation model (DEM) differencing. We also explore the processes governing the mass balance of South Col Glacier from satellite images and models in order to compare our findings with those of Potocki et al. (2022).

## 2   South Col Glacier elevation change

We used aerial photographs and tri-stereo Pleiades data to generate DEMs and investigate surface elevation changes. The Pléiades DEM was generated from imagery acquired 23 March 2017 using the Ames Stereo Pipeline software (Shean et al., 2016). The 1984 DEM was generated from a subset of images acquired by Bradford Washburn over the wider Everest region on 20 December 1984 using a Wild RC-10 camera (Washburn, 1989) on behalf of the National Geographic Survey. We focus
on three images in which the South Col Glacier is located centrally within the frame to avoid peripheral image distortion, and we processed images from a single flight line in the survey to avoid problems associated with DEM merging. The images were produced by scanning of the original diapositives at 1693 dpi which results in a ground resolution of 0.5 m. The 1984 DEM was generated using PCI Geomatica with the aid of 10 ground control points extracted from the 2017 Pléiades orthoimage and DEM. Both the Pléiades and aerial photograph DEMs were generated at 2 m ground resolution.
We modified the GAMDAM glacier inventory (GGI18) (Sakai, 2019) over the South Col surroundings using the Pléiades high resolution (0.5 m) orthoimage. The revised inventory allowed for coregistration of DEMs over stable, off-glacier areas, following the methods of Nuth and Kääb (2011). We also define the outlines of South Col Glacier, which is not identified



as a single glacier in the original GGI18 inventory (Fig. 1). Following DEM differencing, surface elevation change data (dH) were filtered to remove outliers, with values outside the range of five times the standard deviation of dH estimates within 50 m elevation bands removed below 6800 m a.s.l. Above 6800 m, or from the base of the much steeper Lhotse face, we applied a threshold of three times the standard deviation of dH estimates, as the range of elevation change values were higher here. Any resulting minor data voids were filled using mean values from surrounding pixels within the dH grid.

Uncertainty associated with the average elevation change over glacier surfaces ($\sigma_{\Delta z,g}$) was established following the methods of Rolstad et al. (2009) and Fischer et al. (2015), whereby:

$$
\sigma_{\Delta z,g} = \begin{cases} \sqrt{\sigma_{\Delta h}^2 \frac{A_{cor}}{5 A_g}} & ; A_g \geq A_{cor} \\ A_g \sigma_{\Delta h} & ; A_g < A_{cor} \end{cases}
\tag{1}
$$

where $\sigma_{\Delta h}$ (m) is the standard deviation of dH over stable, off-glacier terrain, $A_{cor} = \pi L^2$, with $L$ being the distance over which dH data are spatially correlated (Rolstad et al., 2009), here measured to be 1193 m, and $A_g$ is the area of the glacier.

Within a 50 m diameter circle centered on the drilling location, the elevation change is $1.7 \pm 2.2$ m for the period 1984-2017, corresponding to an elevation change rate of $0.05 \pm 0.07$ m a$^{-1}$, meaning that the thickness change is statistically not different from zero (Fig. 1). Over the whole glacier area, we find a mean elevation change of $0.01 \pm 0.07$ m a$^{-1}$ for the period 1984-2017. The distribution of dH on South Col Glacier is rather homogeneous and not different from the distribution of dH over ice-free areas or over glacierized areas located within the same elevation range (Fig. A1).

## 3   Surface mass balance processes

We investigate the mass balance processes to further understand the different conclusions between the findings of Potocki et al. (2022) and the non-elevation change result found here. Little is known about the surface mass balance processes at such high altitude, and consequently, we mobilize multiple methods to investigate them. We first analyze high temporal resolution satellite images to document the seasonal evolution of the glacier surface state. We then model the potential magnitude of wind erosion, as it could be a very important process (Litt et al., 2019). Finally, we run different surface energy balance models to discuss the importance of melt.

### 3.1   Seasonal surface state changes from satellite images

Due to its small size, bright environment and persistent cloud coverage during monsoon season, the South Col Glacier is challenging to observe with standard optical satellite imagery. VENµS satellite acquired multispectral images every 2 to 30 days from November 2017 to October 2020 that are suitable to qualitatively document the surface state changes of South Col Glacier (e.g., Bessin et al., 2022). The VENµS mission is jointly developed and managed by the French Space Agency (CNES) and the Israeli Space Agency (ISA). VENµS images consist of 12 narrow spectral bands from blue visible (0.424 µm) to near-infrared (0.910 µm), with a 5 m ground resolution, acquired at 11:00 local time (Dick et al., 2022).

We use 267 VENµS surface reflectance (SRE) multispectral images that we crop to the South Col Glacier surroundings. We horizontally shifted 14 images with poor co-registration, by manually selecting four ground control points. All other images



are well co-registered and orthorectified in the original SRE product. We produce natural color composites using the band

combination 7-4-3, corresponding to red, green and blue bands, respectively. We use the *equalize_adapthist* function from Python *scikit-image* package to obtain a rendering that highlights the blue ice areas (Fig. 2). We rely on visual inspection of the images only, and did not attempt to automatically classify the snow covered areas.

The South Col Glacier exhibits strong seasonal contrasts in terms of snow cover. Here we show only one image per month for the year 2019 (Fig. 2), but the whole image series is available (Brun, 2022). From January to June, the glacier surface is

partially covered by snow and ice is exposed in multiple places. The exposed ice area increases, at least until May, and even sometimes June. After mid June, the number of usable images is limited due to monsoon cloud coverage, and the only image available in July shows that the glacier is covered by a thin layer of snow, as the ice is visible below. The glacier is then covered by an apparently thicker layer of snow in August, September and October. The ice is not visible, except at the serac falls above Western Cwm (upper part of Khumbu Glacier, on the west side of South Col Glacier) in October. The ice re-appears in

November, and its area increases through the course of December (Fig. 2).

The series of VENμS images shows an excellent qualitative agreement with the albedo series measured by the South Col AWS in 2019 (Matthews et al., 2020). The South Col AWS (purple star on Fig. 2) was installed on a rock outcrop, which became covered by snow in early July 2019 (Matthews et al., 2020). Then the albedo of the surface below the AWS remained high until mid October 2019, when the rock outcrop was re-exposed (Matthews et al., 2020).

With VENμS, we observe only three monsoon years (2018, 2019 and 2020), but our findings are similar for the three years. The qualitative interpretation of the VENμS image series hints at a dominant role of wind erosion in the surface mass balance. Large blue ice areas become exposed within a few days in October-November when the available energy is limited for melt (Potocki et al., 2022).

### 3.2 The quantitative importance of wind erosion

In order to test the quantitative importance of wind erosion, we implement a simple model inspired from Amory et al. (2021). For the sake of clarity, the difference between precipitation and erosion is named deposition (and not accumulation).

The parameterization of wind erosion is based on a set of semi-empirical formulations, originally implemented in the polar-oriented regional climate model MAR (Gallée et al., 2001; Amory et al., 2021). MAR has been used to study the cryosphere under various climates, and the erosion/deposition processes have been largely validated in Antarctica where these processes

are critical (e.g., Agosta et al., 2019). The cold and windy environment of Antarctic glaciers is also the most similar to high elevation sites such as South Col Glacier. Here we develop a simplified analytical approach expressed in a 1D vertical frame-work, assuming that snow re-mobilized from the surface by wind is entirely exported by horizontal transport off the glacier boundaries, which is justified by the local topography, with steep slopes around the glacier, especially toward the east (westerly predominant winds). Due to its one-dimensional, offline nature, our analytical erosion model does not account for horizontal

advection of airborne snow from upstream areas and interactions of airborne snow particles with the atmosphere are neglected. Following Amory et al. (2021), fresh snow is assumed to be deposited with a density $\rho$ of 300 kg m$^{-3}$, the erosion rate is parameterized as a function of surface snow density only, and erosion is not allowed to occur when $\rho \geq 450$ kg m$^{-3}$ or when





the air temperature exceeds the freezing point. In this approach the snowpack is described with a 3-layer model, representing a range of $\rho$ values from fresh snow (top layer; $\rho = 300$ kg m$^{-3}$), erodible aging snow (intermediate layer; 300 kg m$^{-3}$ <

$\rho < 450$ kg m$^{-3}$) and non-erodible firn (bottom layer; $\rho \geq 450$ kg m$^{-3}$). This discretization limits the eroded snow mass to the available erodible mass budget and allows accumulation in the firn. Snow densification of the top layer is parameterized following a linear densification rate as implemented in MAR, and is considered to occur under the action of erosion only. Note that our model does not resolve the surface energy balance and other mass balance processes (sublimation and melt), and inherently ignores the related snow metamorphism. The snowpack is initialized as a single non-erodible firn layer. The model

is described in more details in Appendix A. The meteorological inputs (precipitation, wind speed, air temperature and surface pressure) required to drive the parametrization of wind erosion are directly taken from Potocki et al. (2022), and cover the period 1950-2019 as they originate from ERA5 downscaled data (Hersbach et al., 2020). We use their uncorrected precipitation rates (averaging at 191 mm a$^{-1}$), as the tuned estimates (averaging at 66.9 mm a$^{-1}$) aimed at correcting for missing wind erosion in Potocki et al. (2022).

The dominant meteorological conditions at South Col Glacier were previously described (Matthews et al., 2020; Potocki et al., 2022). We focus on wind and precipitation, which both have a strong seasonal cycle (Fig. 3a). Most of the precipitation falls during the monsoon (JJAS). In general, the winds are extremely strong, with mean daily values approaching 20 m s$^{-1}$ in winter when the westerlies hit the topography (Maussion et al., 2014). During monsoon, the winds progressively decrease, from a daily mean of around 10 m s$^{-1}$ in early June, to 2.5 m s$^{-1}$ in the core of the monsoon season (July and August). After

mid September, the wind speed increases very sharply, marking the end of monsoon (Khadka et al., 2021). As a consequence, the precipitation falling before June is not deposited, and the deposition efficiency (ratio of daily deposition over precipitation, defined only when precipitation is non-zero) is close to zero (Fig. 3b). During the course of the monsoon, the deposition efficiency gradually increases, and a large proportion of falling snow is not eroded. The cumulative deposition reaches a maximum in mid September and ultimately decreases in October, when winds strengthen again, meaning that erosion becomes

larger than precipitation, thus leading to a negative deposition efficiency (Fig. 3b). At that time of year, in our model, the snow reaches the density of 450 kg m$^{-3}$, which does not allow any additional erosion.

Over the period 1950-2020, the annual precipitation ranges from 147 to 259 mm w.e. (mean value 191 mm w.e.), and only 0 to 51 % (mean value 25 %) of the precipitation is deposited, ranging from 0 to 97 mm w.e. (mean value 48 mm w.e.), the rest being eroded by wind. The interannual variability of the deposition efficiency is high, and the model results are very sensitive

to the wind (Figs. 3c and d). With a wind speed increased by 30 to 40 %, the annual deposition is almost reduced to zero (Fig. 3d). Conversely, if the wind speed is 40 % lower, the mean annual deposition is about 120 mm w.e., which is 63 % of the mean annual precipitation. The results are also very sensitive to the total amount of precipitation, with an increase of 40 % in precipitation corresponding to an increase in the deposition of 80 % (Fig. 3d).

The model of wind erosion is a simple model that has large limitations. It takes into account only wind erosion and time-

dependent densification of snow, which acts as a negative feedback on erosion. Despite these limitations, the results are in good qualitative agreement with the VENµS images. For instance, the pre-monsoon and early monsoon (MAM) snowfalls disappear systematically both in the model and images. Then, the images show that the glacier is covered by snow during the core of the



monsoon (August), which is when the model finds the highest deposition efficiency (Fig. 3b). This pattern is conserved when testing the model sensitivity to the wind and precipitation input (Fig. A5). However, the images show erosion, or sublimation,

during winter (Fig. A2). This is not predicted in the model, because the snow reaches a density of 450 kg m$^{-3}$ that is the maximum imposed in the model, preventing further erosion. It is noteworthy that sublimation accounts for 100-400 mm w.e. a$^{-1}$ (Potocki et al., 2022), which is larger than the mean annual deposition simulated within the range of precipitation and wind values tested here.

From the analyses of the satellite images and the modeling of wind erosion, we conclude that large parts of the fallen snow

are likely eroded or re-mobilized after deposition. Our findings have two main implications. First, we suggest that the South Col Glacier is entirely covered by snow during large parts of the monsoon season (at least during the whole months of August and September), strongly limiting the potential for ice melt. Second, the fact that the deposition efficiency is not constant in time, together with the high magnitude of snow erosion, imply that wind erosion is a major ablation process, that is not constant in time. As a consequence, using "net precipitation" (defined as precipitation after removal of snow by wind erosion in

Potocki et al. (2022)) artificially balances the surface mass losses of a snow surface for an arbitrary period. Indeed, the snow that falls during monsoon would have a longer residence time compared to snow which fell outside the monsoon, and would thus integrate the surface energy balance over a much longer period. As a consequence, the net precipitation rates estimated in Potocki et al. (2022) are likely affected by very large uncertainties.

From the wind erosion model, we demonstrate that wind erosion is an important surface mass balance component. However, our model is intrinsically limited because it does not consider the other mass balance processes, such as melt and sublimation. Despite the availability of VENμS images, we cannot determine precisely the temporal share of ice exposed or snow covered conditions for South Col Glacier. We therefore investigate the ablation processes by modeling the glacier mass balance over icy surfaces to consider a worst-case scenario that maximizes ablation (relative to a snow covered scenario).

## 3.3 The challenge of modeling the surface mass balance

Modeling the changes of ice or snow masses in response to atmospheric forcing is a complex task which involves resolving heat transport and conservation, with concurrent phase changes (e.g., Tubini et al., 2021). To model the surface mass balance of the South Col Glacier, Potocki et al. (2022) relied on the COSIPY model (Sauter et al., 2020). They notably found that in the case of a yearly snow free surface, the COSIPY model predicts a substantial 1508 mm w.e. a$^{-1}$ of melt on average for the

simulation period 1950-2020. The goal of this section is to assess the robustness of this large modeled annual melt. For this purpose, we performed similar mass balance simulations of an ice surface, with different models.

We compared three different models forced with the same meteorological inputs, taken from Potocki et al. (2022), to simulate the ice ablation for the year 2019 (arbitrarily chosen). First, we use the standard version of COSIPY run with an hourly

time-step, as in Potocki et al. (2022). This simulation is named hereafter COSIPY_P22. We then used a slightly modified version of COSIPY with (i) a one minute time-step and (ii) a modified computation of the heat conduction flux between the





surface and the ice below, that is referred to as the ground heat flux in COSIPY (Sauter et al., 2020) and in Potocki et al. (2022). Indeed, in the default COSIPY settings used in Potocki et al. (2022) the heat conduction flux is computed by considering the temperature gradient on the first 10 cm of the ice column. With this implementation the temperature gradients remained rather

small, and we thus modified the COSIPY source code to compute the temperature gradient as close as possible to the surface. Specifically, we use the temperature values at the surface and at the node below (usually at 1 cm depth), in order to compute the temperature gradient in the vicinity of the surface. This simulation is named hereafter COSIPY_grad. Finally, we run a simulation using the detailed snowpack model Crocus, with a version adapted to an icy surface. This version was notably used in the Alps to model glacier mass balance (Vionnet et al., 2012; Réveillet et al., 2018). The model is described in more

detail in Appendix B. This simulation is named hereafter Crocus_ice. In order to maintain an icy surface in Crocus_ice and COSIPY_grad, we set the precipitation to zero, thus enhancing the potential for melt.

All three simulations are initialized with the thermal state predicted by a COSIPY simulation similar to COSIPY_P22, that ran from 2010-01-01 to 2019-01-01.

Among the three simulations, only the COSIPY_P22 predicts a significant amount of melt (1368 mm w.e. for the year 2019), while COSIPY_grad only predicts 15 mm w.e. a$^{-1}$ and Crocus_ice predicts no melt at all (Fig. 4). Note that a small amount of melt that immediately refreeze could occurs within the same time step (i.e. 15 min) but is not shown by the outputs. This large difference cannot be accounted for by differences in the radiative or turbulent fluxes, as they are similar for all three simulations (Fig. A3). In particular, the daily sublimation rate is very close for the three simulations (Fig. 4) with a cumulative sublimation

of 453, 482 and 413 mm w.e. simulated by COSIPY_P22, COSIPY_grad and Crocus respectively for the year 2019. Furthermore, in all three simulations, the surface reaches the melting point during daytime in the monsoon period (Fig. 4), but only COSIPY_P22 predicts a large amount of melt. The simulated surface temperatures are very close for COSIPY_grad and Crocus (RMSE = 0.9 K, bias = 0.6 K) while they are almost systematically higher for COSIPY_P22 than for Crocus (RMSE = 3.5 K, bias = 2.7 K).


    As the three models predict surface temperature reaching melting temperature and similar surface fluxes, the discrepancies between their predicted melt can likely be explained by differences in the representation of energy absorption, distribution and transport between the three models.

In COSIPY_P22 and COSIPY_grad the surface temperature is computed by searching for the temperature that equilibrates the

energy fluxes at the surface, assuming that a given portion (80 %) of the incoming shortwave radiative flux is absorbed right at the surface and does not penetrate further. If the equilibrium temperature is above the melting point, the surface temperature is capped at the melting temperature and the excess energy is used for melting the surface. The computation of this equilibrium temperature requires to compute the amount of energy transported by conduction from the surface into the ice column, the so-called ground heat flux in Sauter et al. (2020) and Potocki et al. (2022), and actually representing the subsurface heat

conduction flux. As explained above, in the standard COSIPY version this subsurface heat conduction flux is estimated using the first 10 cm of the ice column, and is therefore potentially not representative of the strong temperature gradient that can be





present in the direct vicinity (i.e. the first centimeters) of the surface. This can lead to a poor estimate of the subsurface heat conduction flux, affecting the amount of energy accumulated at the surface and, in this case, favoring melt. This phenomenon is further exacerbated by the use of an hourly time-step and the decoupling of the energy absorption and of the internal heat

diffusion processes in COSIPY: energy is accumulated during a full hour before it can be removed by internal diffusion, leading to temperature and melt overshoots. The reduction of the time-step to a minute in COSIPY_P22 reduces the predicted melt by about 30 % (not shown here). We note that the simulation at one hour time-step and a temperature gradient calculated very close to the surface is unrealistic as it results in a mean annual subsurface flux of 600 W m$^{-2}$.

In comparison, in COSIPY_grad the computation of the subsurface conduction heat flux is performed as close as possible to

the surface. This, in combination with a smaller time step, allows the model to efficiently evacuate energy from the surface, hindering melting. As seen in Fig. A4, the COSIPY_grad simulation predicts much higher subsurface heat conduction fluxes during the monsoon period, strongly limiting the surface melt.

Finally, in Crocus the surface temperature is computed by considering a finite thickness for the first layer as opposed to COSIPY, which is a skin temperature model (e.g., Covi et al., 2022). Consequently Crocus does not require the computation

of the surface temperature that equilibrates the surface energy fluxes, and thus does not need to approximate the subsurface heat conduction flux. Furthermore, in Crocus the absorption and conduction of energy are performed simultaneously, making it less likely to locally accumulate enough heat for melting.

This numerical experiment demonstrates that the structure and configuration of a model can strongly affect the way the

energy is spatially allocated and transported, leading to large variations in predicted melt despite solving, in principle, the same physical processes. Notably, the COSIPY_grad and CROCUS_ice simulations highlight that physically consistent modeling configurations can lead to little or no melt when considering the energy budget of an ice surface in the conditions of South Col Glacier. In view of this large modeling uncertainty, it is therefore not possible to definitely conclude on the amount of surface melting when ice is exposed on South Col Glacier (i.e. when snow-free conditions are observed in June and July) solely from

a modeling point of view. However, the very large annual melt totals in COSIPY_P22 appear, at least, very questionable. This is reinforced by the little evidence for large amounts of melt happening at South Col Glacier. For instance, there are no large supraglacial stream features visible in the aerial photographs or satellite imagery, even though they would be expected to develop when 1.5 m w.e. of ice is melted every year. Similarly, in the photographs of the glacier surroundings, there is no evidence of runoff, such as stones being embedded into in re-frozen water, or crevasses being filled by re-frozen ice, all of which are

usually observed when melt occurs on cold glaciers. Yet, these qualitative observations cannot be used as model evaluation. We thus highlight here that snow and ice energy balance models are developed and tested in specific conditions, and might not be applied directly to other conditions, such as the very specific conditions of South Col glacier, without extensive validation.





## 4 Implications for the interpretations of the South Col Glacier ice core

In the original interpretation of the South Col Glacier ice core, (Potocki et al., 2022) dated the top 10–69 cm of the core and found an age of 1966 ± 179 years. Based on a layer counting estimate of annual accumulation of 27 mm w.e. a$^{-1}$, they concluded that ~55 m of ice was missing and has been removed. Relying on surface mass balance modeling with COSIPY, they estimated contemporary thinning rates approaching 2 m a$^{-1}$, and thus dated the initiation of thinning in the 1990s, leading to a cumulative mass loss of ~55 m by 2019.


The interpretation of Potocki et al. (2022) relies on two assumptions: (i) a continuous and stable accumulation of ~ 27 mm w.e. a$^{-1}$ for a 2000 year period and (ii) the absence of ice flow and ice emergence that would transport old ice from the accumulation zone to the surface of the ablation zone (e.g. Fig. 1, Jouvet et al., 2020).

Regarding (i), given the small value for accumulation, and the magnitude and variability in wind erosion, it seems difficult to accept that accumulation is stable and continuous. Moreover, the accumulation is estimated based on 10 m of ice, which consequently corresponds to an approximate duration of only 330 years. The assumption that similar surface mass balance conditions persisted for up to the past 2000 years is therefore based on a large extrapolation. If the accumulation was not continuous, i.e. if there are some missing years in the core records, then the 1966 ± 179 year old ice exposed near to the surface 270 would not imply 55 m of missing ice. More precise dating of the deeper sections South Col Glacier ice core could help resolve the question of continuity of accumulation.

Regarding (ii), Potocki et al. (2022) neglected ice flow, and thus did not distinguish surface mass balance and elevation/thickness change. However, South Col Glacier is flowing at an unknown velocity as evidenced by the presence of a bergschrund, 275 large crevasses and primary stratification visible on the Pléiades ortho image (Fig. A6). Given that the ice core has a density of 0.89 kg m$^{-3}$, and that no firn is mentioned in Potocki et al. (2022), we suggest interpret the core was drilled in the ablation area of South Col Glacier. Based on the drilling location, the glacier topography and the presence of exposed ice visible on Pléiades ortho image, we delineate a minimum extent of the ablation area of South Col Glacier (Fig. A6). Our observations from DEM difference (Section 2) show that the elevation change of South Col Glacier is close to zero for the period 1984-2017 implying 280 that ablation is compensated by the ice flux ($\Phi$ in m$^3$ a$^{-1}$) coming from the accumulation area, leading to (e.g., Cuffey and Paterson, 2010):

$$\Phi = L\bar{H}\bar{U} = \dot{b}_{abl}A_{abl} \tag{2}$$

where $L$ is the length of the section ($L = 670$ m here, Fig. A6), $\bar{H}$ is the mean depth along the section (we conservatively assume $\bar{H} = 50$ m here), $\bar{U}$ is the depth-averaged velocity along the section (in m a$^{-1}$), $\dot{b}_{abl}$ is the area averaged surface mass 285 balance in the ablation area (assumed to be 2 m a$^{-1}$) and $A_{abl}$ is the area of the ablation area ($A_{abl} = 0.14$ km$^2$ here). This back of the envelope calculation leads to a depth-averaged velocity of 8.5 m a$^{-1}$, which is unrealistically high (for reference, the drilling site is located only 150 m downstream of the bergschrund and the velocity deformation is estimated at 0.2 to 0.7 m





a$^{-1}$ for an ice thickness of 50 m and is an order of magnitude smaller for 25 m of ice; see Appendix C).

Finding ice that is as old as 1966 $\pm$ 179 years at the surface suggests that South Col Glacier is flowing very slowly. Given that the core was drilled less than 150 m downstream of the bergschrund, the maximum velocity averaged along the flow line is approximately 7 cm a$^{-1}$, but could be much smaller as we do not know where the ice formed along this flowline. The South Col Glacier is thus likely a very continental type of glacier with small ice fluxes. This would be in good agreement with our modeling of wind erosion and surface mass balance, suggesting arid conditions. A large fraction of precipitation (> 60 %) is
not deposited, limiting the incoming precipitation deposition. Then sublimation is likely the dominant ablation process, that removes 90 to 450 mm w.e. of snow or ice per year (Potocki et al., 2022), leading to a very small mass turnover (i.e. absolute value of surface mass balance), and therefore small ice fluxes (e.g., Cuffey and Paterson, 2010).

    One alternative process is that avalanches occurring at the foot of Everest south east face feed the accumulation area of the glacier with dense snow that is more difficult to erode, leading to a local excess of mass, and thus actual accumulation.
For the rest of the glacier area, deposition and sublimation, that are both controlled by wind, maintain a thin balance between accumulation and ablation.

## 5   Conclusions

In our study, we demonstrate on the basis of remote sensing information that no significant elevation change occurred at South
Col Glacier between 1984 and 2017. This is in contrast with the postulated strong melt, or thinning, rates of almost 2 m a$^{-1}$ at the location of the ice core during the recent past proposed by Potocki et al. (2022). Our results show that large magnitude of melt is unlikely to happen on South Col Glacier for three main reasons: i- the glacier is covered by snow during large parts of monsoon season, limiting the net incoming shortwave radiation, ii- when the glacier surface consists of exposed ice, the occurrence of melt is highly dependent on modeling choices (i.e. model structure and parameters) for similar meteorological
inputs, and the COSIPY model structure might not be well suited for this particular application, and iii- first order estimates of ice fluxes do not allow for a large amplitude of mass turnover.

    The surface mass balance processes happening in the extreme meteorological context of South Col Glacier are complex, and our study does not reach any definitive conclusion about the relative importance of each of these processes. The lack
of direct observations hampers our ability to decipher the dominant glaciological processes, and thus to model the glacier recent and future evolution in a realistic way. Specifically, stake measurements would be needed to measure the surface mass balance and surface velocity in a direct way, ground penetrating radar measurements would help constrain the ice thickness, and a number of subsurface temperature, snow-depth, snow transport or turbulent fluxes measurements would help constrain the processes. Without more data constrained knowledge, it appears currently impossible to conclude about the sensitivity of
South Col Glacier to climate change, nor to predict its future evolution.





*Code and data availability.* The dH map is available upon request and will be available on a public repository, pending publication.,The Pléiades DEM is available at https://doi.org/10.5281/zenodo.6979691 (Berthier, 2022). License restrictions apply to the Pléiades ortho images, that are available upon request to E.B., pending the signature of license agreement with CNES.

VENµS data are available at: https://doi.org/10.5281/zenodo.6685515 (Brun, 2022)

The wind erosion code is available at: https://github.com/antonplanchot/wind-snow-scg (Planchot and Amory, 2022)

Mass balance simulations presented in this study are available at https://doi.org/10.5281/zenodo.7006744, as well as forcing and initialisation information to reproduce Crocus simulations. SURFEX/ISBA–Crocus model based on version v8.0 has been used in this study, and is an open source code hosted on an open git repository at CNRM.

## Appendix A: Parameterizing wind erosion

The contribution of wind erosion of snow to the local glacier mass balance results from three-dimensional interactions of the atmospheric flow with the surface which thus, in principle, cannot be properly treated with a one-dimensional model. Although the erosion process, which describes the removal of snow from the surface by wind, may be expressed in a one-dimensional vertical framework, the net export of snow by horizontal transport, however, needs resolving in the horizontal dimensions by definition. Due to the relatively small dimensions of the glacier (0.2 km$^2$) and its complex topographical surroundings,

capturing the erosion process over this full range of spatial scales requires an eddy-resolving, three-dimensional model with a very fine, meter-scale horizontal resolution and appropriate lateral boundary fluxes. This approach would also require a very large amount of computational resources and is thus not suited for climatological time scales. As a much more computationally efficient alternative, we used a simplified approach to parameterize wind erosion of snow as a one-dimensional vertical model.

Our parameterization is inspired from the erosion scheme of the regional climate model MAR (Gallée et al., 2001; Amory

et al., 2021) and assumes a complete export of eroded snow outside of the glacier boundaries by horizontal wind transport. This assumption is crude but has some theoretical support. Due to the geometry and the NE-SW orientation of the South Col Glacier (Fig. 1), westerlies are the dominant and strongest local winds (Fig. A7) as a result of local topographical channeling. They may thus mostly be responsible for snow erosion and export off the eastern side of the glacier. An illustration of this process is the striking expansion of blue ice surfaces revealed by analysis of VENµS images during the last quarter of year

2019 (Fig. 2), when strong wind speeds of high erosive potential occur after the monsoon season. Another way to support this assumption is to confirm that a snow model equipped with a parameterization of wind erosion can reproduce this feature.

Wind erosion is usually considered to initiate when the friction velocity, $u_*$ (m s$^{-1}$), exceeds a threshold value, $u_{*t}$ (m s$^{-1}$), mostly determined by snow microstructural characteristics at the surface including snow density, grain size and bond number and strength (Schmidt, 1980). In our model, the friction velocity $u_*$ is derived from the 2-m local wind speed forcing U, directly

taken from Potocki et al. (2022) and assuming a logarithmic wind profile:

$$u_* = \frac{kU}{\ln(2/z_0)} \tag{A1}$$

where k is the von Kármán constant (0.4) and $z_0$ the roughness length for momentum set to $10^{-4}$ m. Following Amory et al. (2021), in the absence of observational characterization of local surface snow characteristics, the threshold friction velocity $u_{*t}$





is expressed as a function of snow surface density only:

$$u_{*t} = u_{*t0} \exp\left( \frac{\rho_{ice}}{\rho_0} - \frac{\rho_{ice}}{\rho_s} \right) \tag{A2}$$


with $u_{*t0}$ the expression for the standard friction velocity (0.211 m s$^{-1}$), $\rho_{ice}$ the density of ice (920 kg m$^{-3}$), $\rho_0$ the density of fresh snow (set to 300 kg m$^{-3}$) and $\rho_s$ the density of surface snow (kg m$^{-3}$).

When $u_* > u_{*t}$, the air temperature is below the freezing point and the surface snow density is below 450 kg m$^{-3}$ (see below), the particle ratio in the saltation layer $q_{salt}$ (kg kg$^{-1}$ ; mass of saltating snow particles per unit mass of atmosphere) is computed as a function of the excess of shear stress responsible for removal of snow particles from the surface:


$$q_{salt} = \frac{u_*^2 - u_{*t}^2}{gh_{salt}} e_{salt} \tag{A3}$$

where $g$ = 9.81 the gravitational acceleration (m s$^{-2}$), $h_{salt} = 0.08436u_*^{1.27}$ is the height of the saltation layer (m) and $e_{salt} = \frac{1}{3.25u_*}$ is the saltation efficiency expressed as a dimensionless coefficient inversely proportional to the friction velocity. A surface turbulent flux of snow, referred to as potential erosion $E_p$ (m s$^{-1}$ kg kg$^{-1}$), is then approximated assuming that it follows a bulk formula:


$$E_p = C_D U(q_{salt} - q_s) \tag{A4}$$

where $C_D$ is a drag coefficient ($10^{-3}$) similar to that used for sensible and latent heat fluxes, $U$ is the near-surface wind at standard level (10 m) and $q_s$ is the snow particle ratio (kg kg$^{-1}$) at the same level. Assuming that any re-mobilized snow is quickly removed by horizontal transport, then $q_{salt}$ - $q_s \approx q_{salt}$, and re-expressing $U$ assuming a logarithmic wind profile, $E_p$ writes:


$$E_p = C_d q_{salt} \frac{u_*}{k} \ln\left( \frac{10}{z_0} \right) \tag{A5}$$

Equation A3 contains semi-empirical formulations assumed to implicitly account for all the physical processes that contribute the airborne snow mass, including the gravitational settling of snow particles. Equation A5 expresses turbulent vertical exchange between the saltation layer and the overlying atmosphere but not settling. However, the saltation layer contains significant quantities of snow under strong winds (Nemoto and Nishimura, 2004; Huang et al., 2016), in which case turbulence is a dominant contributor over settling which is consequently ignored. We then deduce an expression for the maximum amount of erodible snow $ER$ (kg m$^{-2}$) during one time step $dt$ (1 h):


$$ER = E_p \rho_{air} dt \tag{A6}$$

where $\rho_{air}$ = P/(RT) is the density of air (kg m$^{-3}$), P the surface pressure (Pa), R = 287 the specific gas constant (J kg$^{-1}$ K$^{-1}$) and T the air temperature (K). The effective erosion $ER_{eff}$ (kg m$^{-2}$) is then taken as the minimum between the maximal erosion $ER$ and the available snow mass for erosion (mass content in the top and intermediate layer) in the snow model. Note that u*t is recomputed for each snow layer. If a layer is completely eroded during a time step, the layer below is also eroded






during the same time step. The maximal erosion $ER$ of this layer is then computed taking into account the time spend to erode the upper layer.


In natural environments, wind erosion contributes to the densification of the snow surface due to the combined actions of wind and saltation which break original crystal shapes and favour the formation of smaller, rounded snow grains (Sato et al., 2008), leading to enhanced sintering, more efficient mechanical packing and increased density (Vionnet et al., 2013). Erosion-induced densification, together with the exposure of denser snow or ice layers through erosion, both naturally contribute to

reduce the likelihood of additional erosion. In our model, erosion-induced densification is applied to the top snow layer that has experienced erosion following a linear densification rate from the fresh snow value $\rho_0$ (assumed to be representative of snow that have been barely altered by post-depositional processes) to the prohibitive density value for snow erosion $\rho_{max}$:

$$\frac{d\rho_s}{dt} = \frac{\rho_{max} - \rho_0}{\tau_{ER}} \tag{A7}$$

where $\rho_{max} = 450\,\mathrm{kg\,m^{-3}}$ and $\tau_{ER}$ is the characteristic time scale for erosion-induced densification set to 24 h. As our model

time step (our hour) is small compared to $\tau_{ER}$, the negative feedback of snow erosion within one time step can be neglected and is only active from one time step to another.

After the densification step, if the snow density exceeds the density criteria of the layer, it is transferred to the next denser layer. When snow is moved from the top layer to the intermediate layer, the density of the intermediate layer is recomputed by weighted average of the former thickness of each layers.

Our model is a simplified approach aiming at quantifying the effect of wind erosion on the mass balance of the South Col Glacier. Other ablation processes (melt, sublimation) resulting from the surface energy balance, snow metamorphism processes and snowpack internal changes in snow characteristics which can lead to additional densification are neglected. However, we expect few implications since erosion of a snow layer of density above $\rho_{max}$ of $450\,\mathrm{kg\,m^{-3}}$ is prohibited. Even if some internal model parameters ($\rho_0$, $\rho_{max}$, $\tau_{ER}$) have been scaled for Antarctic conditions (Amory et al., 2021), we simply re-used without

additional tuning the original configuration of MAR, considering the similarities in the climatological context between Coastal Antarctica and the South Col Glacier (extreme wind speeds and relatively low air temperatures).

The erosion parameterization proposed above is admittedly very approximate. It ignores notably the sublimation of airbone snow (Gallée et al., 2001). However, as we assume a complete export of drifting snow, whether the suspended snow is finally exported in the solid or vapor phase makes no difference. Although other formulations of wind erosion could be devised, a

more constrained parameterization could not necessarily be developed without additional measurements or through a coupling with a multidimensional model and/or a multilayer snow model. Our main point here is that, in order to explain the gradual removal of snow and resulting appearance of blue ice areas as revealed from analysis of VENμS images and understand the reasons behind its temporal variability, it is necessary to account for wind erosion.





## Appendix B: Crocus Model

Crocus is a full energy balance, one-dimensional snowpack model, driven by meteorological variables. It simulates a layered snowpack with a Lagrangian representation, each layer being characterised by its thickness, density, temperature, liquid water content and two semi-empirical variables to describe the snow/ice microstructure (i.e. SSA and sphericity).The model solves the heat diffusion equation in the snowpack at a 15 min time step considering the different energy fluxes between the surface and the atmosphere and between the bottom of the snowpack and the soil. Physical processes such as solar radiation absorption, liquid

water percolation, snow metamorphism and settlement are also considered by the model. While it was originally developed to simulate seasonal snowpack and to assist in avalanche hazard forecasting over the French mountain ranges (Vionnet et al., 2012), the model can be used on icy surfaces, considering an ice layer as a specific snow layer with a density of $917\,\mathrm{kg\,m^{-3}}$ (Gerbaux et al., 2005; Lejeune et al., 2007; Dumont et al., 2012; Réveillet et al., 2018). However, to our knowledge, the model has never been applied to simulate cold glacier mass balance and a careful evaluation would be necessary to further apply

Crocus in such context.

    Crocus model is forced using the same hourly meteorological forcing described in Potocki et al. (2022). The 'ice simulation' is performed without considering the precipitation (to ensure simulating icy surface mass balance only). Initialisation of the ice thickness and thermal profile is initialized with the state predicted by a COSIPY simulation that ran from 2010-01-01 to 2019-01-01 from Potocki et al. (2022). Model parameterization (i.e. ice albedo, ice roughness values, atmosphere stability

correction) is done following the study of Potocki et al. (2022). More details, data to reproduce the simulations and simulation outputs are found at: https://doi.org/10.5281/zenodo.7006744

## Appendix C: Estimating the surface velocity due to ice deformation

The surface velocity ($u_s$ in m s$^{-1}$) of a glacier can be approximated as (Cuffey and Paterson, 2010):

$$u_s = u_b + \frac{A}{n+1}\tau_b^n H \qquad\qquad (C1)$$

where $u_b$ (in m s$^{-1}$) is the basal velocity, assumed to be zero for cold glaciers, $A$ (in s$^{-1}$ Pa$^{-3}$) is the creep parameter , $n$ is the exponent of Glen's flow law, $H$ (in m) in the ice thickness (Cuffey and Paterson, 2010). $\tau_b$ (in Pa) is the basal shear stress, expressed as:

$$\tau_b = \rho_{ice}gH\alpha \qquad\qquad (C2)$$

    Where $\rho_{ice} = 0.9 \times 10^3$ kg m$^{-3}$ is the ice density, $g = 9.81$ m s$^{-2}$ is the gravitational acceleration, and $\alpha = 0.36$, is the slope

(corresponding to 20 degrees). We take $n = 3$, and the values recommended for $A$ (Table 3.4, pp 75 in Cuffey and Paterson, 2010). We assume 1 year = $3.16 \times 10^7$ s. The results are shown in Table C1.





**Table C1.** Surface velocity (in m a$^{-1}$) for different ice temperatures and ice thicknesses

| T (°C) / Thickness (m) | 5 | 10 | 25 | 50 | 100 |
|---|---|---|---|---|---|
| 0 | $7.6\times10^{-4}$ | $1.2\times10^{-2}$ | $4.8\times10^{-1}$ | $7.6\times10^{0}$ | $1.2\times10^{2}$ |
| -5 | $3.0\times10^{-4}$ | $4.7\times10^{-3}$ | $1.8\times10^{-1}$ | $3.0\times10^{0}$ | $4.7\times10^{1}$ |
| -10 | $1.1\times10^{-4}$ | $1.8\times10^{-3}$ | $6.9\times10^{-2}$ | $1.1\times10^{0}$ | $1.8\times10^{1}$ |
| -15 | $6.7\times10^{-5}$ | $1.1\times10^{-3}$ | $4.2\times10^{-2}$ | $6.7\times10^{-1}$ | $1.1\times10^{1}$ |
| -20 | $3.8\times10^{-5}$ | $6.1\times10^{-4}$ | $2.4\times10^{-2}$ | $3.8\times10^{-1}$ | $6.1\times10^{0}$ |
| -25 | $2.2\times10^{-5}$ | $3.5\times10^{-4}$ | $1.4\times10^{-2}$ | $2.2\times10^{-1}$ | $3.5\times10^{0}$ |

*Author contributions.* F.B., M.R. and C.A. designed the study. O.K. and E.B. processed the DEMs and DEM differences, M.R., K.F., J.B. and M.D. ran the Crocus simulation and analyzed the results, C.A. and A.P. developed the wind erosion model and analyzed the results. F.B. led the writing and all the authors contributed to it.

*Competing interests.* Some authors are members of the editorial board of The Cryosphere. The peer-review process was guided by an independent editor, and the authors have also no other competing interests to declare.

*Acknowledgements.* CNRM/CEN is part of Labex OSUG@2020. Marie Dumont, Julien Brondex and Kevin Fourteau have received funding from the European Research Council (ERC) under the European Union's Horizon 2020 research and innovation program (IVORI (grant no. 949516)). E.B. acknowledges support from the French Space Agency (CNES). We thank Evan Miles and Vincent Favier for insightful
discussions and comments.



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



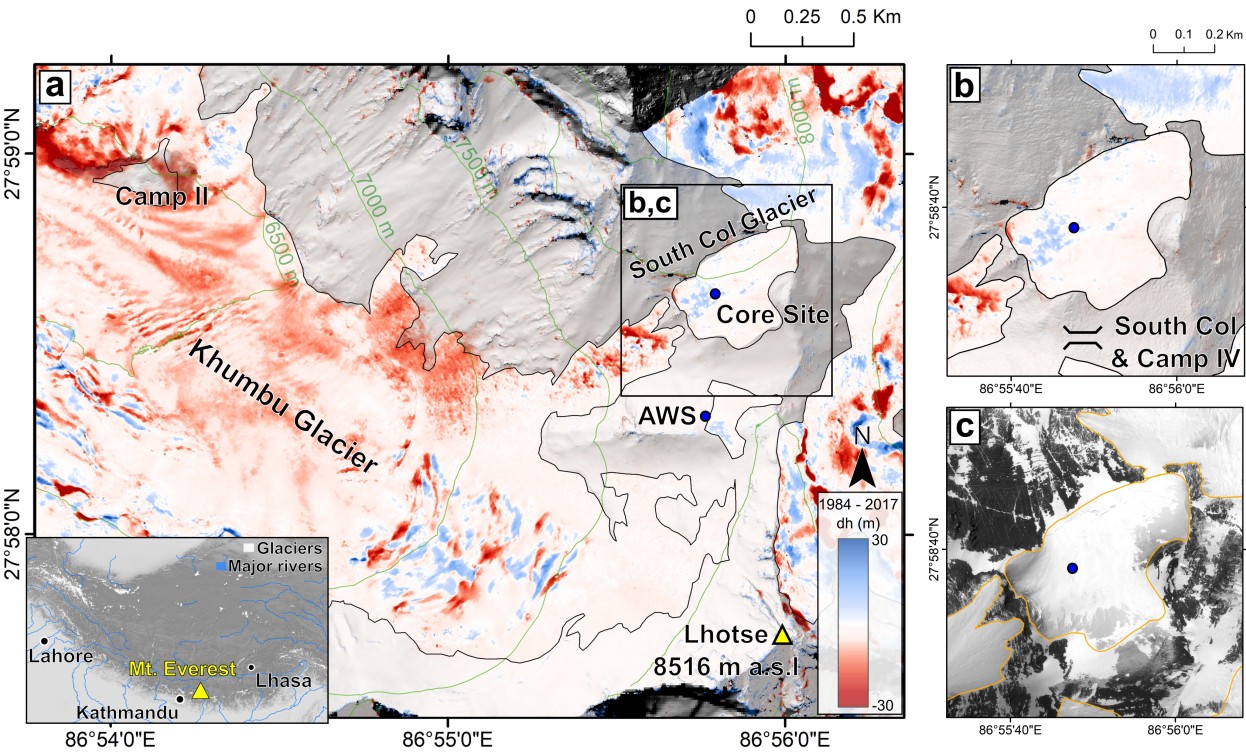

**Figure 1.** Surface elevation change over the Western Cwm (a) between 1984 and 2017, and over the South Col Glacier (b). The location of the ice core and AWS from Potocki et al. (2022) are shown with blue dots. Background is a shaded relief from the Pléiades DEM. The conditions at the surface of the South Col Glacier on the 23 March 2017 are captured by a Pléiades orthoimage in panel c (Pléiades, copyright CNES 2017, Distribution Airbus DS).

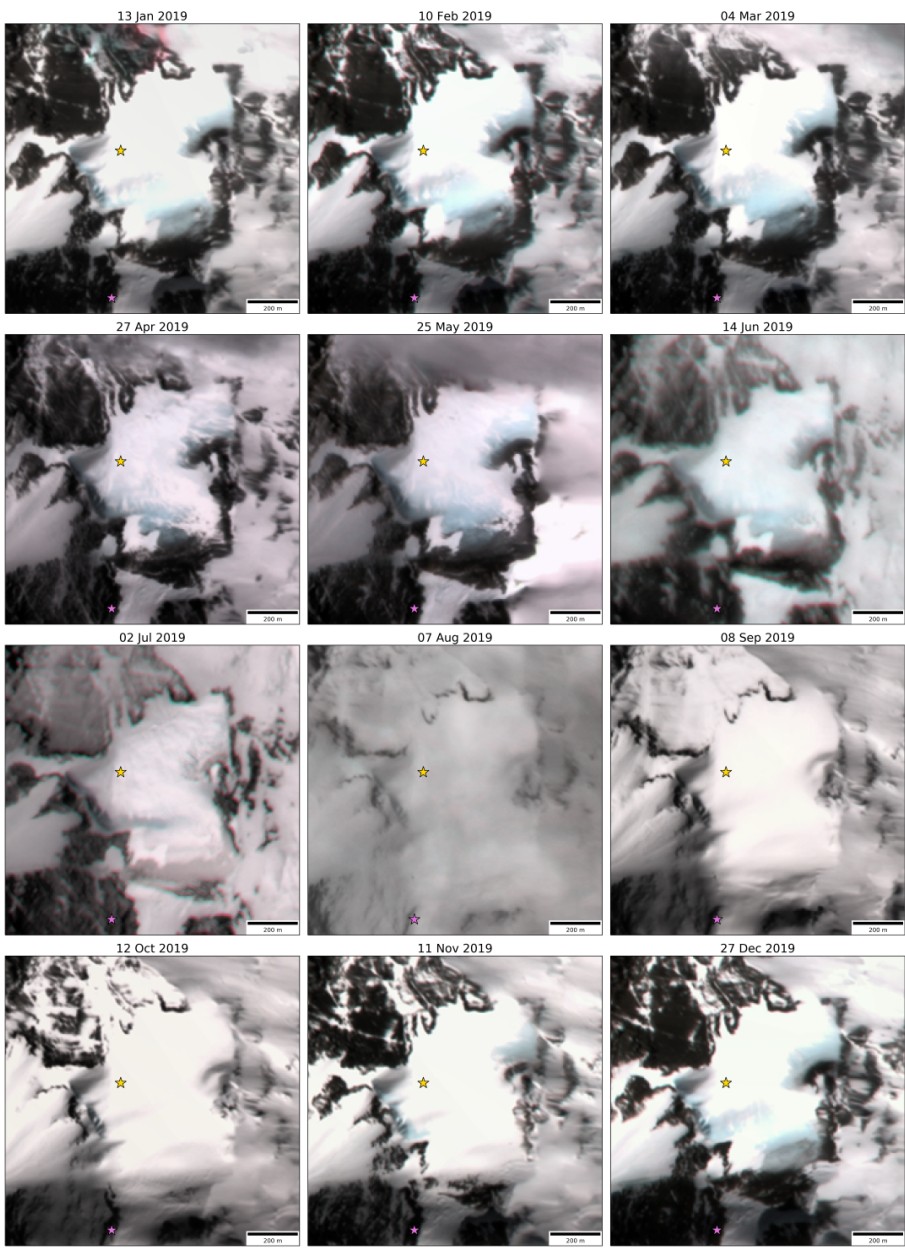

**Figure 2.** Natural color composites (7-4-3) of VENµS images (CC BY-NC 4.0) showing the seasonal variability of the surface snow cover of South Col Glacier. The yellow star shows the location of the drilling site and the purple one, the location of the South Col AWS. Note the striking expansion of blue ice areas outside the monsoon months, i.e. Oct. – Dec. 2019, or Jan. – April 2019.

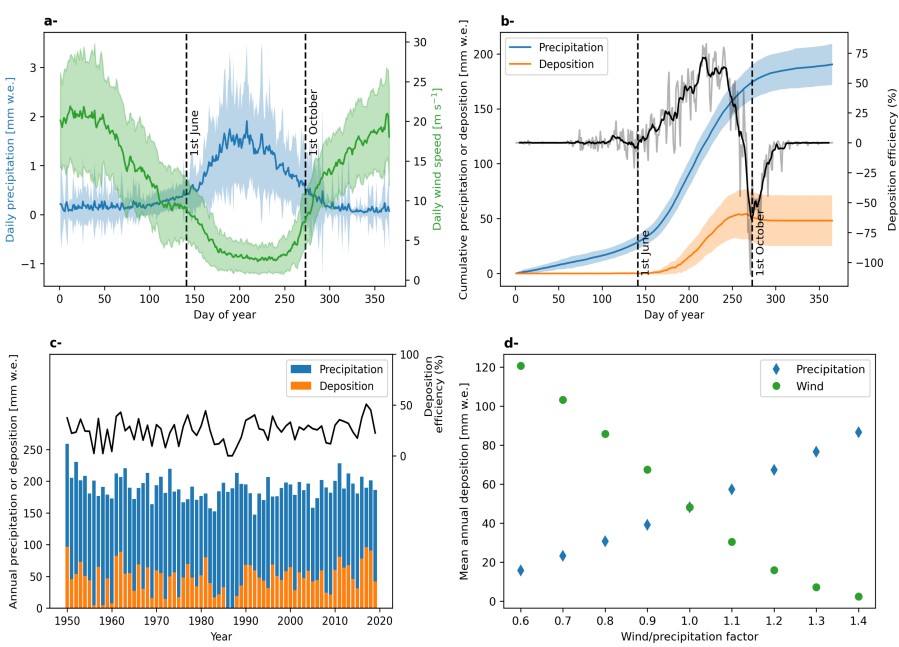

**Figure 3.** Seasonal cycle of wind and precipitation from downscaled ERA5 data (a) and cumulative precipitation/deposition from the wind model (b) for the period 1950-2020. The black curve on panel b shows the 9 day moving average of the daily ratio of deposition over precipitation. Panel c shows the annual values of precipitation, deposition and deposition efficiency. The model sensitivity to the wind and precipitation factors is shown on panel d, where the averages are calculated for the period 1950-2020.



**Figure 4.** Daily melt rate, sublimation rate and surface temperature simulated by COSIPY_P22 (dashed blue), COSIPY_grad (blue) and Crocus (orange) models for the year 2019. Daily ablation rates are computed from the hourly sum. The daily surface temperature corresponds to the maximum simulated during the day. Note that the initial thermal state is identical for these three simulation and is predicted by a COSIPY simulation similar to COSIPY_P22, that ran from 2010-01-01 to 2019-01-01.



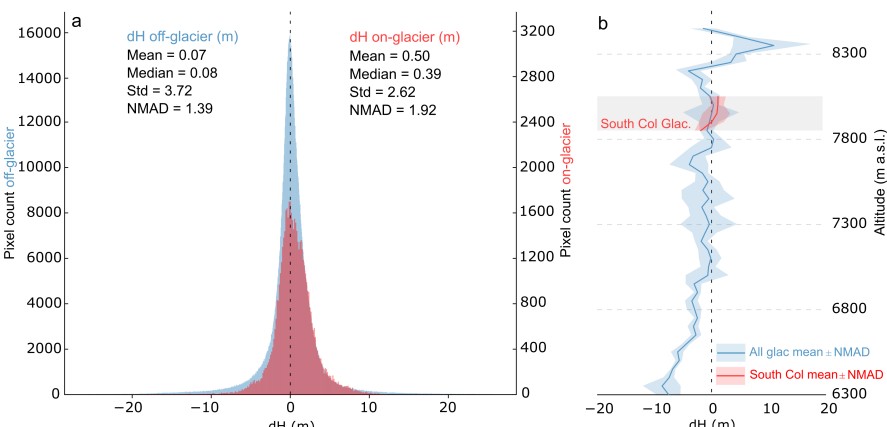

**Figure A1.** Distribution of the elevation change values off-glacier and on South Col Glacier (a). Elevation changes for 1984-2017 as a function of elevation for every 50 m elevation bin for the whole glacierized area and for South Col Glacier (b).

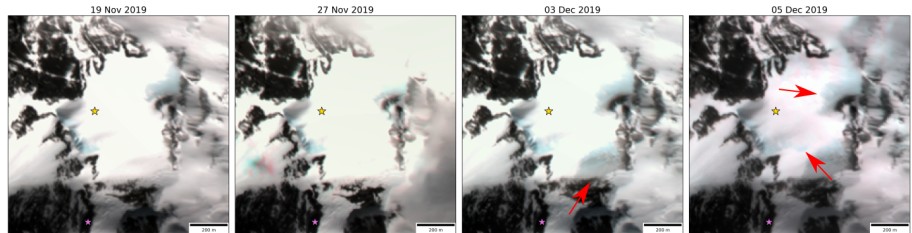

**Figure A2.** Natural color composites (7-4-3) of VENμS images (CC BY-NC 4.0) showing episodes of snow cover loss in November and December 2019. the yellow star shows the location of the drilling site and the purple one, the location of the South Col AWS. The red arrows point to the area of large changes.



**Figure A3.** Day-of-year mean energy fluxes for the ice simulations with (a) COSIPY_P22, (b) COSIPY_grad and (c) CROCUS_ice for the year 2019.

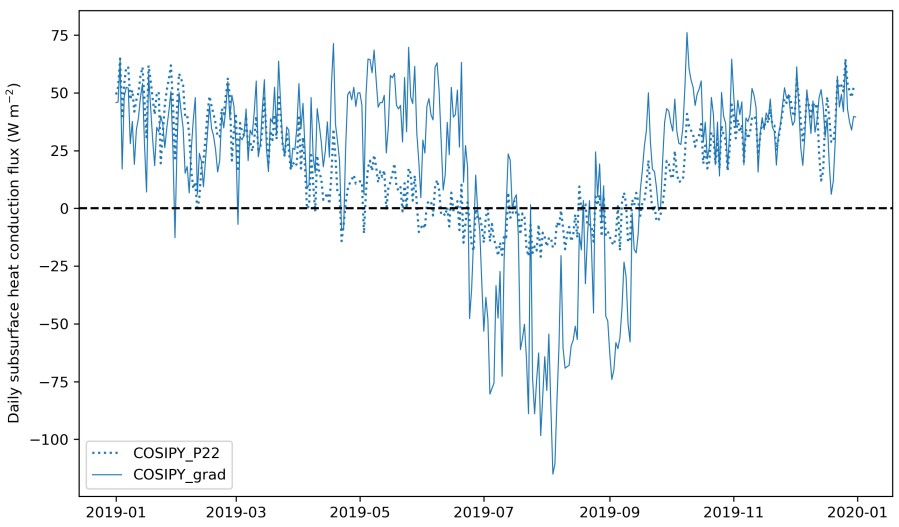

**Figure A4.** Daily averages of ground heat flux for the COSIPY_P22 and the COSIPY_grad simulations.

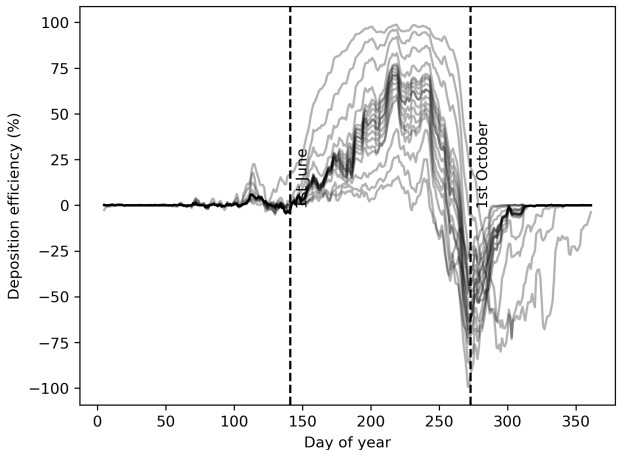

**Figure A5.** Deposition efficiency averaged over the period 1950-2020. Each line represents one simulation with a varying wind or precipitation factor. All the lines are smoothed with a 9 day running mean.



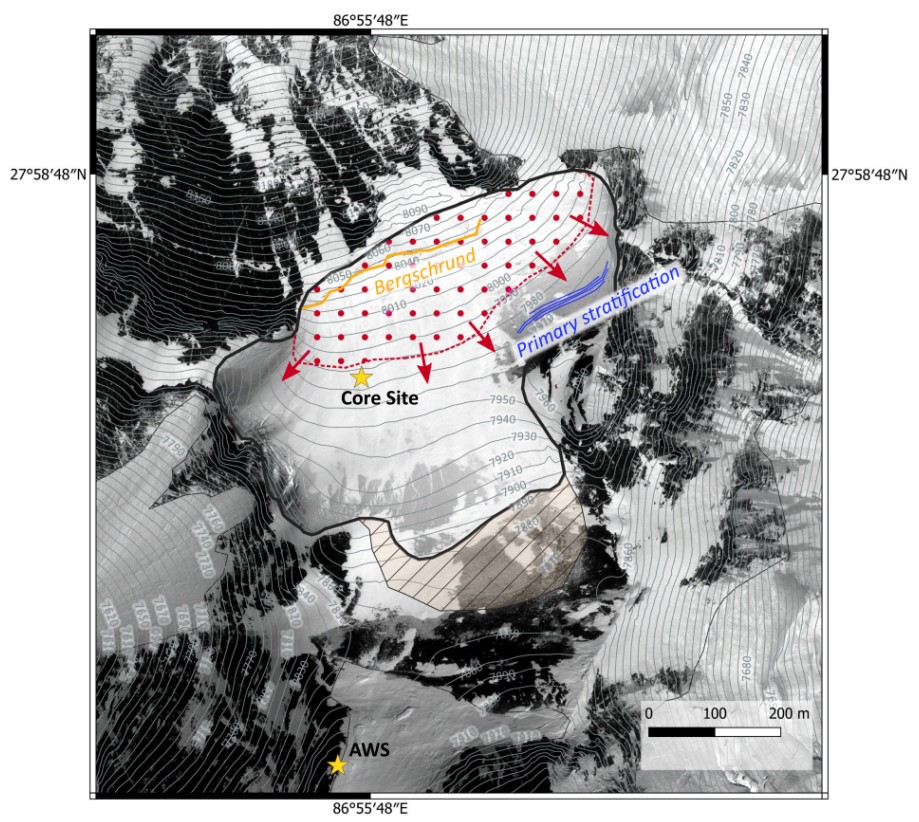

**Figure A6.** Pléiades orthoimage of South Col Glacier (23 March 2017), showing the location of some remarkable features. A tentative accumulation area of South Col Glacier is highlighted with dots. The dashed line is the separation between the accumulation and ablation area, along which the ice flux Φ is calculated. Red arrows show the direction of the ice flux. The shaded hashed area represent glacierized area that might belong to South Col Glacier, but it is not possible to conclude solely from satellite imagery. Contour lines in gray are plotted every 10 m and extracted from the Pléiades DEM, the elevation corresponds to height relative to the ellipsoid WGS84. Pléiades, copyright CNES 2017, Distribution Airbus DS.


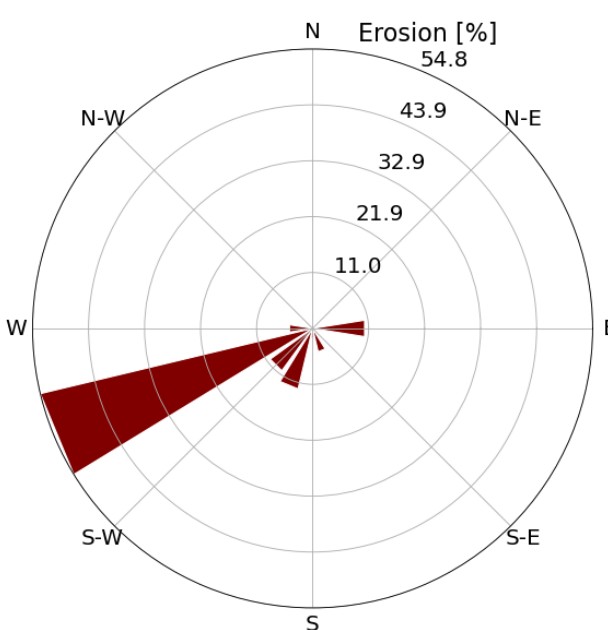

**Figure A7.** Erosion as a function of wind direction for the period where wind data are available from the AWS, showing the dominant role of WSW winds.