# Peer review of "Brief communication: Everest South Col Glacier did not thin during the last three decades"

_The Cryosphere, 2022_

## Referee Comment (RC1)

The brief communication by Brun et al. (2022) "*Everest South Col Glacier did not thin during the last three decades*" presents measurements of the mass change of Everest South Col Glacier between 1984 and 2017 from differencing of digital elevation models, and compares these to results from a set of sensitivity experiments using the COSIPY and CROCUS models. The manuscript is written in response to a paper by Potocki et al. (2022) which calculated mass loss from this glacier of 1.5 m w.e. a$^{-1}$ from analysis of an ice core and the COSIPY model. The Brun et al. (2022) study finds that mass change for this glacier is within uncertainty of zero. Everest South Col Glacier is a small (0.2 km$^2$) ice mass at 8,020 m a.s.l. on the southern side of Sagarmatha, located above the headwall of Khumbu Glacier and on the climbing route for this mountain from the south. While it is rare that so much effort is dedicated to determining the mass change of such a minor glacier, the location at extremely high elevation is used to justify the attempt with the implication that if glaciers are losing mass at present at the highest elevations in the Himalaya, then widespread mass loss is expected at all elevations.

In the first part of my review, I make several major comments and some minor (editorial) comments on the work by Brun et al. (2022) which I request that the authors address in revising their manuscript. In the second part of the review and because this work is a response to a previous study, I compared and evaluated the results of both papers. The authors of Potocki et al. (2022) wrote a response to Brun et al. (2022) in this discussion, led by the second author rather than the more junior first author, which I discuss in the second part of my review and refer to as Mayewski et al. TCD.

**Review of Brun et al., 2022: Major comments**

1. Glacier mass change data. Brun et al. first present an analysis of DEM data to demonstrate that no mass loss has occurred from Everest South Col Glacier between 1984 and 2017 CE. This method is well established and has been thoroughly tested for glaciers in this location, notably in the recent paper by King et al. (2020) of which one of the co-authors of Petocki et al. (2022), Alexander Tait, is also an author and therefore aware of the method. It would have been justified to end the paper after these analyses, as the results are convincing and of greater value than the modelling for the reasons outlined below.

These results could be described in more detail to make this section more accessible to a wider audience. Mayewski et al. TCD have interpreted areas of negative surface elevation change in the upper accumulation area of Khumbu Glacier as mass loss indicative of glacier wide mass balance rather than redistribution of mass within the glacier.

2. Surface energy-mass balance modelling. Calculating the surface energy balance of a glacier in this setting is extremely challenging, as extreme winds strongly affect the accumulation and removal of snow from the glacier surface, melt processes are dominated by sublimation, and the influence of the Indian Summer Monsoon on glacier mass balance is unknown at these elevations. I consider this glacier an unsuitable candidate for any surface energy-mass balance modelling study unless a model was developed specifically for this location and constrained by detailed and representative atmospheric and glaciological data (i.e., collected at the site of this glacier over several years, rather than using empirically derived values from other settings).

Brun et al. (2022) have taken a pragmatic approach by reproducing the COSIPY model parameterisation used by Potocki et al. (2022) in a sensitivity test that considers a graduated mass balance parameterisation of the same model and a comparison with results from a snow model, CROCUS. Their results demonstrate that the simulated glacier mass change is sensitive to the model time step used, and that there are large uncertainties associated with such calculations. The model results are useful as a comparison with the approach of Potocki et al. (2022), but I suggest that the modelling work from both papers is phrased more cautiously; as potentially useful to identify where the largest uncertainties arise in estimating the mass balance of South Col Glacier, but unlikely to accurately represent glacier change.

Brun et al. (2022) also used a snow deposition model to quantify the impacts of wind on snow accumulation at this glacier and determine when the ice surface is free of snow and hence may melt. These results are compared with satellite imagery and show good agreement. This model application is more valuable than COSIPY for investigating South Col Glacier, but still contains large uncertainties. As discussed below, these results illustrate the limitations of the interpretation of the ice core data by Potocki et al. (2022).

3. Description of glacier geometry. I find it strange that Brun et al. (2022) assign an ablation area to South Col Glacier in Figure A6 and would expect emergence of ice to be minimal for this glacier as they predict. As discussed below, this interpretation seems rather strange based on the glacier's elevation relative to the local ELA. I suggest revising this figure and reframing the interpretation of the glacier as an accumulation area only.

**Review of Brun et al., 2022: Minor comments**

Line 1 and 8: Is the glacier "iconic"? It's very high, but otherwise I suggest it is not widely known.

L9: remove "large" as this is relative to the glacier in question; "…glaciers thin at rates often exceeding…".

L14 and elsewhere: check use of compound adjectives; hyphenation is not used with an adverb (ending in "-ly").

L24: "challenge of conducting scientific…"

L28: worth noting here that the South Col AWS recorded only about five months of data (May–end summer 2019). An earlier AWS at this location installed by the Ev-K2-CNR project measured three years of discontinuous data that did represent the entire annual cycle and could be of use if further field data are required.

**Comparison of the results and conclusions of Brun et al. (2022) with those of Potocki et al. (2022), in consideration of the response by Potocki et al. in TCD.**

A key question addressed by both papers is; what is the duration of snow cover on the glacier surface? This would indicate when the bare ice surface is exposed to incoming solar radiation and ice melt could occur. However, the occurrence of seasonal melt does not imply net annual mass loss. Determining mass change over a representative timescale of several decades requires observations of longer-term change as provided by both papers.

In the case of both papers, I consider that the COSIPY model is unsuitable for application to South Col Glacier and the associated uncertainties render the results insignificant. My group's

ongoing work applied COSIPY to Khumbu Glacier including the area occupied by South Col Glacier (https://doi.org/10.5194/egusphere-egu21-8663). COSIPY was forced by downscaled CORDEX RCM outputs and constrained by AWS data including the five months of data from the Nat Geo South Col AWS (Matthews et al., 2020). In each simulation, the net annual mass balance at the location of South Col Glacier was strongly positive (>7 m w.e. a$^{-1}$). We can debate the strengths and limitation of any of these model parameterisations but any existing glacier surface energy-mass balance model is unlikely to be suitable for South Col Glacier due to the significant differences in the processes that control mass balance at 8,000 m a.s.l. compared to glaciers for which these models were developed at lower elevations where the mass balance is better understood by established glaciological theory. The different datasets used by each study (e.g., DEMs of difference/an ice core) are more important indicators of glacier mass change.

More important than debating the parameterisation of models that are likely not meaningful, we should consider the glaciological context of South Col Glacier. The elevation of the glacier is about 2,000 m above the equilibrium line altitude (ELA) for this region, determined for Khumbu Glacier as about 6,000–6,400 m a.s.l. (Rowan et al., 2015; 2021). While glaciers usually melt during the ablation season due to warm air temperatures and high incoming solar radiation, this does not equate to mass loss year-on-year. It is difficult to see why a glacier 2,000 m above the local ELA would have a net annual negative mass balance. Mayewski et al. TCD refute the suggestion by Brun et al. (2022) that their core is collected from the glacier ablation area. As the entire glacier is located well above the local ELA, the entire glacier should be "accumulation area" and therefore as a small cold-based glacier, the mass of South Col Glacier is likely to remain stable over decadal timescales.

Mayewski et al. TCD refer to the glacier as have a stagnant area. This term is used to describe the tongues of debris-covered glaciers such as Khumbu Glacier where the velocity of ice flow has declined rapidly as the glacier has lost mass in recent decades. The term is not accurately applied by Mayewski et al. TCD in context of South Col Glacier, which has not undergone a change in glacier dynamics but instead has a typical (slow, deformation only) flow regime as a cold-based glacier.

As referenced by Mayewski et al. TCD, Figure 1 of Brun et al. (2022) shows areas of negative surface elevation change in the upper accumulation area of Khumbu Glacier. These bands are often interspersed with bands of positive surface elevation change, and I would interpret that they are evidence of large avalanches onto and within the glacier, and opening or closing of crevasses close to the bergschrund. Again, it is the net annual mass balance over a representative period of years (i.e., the integration of these features across the entire glacier) that tells us if the glacier is losing mass. These features in the DEMs of difference are not evidence of glacier mass loss but represent mass redistribution within the glacier. The mass gain of >7m a−1 predicted by our COSIPY simulations of South Col Glacier indicate the source of these avalanches—75% of accumulation to Khumbu Glacier and neighbouring valley glaciers occurs by avalanching of snow from the steep slopes, in some cases initiated by wind erosion of snow at the ridge crests (Benn and Lehmkuhl, 2000).

Potocki et al. (2022) interpret their ice core as representing the accumulation area of the glacier and the age of the ice collected near the glacier surface (0.1–0.7 m core depth) is about 2,000 years old ("1966 ± 179 years ago"). This period is then multiplied by the annual layer thickness for the entire core (27 mm w.e. a$^{-1}$) to estimate mass loss (apparently without any correction from water equivalent to ice thickness accounting for ice density?). This calculation

assumes that the age of the ice at the glacier surface is the same as at the depth measured in the core and that the annual layer thickness is consistent throughout. From their Supplementary Information, it appears that annual layers were only measured in a 0.1 m section of the core at about 6 m depth. The representativeness of these values is determined by comparison with a core from East Rongbuk Glacier at 6,518 m (Kaspari et al., 2009). However, the annual layer thickness at South Col Glacier could be much thinner if wind erosion is accounted for. The snow deposition model results from Brun et al. (2022) suggest that at South Col Glacier nearly all precipitation can be eroded from the glacier surface by wind, which would not be the case at East Rongbuk Glacier or Khumbu Glacier where the majority of accumulation is sourced from avalanching. It is therefore possible that since the Sol Col Glacier last expanded and formed the moraines identified by Petocki et al. (2022) that most or all of the annual snow accumulation is scoured off by wind and that the exposed ice surface represents the last period when the glacier expanded. These moraines are undated at South Col Glacier, but there are three possible equivalent ice-marginal moraine ages at Khumbu Glacier dated to 1.3 ± 0.1 ka, 0.9 ± 02 ka and 0.6 ± 0.16 ka (Hornsey et al., 2022).

The 'space-for-time substitution' suggested by Mayewski et al. TCD reasons that because snow melts at Camp 2 in April–May then ice must melt at South Col Glacier in July–August. Their photographs show melt water on the surface of the accumulation area of Khumbu Glacier at Camp 2 (6,464 m a.sl.) on a patch of rock debris. The low albedo of the debris combined with high incoming solar radiation is likely to promote snow melt, but again is not evidence for net annual glacier mass change. The substitution reasoning is not convincing; South Col is up to 20 degrees colder based on the ERA data presented by Mayewski et al. TCD and the incident radiation would presumably be much lower than given here due to monsoon cloud cover. The uncertainties in this estimate of seasonal melt seem similar or greater than those in the COSIPY experiments.

In summary, there are limitations to both the DEM differencing and ice core methods. I am convinced that both papers present their results accurately and have not made errors in their data processing. However, the DEM differencing presented by Brun et al. (2022) quantifies mass change across the glacier over a representative period of several decades. The ice core collected by Potocki et al. (2022) represents only one point on the glacier and is open to an alternative interpretation in context of the erosion of snow by wind from the glacier surface. I suggest that the model results are discounted as indicative of glacier mass balance due to the limitations of simulating this extreme environment without a dedicated model driven by spatially and temporally representative measurements from South Col Glacier. The remaining question is if there is value or feasibility in collecting direct glaciological measurements from South Col Glacier. Installing and managing equipment over a sufficient timescale (>5 years) would be very challenging and expensive. We therefore need to rely on high-quality remotely sensed observations, including those presented by Brun et al. (2022), which in the last 15 years or so have greatly improved understanding of recent glacier change in the Himalaya.

**References**

Benn DI, Lehmkuhl F. 2000. Mass balance and equilibrium-line altitudes of glaciers in high-mountain environments. Quaternary International 65–66 : 15–29. DOI: 10.1016/S1040-6182(99)00034-8

Brun, F., King, O., Réveillet, M., Amory, C., Planchot, A., Berthier, E., Dehecq, A., Bolch, T., Fourteau, K., Brondex, J., Dumont, M., Mayer, C., and Wagnon, P. 2022. Brief communication: Everest South Col Glacier did not thin during the last three decades, The Cryosphere Discussions. [preprint], https://doi.org/10.5194/tc-2022-166.

Hornsey J, Rowan AV, Kirkbride MP, Livingstone SJ, Fabel D, Rodes A, Quincey DJ, Hubbard B, Jomelli V. 2022. Be-10 Dating of Ice-Marginal Moraines in the Khumbu Valley, Nepal, Central Himalaya, Reveals the Response of Monsoon-Influenced Glaciers to Holocene Climate Change. Journal of Geophysical Research: Earth Surface 127 DOI: 10.1029/2022JF006645

Kaspari S, Hooke RLeB, Mayewski PA, Kang S, Hou S, Qin D. 2008. Snow accumulation rate on Qomolangma (Mount Everest), Himalaya: synchroneity with sites across the Tibetan Plateau on 50–100 year timescales. Journal of Glaciology 54 : 343–352. DOI: 10.3189/002214308784886126

King O, Bhattacharya A, Ghuffar S, Tait A, Guilford S, Elmore AC, Bolch T. 2020. Six Decades of Glacier Mass Changes around Mt. Everest Are Revealed by Historical and Contemporary Images. One Earth 3: 608–620. DOI: 10.1016/j.oneear.2020.10.019

Matthews, T., Perry, L. B., Koch, I., Aryal, D., Khadka, A., Shrestha, D., Abernathy, K., Elmore, A. C., Seimon, A., Tait, A., Elvin, S.,Tuladhar, S., Baidya, S. K., Potocki, M., Birkel, S. D., Kang, S., Sherpa, T. C., Gajurel, A., and Mayewski, P. A.: Going to Extremes: Installing the World's Highest Weather Stations on Mount Everest, Bulletin of the American Meteorological Society, 101, E1870–E1890, https://doi.org/10.1175/BAMS-D-19-0198.1, 2020.

Potocki, M., Mayewski, P.A., Matthews, T. et al. Mt. Everest's highest glacier is a sentinel for accelerating ice loss. npj Climate Atmospheric Sciences 5, 7 (2022). https://doi.org/10.1038/s41612-022-00230-0

Rowan AV, Egholm DL, Quincey DJ, Glasser NF. 2015. Modelling the feedbacks between mass balance, ice flow and debris transport to predict the response to climate change of debris-covered glaciers in the Himalaya. Earth and Planetary Science Letters 430 : 427–438. DOI: 10.1016/j.epsl.2015.09.004

Rowan AV, Egholm DL, Quincey DJ, Hubbard B, King O, Miles ES, Miles KE, Hornsey J. 2021. The Role of Differential Ablation and Dynamic Detachment in Driving Accelerating Mass Loss From a Debris-Covered Himalayan Glacier. Journal of Geophysical Research: Earth Surface 126 DOI: 10.1029/2020JF005761

*Ann Rowan, University of Bergen, Norway. 10/10/22*

---

## Referee Comment (RC4)

**Review of "Brief communication: Everest South Col Glacier did not thin during the last three decades" by Brun et al.**

**Horst Machguth and Enrico Mattea,** Department of Geoscience, University of Fribourg, Switzerland

**1. Introduction**

The study of *Brun et al.* argues that South Col Glacier has not changed substantially since 1984. Their finding contradicts *Potocki et al.* (2022) – in the following *Potocki et al.* – who claim that the glacier has thinned at their drill site (8020 m a.sl.) by about 55 m. The timing of the thinning is unclear but it is suggested by Potocki et al. that the climate at that elevation warmed substantially in the 1950s and even more substantially in the later 1990s. We divide out review in a general section which refers to both studies, and a section of detailed comments focusing on *Brun et al.*

**2. General comments on South Col Glacier changes**

*Brun et al.* contradict that South Col Glacier has thinned dramatically. They do so by comparing two digital elevation models representing different points in time. The evidence provided by *Brun et al.* appears sound and corresponds to state-of-the-art. Nevertheless, we chose an independent way of assessing whether the glacier has changed or not. To do so, we compared historical photos of South Col Glacier to recent images. We obtained images taken during the 1956 Swiss Everest/Lhotse expedition; the second ever expedition to summit Everest and the first to summit Lhotse. The images are publicly accessible at https://alpinfo.ch/en/portrait/historical-notes/expeditions/everest-lhotse-1956/ . We have also considered photos from the two Swiss 1952 Everest expeditions (also publicly available via the above link), but found that images from 1956 are optimal for the comparison. We compare two historical images from 1956 to recent images. Figure 1 shows the perspective from somewhere in the vicinity of the South Col AWS, looking slightly down on the plateau of South Col Glacier. Figure 2 shows the view towards the tongue of South Col Glacier, looking upward from what was maybe Camp V or VIa during the 1956 expedition.

[Figure]

[Figure]

**Fig. 1:** *Comparison of a historical image of South Col Glacier, recorded in May 1956 (left) to an image taken in May 2022 (right). For scale an approximate elevation difference has been drawn based on the Pléiades DEM by Brun et al. Note that the two images were taken from two different viewpoints, with the 1956 picture taken from a point somewhat more east and closer to South Col. Image courtesy of the Swiss Foundation of Alpine Research (1956) and Tim Mosedale (2022, https://timmosedale.co.uk/).*

The comparison in Fig. 1 shows, if at all, small changes in South Col glacier. There is no support for the claim of *Potocki et al.* that the glacier has thinned in excess of 50 m. The near-absence of changes is most obvious at the glacier margins. Admittedly, changes are more difficult to assess in the glacier centre where *Potocki et al.* have drilled. Nevertheless, the glacier appears also to have changed little at the drill site. If the glacier would have thinned 55 m in the centre, then one would expect an even more pronounced change at its tongue. Such a thinning would be obvious as the ice cliff is in close proximity to the drill site (less than 200 m apart). However, recent imagery shows that the glacier tongue is at the same location as it was in 1956 (Fig. 2) and the ice thickness appears unchanged. The tongue appears similarly active as in the 1950s (see for example an excellent 2022 overview of South Col Glacier: https://www.mountainpanoramas.com/___p/___p.html?panoid=2022_M1&labels=on).

[Figure]

[Figure]

**Fig. 2:** Tongue of South Col Glacier in 1956 and 2008. *Left: detail of a historical image of South Col Glacier, recorded in May 1956 by the Swiss Everest/Lhotse expedition. Right: detail of an image taken in May 2008. Note that the two images were taken from somewhat different viewpoints, with the 1956 image taken from a higher elevation. Image courtesy of the Swiss Foundation of Alpine Research (1956) and* [https://spaceref.com/science-and-exploration/scott-parazynski-everest-photo-update-4-june-2008](https://spaceref.com/science-and-exploration/scott-parazynski-everest-photo-update-4-june-2008) *(2008).*

We investigated whether glacier changes are detectable near the actual South Col where the glacier surface appears very flat and the ice looks rather thin (directly east (right) of the "Everest South Col" marker in the linked panoramic photograph). Interestingly, also there the situation in 1956 appears similar to more recent photos (not shown). We note that there are many good images available for South Col, for different points in time, providing excellent possibilities to investigate whether glacier changes took place or not.

**3. General comments on the surface mass balance modelling**

We argue that uncertainties in available model input parameters are too large to make any reliable statement on the mass and energy balance at South Col Glacier, based on model simulations. These uncertainties also affect the model comparison presented in *Brun et al.* We detail our argumentation on the example of ice albedo at South Col and by running a third surface mass and energy balance model for South Col Glacier.

**The problematic of parameter uncertainties:** Both *Brun et al.* and *Potocki et al.* use an ice albedo value of 0.4. The value has been measured at Base Camp, at approximately 5400 m a.sl. (*Matthews et al.,* 2020). The ice at the surface of South Col Glacier is referred to as blue ice (*Brun et al.*). Albedo

values for Antarctic blue ice are substantially higher, typically in the range of 0.6 to 0.65. (e.g. *Reijmer et al.* 2001; *Genthon et al.* 2007; *Smedley et al.* 2020). While Antarctica's blue ice areas might not be representative for South Col blue ice, it is also questionable to use an albedo value measured on glacier ice 2500 m lower than South Col. At Base Camp melt processes and surface ice conditions differ substantially from South Col. Also, the glacier ice has been formed under different conditions (cold on South Col vs. possibly temperate at Khumbu glacier), further affecting ice albedo.

Both *Potocki et al*. and *Matthews et al.* (2020) do not perform sensitivity tests with ice albedo, regardless of the extreme importance their studies assign to short wave radiation. In the case of South Col, critical uncertainties are not limited to surface albedo alone. A parameter sensitivity study is mandatory as soon as some parameters have relevant uncertainties. In the case of South Col, even a simple sensitivity study, as demonstrated below, might show that the range in possible outcomes of model simulations is simply too large for model results to be deemed reliable.

A thorough sensitivity analysis is also missing in *Brun et al.* While COSIPY is used in two different constellations of model numerics, CROCUS is not subject to any sensitivity assessment. Sensitivity to uncertainties in model input has neither been evaluated for the COSIPY variants nor for CROCUS. We understand that this is beyond the scope of the study by *Brun et al.* They also clearly state that uncertainties in any simulation for South Col are too large, given our current knowledge of meteorological conditions and mass balance. Nevertheless, we gained the impression that *Brun et al.* somewhat consider CROCUS the benchmark for other models. While this could be true, it would require demonstrating that CROCUS is more robust to changes in model numerics and comparing sensitivity of all models to input parameter perturbations. In this sense, we would like to ask *Brun et al.* to check their manuscript for any explicit or implicit "model hierarchy" and to further emphasize the problematic of poorly constrained model input and the absence of a parameter sensitivity study.

**Simulating South Col Glacier surface mass and energy balance using EBFM:** The energy balance and firn model (EBFM, *van Pelt et al.,* 2012) was developed following *Klok and Oerlemans* (2002) and the subsurface model SOMARS by *Greuell and Konzelmann* (1994). The model has recently been modified for use on Abramov Glacier, Kyrgyzstan (*Kronenberg et al.,* 2022) and for Colle Gnifetti, Swiss Alps (*Mattea et al.,* 2021). Here we deploy the version by *Mattea et al.* (2021) for a series of South Col model sensitivity experiments.

The model uses a skin layer formulation, calculating surface energy fluxes from meteorological variables. The surface energy balance equation is solved for surface temperature and mass fluxes, including melt and sublimation rates. Surface albedo is bounded by constant values for fresh snow, firn, and ice (respectively $\alpha_{fresh}$, $\alpha_{firn}$ and $\alpha_{ice}$); it evolves as an exponentially decaying function of time since the last significant snowfall (defined by a minimum precipitation rate $P_{min}$).

At each time-step, the computed surface boundary conditions drive a Lagrangian simulation of the glacier subsurface: it consists of a stack of $NL$ layers able to move freely along the depth axis, following the addition or removal of mass at the surface. A new layer is added at the top whenever snowfall and riming push the topmost layer thickness beyond threshold $z_s$.

Notable omissions in the EBFM include penetration of short-wave radiation and wind erosion of snow (less significant for the simulation of an ice surface). As in COSIPY and Crocus, terrain reflections and topographic shading are also ignored; they are expected to play a minor role in the overall energy balance (e.g. *Mattea et al., 2021*).

We use the same downscaled ERA5 data of *Potocki et al.* and *Brun et al.* to force the model. Table 1 reports the main results of our sensitivity runs.

**Table 1:** summary of sensitivity EBFM runs. Shaded column headings indicate model parameters, the other columns are model results. Melt $M$ is shown in **bold**.

| Id | Period | Spinup | NL | $P_{min}$ | $z_s$ | $\alpha_{fresh}$ | $\alpha_{firn}$ | $\alpha_{ice}$ | M | S | D | E | C | $\alpha_{mean}$ | $Q_g$ |
|---|---|---|---|---|---|---|---|---|---|---|---|---|---|---|---|
| 1 | 2000-2019 hourly | None | 50 | 2.5e-8 | 0.10 | 0.83 | 0.52 | 0.39 | **0.0007** | 0.241 | 0.012 | 0.000 | 0.000 | 0.80 | 0.25 |
| 2 | 2000-2019 hourly | None | 50 | 2.5e-6 | 0.10 | 0.83 | 0.52 | 0.39 | **0.7750** | 0.428 | 0.004 | 0.062 | 0.000 | 0.47 | 3.2 |
| 3 | 2000-2019 hourly | None | 50 | 2.5e-6 | 0.01 | 0.83 | 0.52 | 0.39 | **0.5948** | 0.489 | 0.002 | 0.039 | 0.000 | 0.45 | 17 |
| 4 | 2000-2019 hourly | None | 2000 | 2.5e-6 | 0.01 | 0.83 | 0.52 | 0.39 | **0.5201** | 0.487 | 0.002 | 0.036 | 0.000 | 0.46 | 18 |
| 5 | 2000-2019 hourly | None | 500 | 2.5e-6 | 0.01 | 0.83 | 0.52 | 0.39 | **0.5529** | 0.493 | 0.002 | 0.038 | 0.000 | 0.46 | 20 |
| 6 | 2019 minutely | None | 500 | 2.5e-6 | 0.01 | 0.83 | 0.52 | 0.39 | **0.3539** | 0.327 | 0.005 | 0.036 | 0.000 | 0.63 | 6.9 |
| 7 | 2019 hourly | None | 500 | 2.5e-6 | 0.01 | 0.83 | 0.52 | 0.39 | **0.3520** | 0.342 | 0.005 | 0.032 | 0.000 | 0.63 | 11 |
| 8 | 2000-2019 hourly | None | 500 | 2.5e-6 | 0.01 | 0.83 | 0.60 | 0.60 | **0.0490** | 0.355 | 0.007 | 0.007 | 0.000 | 0.65 | 7.8 |
| 9 | 1950-1999 hourly | None | 500 | 2.5e-6 | 0.01 | 0.83 | 0.60 | 0.60 | **0.0282** | 0.323 | 0.006 | 0.005 | 0.000 | 0.65 | 6.8 |
| 10 | 2000-2019 hourly | 50 yr, run 9 | 500 | 2.5e-6 | 0.01 | 0.83 | 0.60 | 0.60 | **0.0178** | 0.351 | 0.007 | 0.004 | 0.000 | 0.65 | 7.8 |
| 11 | 1950-1999 hourly | None | 500 | 2.5e-6 | 0.01 | 0.83 | 0.52 | 0.39 | **0.3486** | 0.439 | 0.003 | 0.026 | 0.000 | 0.48 | 13 |
| 12 | 2000-2019 hourly | 50 yr, run 11 | 500 | 2.5e-6 | 0.01 | 0.83 | 0.52 | 0.39 | **0.4609** | 0.480 | 0.003 | 0.032 | 0.000 | 0.46 | 14 |

**Notes:**
1. *M, S, D, E* and *C* are annual means of melt, sublimation, deposition, evaporation and condensation. $\alpha_{mean}$ is the mean surface albedo. $Q_g$ is the mean annual subsurface heat flux.
2. $P_{min}$ is in m w.e. s$^{-1}$; $z_s$ in m; *M, S, D, E, C* in m w.e. yr$^{-1}$; $Q_g$ in W m$^{-2}$; all other parameters are dimensionless.
3. Changes in model setup from one model run to the next are highlighted.

The ERA5 meteorological series contains extremely frequent, small precipitation events, which constantly reset surface albedo to the fresh snow value ($\alpha_{mean}$ in run 1). As such, in runs 2 through 12 we increase $P_{min}$ by 100x, to effectively disable the albedo from being restored in most cases.

With a layer thickness limited at 10 cm and a time-step of 1 hour (run 2), the EBFM calculates mean annual melt amounts of 0.78 m w.e. over 2000-2019, which corresponds to half of the values

reported by *Potocki et al.*; notably, mean albedo over the modeled period is 0.47, which is about 25 % lower than Antarctic blue ice albedo.

Even before altering physical parameters, we note that the numerical setup has a significant impact on model results: forcing the use of 10x thinner grid layers (run 3), computed melt amounts drop by about 30 %. (Mean subsurface heat flux also has a five-fold increase, but the value remains reasonably low – unlike what is reported for COSIPY by *Brun et al.*). The maximum depth of simulation also affects the results somewhat (runs 3, 4, 5): annual melt amounts increase for shallower grids, from 0.52 m w.e. (at 20 m maximum depth), to 0.55 and 0.59 (respectively at 5 m and 50 cm). Unlike the COSIPY result by *Brun et al.*, a finer time resolution of 1 minute (with linearly interpolated climate variables) does not reduce melt at all in the EBFM (runs 6 and 7).

Most importantly, in run 8 we test a standard value of 0.60 for the albedo of blue ice. This simple change reduces melt rates by 90-95 % compared to the glacier-ice default value of 0.39 (run 5). A further reduction by more than 60 % occurs when running the model after a spin-up period of 50 years (run 10). The latter observation also holds true for simulations with the default (lower) albedo values (run 12).

Such a high sensitivity to the albedo parameters indicates a very high degree of uncertainty in the simulated energy balance at South Col, and raises serious concerns on the applicability of any albedo values not measured *in situ*.

Computed sublimation rates in the EBFM are in all cases comparable (20-50 cm w.e. yr$^{-1}$) to the results of *Potocki et al.* and *Brun et al.* Still, parameters involved in the calculation of turbulent fluxes (such as surface roughness lengths) are known to be poorly constrained, especially in high accumulation areas (e.g. *Mattea et al., 2021*). Therefore, if sublimation plays a major role in the surface mass fluxes at South Col, its modeling uncertainties are likely also significant for the overall error budget, and should be investigated.

In conclusion, also a skin-layer, or skin-temperature model appears to be able to predict, in its basic configuration, no melt for the South Col Glacier. Relatively small perturbations in model parameters, however, are sufficient to change model output substantially, reaching from almost zero up to ~50% of the melt simulated by *Potocki et al.*

**4. Detailed comments on the manuscript by *Brun et al.**

Line 26: We suggest spelling out JJAS where it is first mentioned.

Line 36: This communication is not brief. Depending on the editorial guidelines, it could also be published as a normal paper.

Lines 54-57: Could you quantify "as the range of elevation change values were higher here"? How are "minor data voids" defined?

Lines 156/157: The article by *Brun et al.* criticises most results from *Potocki et al*. However, here one of their results is cited as if correct, to support the argumentation by *Brun et al*. For the purpose of assessing the study, we suggest, to argue based on independent results also where the results from *Potocki et al.* fit the own argumentation.

Line 177: Citation: It appears that this is basic knowledge generally understood. We would like to ask the authors to cite a more original reference.

Lines 262/263: Please add at least one more original citation or remove the citation here. It is long known how ice dynamics transport ice from the accumulation to the ablation area.

Lines 278-289: We do not fully understand the idea behind this calculation. The authors first state that the glacier is in balance, as shown by the DEM differencing. Then the ablation values of *Potocki et al.* are used to estimate at which speed the ice would need to flow so both conditions are fulfilled, that is (i) glacier is in equilibrium and (ii) ablation is ~2 m a$^{-1}$. But *Potocki et al.* do not claim that the glacier is in equilibrium, hence we do not understand what is supposed to be shown here.

Line 290-293: The argumentation could be clearer. The motivation of the previous paragraph is already unclear to us (see above). Then, without an introduction or explanation, another method, based on other assumptions, is used to estimate flow velocity in South Col glacier.

Line 293: Is the term "continental" the right term here? It is a monsoon influenced glacier, likely cold and frozen to the bed. While indeed arid (albeit not only because of low precipitation but also because of strong wind erosion), annual fluctuations in air temperature, characteristic for continental glaciers, are not particularly large (e.g. Suppl. Fig. 8 in *Potocki et al.*).

Figures 1 and A6: *Mayewski et al.* state in their comment on *Brun et al.* that the latter placed the drill site at the wrong location. It appears to us that the drill location as visualized by *Brun et al.* is correct, the only difference being that *Brun et al.* express elevation in meters above the ellipsoid while *Potocki et al.* use elevation in meter above the geoid. Nevertheless, it appears that in Fig. 1 the contour lines are in meters above the ellipsoid (the drill site is slightly below 8000 m) while the elevation of Lhotse peak is given in meters above the geoid. The caption of the figure does not indicate which elevation datum was used. As readers might be familiar with elevations of South Col, Lhotse Peak and Everest, and these well-known numbers are in meters above sea level, we suggest that elevations on maps are expressed in relation to the same datum.

**Acknowledgements**

We thank Tim Mosedale (https://timmosedale.co.uk/) and the Swiss Foundation for Alpine Research (https://alpinfo.ch) for granting permission to use their photos of South Col Glacier in the context of this review.

**References**

Genthon, C., Lardeux, P. & Krinner, G. 2007. The surface accumulation and ablation of a coastal blue-ice area near Cap Prudhomme, Terre Adélie, Antarctica. *Journal of Glaciology*, **53**, 635–645.

Greuell, W. & Konzelmann, T. 1994. Numerical modelling of the energy balance and the englacial temperature of the Greenland ice sheet. Calculations for the ETH-Camp location (West Greenland, 1155 m a.s.l.). *Global Planet. Change*, **9**, 91–114.

Klok, E.J. & Oerlemans, J. 2002. Model study of the spatial distribution of the energy and mass balance of Morteratschgletscher, Switzerland. *J. Glaciol.*, **48**, 505–518.

Kronenberg, M., van Pelt, W., Machguth, H., Fiddes, J., Hoelzle, M. & Pertziger, F. 2022. Long-term firn and mass balance modelling for Abramov glacier, Pamir Alay. *The Cryosphere Discussion*.

Mattea, E., Machguth, H., Kronenberg, M., van Pelt, W., Bassi, M. & Hoelzle, M. 2021. Firn changes at Colle Gnifetti revealed with a high-resolution process-based physical model approach. The Cryosphere, 15, 3181–3205.

Matthews, T., Perry, L.B., Koch, I., Aryal, D., Khadka, A., Shrestha, D., Abernathy, K., et al. 2020. Going to extremes: installing the world's highest weather stations on Mount Everest. *Bulletin of the American Meteorological Society*, **101**, E1870–E1890.

Potocki, M., Mayewski, P.A., Matthews, T., Perry, L.B., Schwikowski, M., Tait, A.M., Korotkikh, E., et al. 2022. Mt. Everest's highest glacier is a sentinel for accelerating ice loss. *npj Climate and Atmospheric Science*, **5**.

Reijmer, C.H., Bintanja, R. & Greuell, W. 2001. Surface albedo measurements over snow and blue ice in thematic mapper bands 2 and 4 in Dronning Maud Land, Antarctica. *Journal of Geophysical Research: Atmospheres*, **106**, 9661–9672.

Smedley, A.R.D., Evatt, G.W., Mallinson, A. & Harvey, E. 2020. Solar radiative transfer in Antarctic blue ice: spectral considerations, subsurface enhancement, inclusions, and meteorites. *The Cryosphere*, **14**, 789–809.

van Pelt, W.J.J., Oerlemans, J., Reijmer, C.H., Pohjola, V.A., Pettersson, R. & van Angelen, J.H. 2012. Simulating melt, runoff and refreezing on Nordenskiöldbreen, Svalbard, using a coupled snow and energy balance model. *The Cryosphere*, **6**, 641–659.

---

## Community Comment (CC1)

**Response to Brun et al. re Potocki et al. (2022)**

**\*** Mariusz Potocki[1], Paul Andrew Mayewski[1], Tom Matthews[2], L. Baker Perry[3], Margit Schwikowski[4], Alexander M. Tait[5], Elena Korotkikh[1], Heather Cliffiord[1], Shichang Kang[6,7], Tenzing Chogyal Sherpa[8], Praveen Kumar Singh[9], Inka Koch[10], and Sean Birkel[1] with additional input from Song Shu[3]

[1]Climate Change Institute, University of Maine, Orono, ME, USA.
[2]Department of Geography, King's College London, London, UK.
[3]Department of Geography and Planning, Appalachian State University, Boone, NC, USA.
[4]Laboratory of Environmental Chemistry, Paul Scherrer Institut, Villigen, Switzerland.
[5]National Geographic Society, 1145 17th St., Washington, D.C., USA.
[6]State Key Laboratory of Cryospheric Sciences, Northwest Institute of Eco-Environment and Resources, Chinese Academy of Sciences (CAS), Lanzhou, China.
[7]University of CAS, Beijing, China.
[8]International Centre for Integrated Mountain Development, Kathmandu, Nepal.
[9]Centre of Excellence in Disaster Mitigation and Management, Indian Institute of Technology Roorkee, Roorkee, Uttarakhand, India.
[10]Department of Geosciences, University of Tübingen, Tübingen, Germany.
**Correspondence:** mariusz.potocki@maine.edu; paul.mayewski@maine.edu; tom.matthews@kcl.ac.uk

In our paper (Potocki et al., 2022 – hereafter 'P22') there were two major findings: (1) that extremely rapid ice loss is possible once a protective snowpack is ablated away; and (2) this appears to have happened at the South Col Glacier (SCG) as evidenced by the presence of surface ice on the SCG dated at ~2000 years ago, indicating the loss of a significant portion of effectively what we consider to be at least currently a "stagnating glacier".

Brun et al. (hereafter B22) challenge both findings of P22. We welcome their paper and agree that more research, despite the extreme conditions involved in undertaking this research, is needed. Notably additional ice coring, mass balance stakes and an ice radar survey to more fully decipher ice dynamics and mass balance. However, we question the evidence used by B22 to underpin their conclusions. Our queries are outlined below – dealing first with finding (1) of P22 ("_could_ the SCG thin so rapidly") and then finding (2) ("_did_ the SCG recently thin"). Note that (1) is related to the surface mass/energy balance modelling of B22; (2) is concerned mostly with their DEM analysis. Accordingly, we organise our comment under the headings: "Energy Balance Modelling" and "DEM Analysis". Note that there is naturally some overlap between these sections.

**Energy Balance Modelling**

B22 conclude that substantial ice melt may not be physically plausible (even under extreme insolation). That is, they challenge the evidence that the SCG _could_ thin at the rate proposed by P22. Their reasoning is that P22's conclusion is not robust to _all_ modelling assumptions. Notably, if the conductive heat flux is calculated by the COSIPY model using a finer

resolution near the surface (the 'grad' experiment), or using a different model (CROCUS) substantial ice ablation cannot occur even if a snowpack is removed. This is a very interesting result, but is it physically plausible? Can such high conductive heat fluxes be maintained if the sub-surface was warming so much? Might increasing the resolution only near the surface (and not at all depths) create a 'cold' bias? Also, the temperature profile used to initialize COSIPY might be inappropriate for the B22 experiment: it was the outcome of a spin-up under the P22 setup -- i.e., with a much-reduced conductive heat flux. If spun up with the P22 grad method, the sub-surface temperature profile would very likely be different. In other words, there's a physical inconsistency between the spin up and the B22 experiments.

In the context of the above, we (P22) note here that both the COSIPY grad and CROCUS model variants are completely untested in an environment like the SCG. Indeed, CROCUS, as B22 explains, is primarily a snowpack model. By contrast, P22 used the 'default' COSIPY model code (i.e., with the conductive heat flux computed using a less-fine resolution near the surface). The setup used in P22 has been tested and shown to perform well against observations, including on Zhadang Glacier (at >5,600 m) on the Tibetan Plateau (a cold and dry environment not too dissimilar from the SCG). Note that the agreement between COSIPY grad and CROCUS and divergence from COSIPY P22 noted by P22 may just highlight a shared weakness of those model configurations; it is not reassurance of their physical realism.

It would also be very helpful if B22 explained which (if any) other COSIPY parameters were changed in their study, and if so, how results varied across those ensemble runs. That is, even if their model variants produce physically plausible results, B22 will only have demonstrated that the P22 conclusions are not robust to _all_ (reasonable) modelling assumptions, but no real insight into just *how* non-robust they are. This is particularly important because P22 performed an uncertainty assessment, perturbing model parameters across a broad range of plausible values. P22's conclusion about the plausibility of substantial ice melt was robust across all scenarios considered.

Taken together then, we caution that B22 give the impression that their conclusions about the plausibility of substantial ice loss should be given as much weighting as those of P22. However, without further work by the B22 authors, this is not the case: (i) the ability of their models to capture the surface energetics at the SCG remains to be determined; and (ii) their uncertainty assessment is too narrow to give an adequate perspective on the robustness of P22's conclusions.

**2. DEM Analysis**

B22 argue that, based on the difference in DEMs constructed for 1984 and 2017, the SCG has not thinned. We welcome this analysis, but have several observations that challenge their findings and overall interpretations:

(1) We note that B22's Figure 1 has our ice core at the wrong elevation. The ice core was not collected below 8000m but rather at 8020 m, which would place it within B22's so-called 'accumulation' zone. In addition, it is not clear why in their Figure A6, B22 define a particular line as the transition between accumulation and ablation based on a single day's image, implying this as the equilibrium line. Such a differentiation would be very sensitive to the timing of image acquisition and should instead be based on a much longer record of mass balance. As briefly acknowledged as a possibility (around L300 of B22), the steep snow slope designated as an accumulation area on Figure A6 is likely comprised of avalanche material (as evidenced below by the tongues of avalanched material) and is, therefore, not a standard snow accumulation region. The SCG surface downslope from the avalanche tongues is clearly exposed ice with patches of seasonal snow cover. Whether or not the SCG currently even has an accumulation area (at best very small) there is ice core and modeling evidence for current thinning/ablation up to at least the elevation (band) of 8020 m, which would be dominated by ablation indicating that the SCG currently has a negative mass balance. At lower elevation bands (than the ice core location) surface ablation/thinning rates of SCG is likely even stronger. The presence of clear banding and the identification of "annual layers" in the SCG ice core suggest that avalanching has not been the accumulation source for the SCG in the past, therefore something has happened, notably the transition to a stagnating glacier with its upper reaches comprised of avalanched snow.

[Figure]

(2) B22's Figure 1 shows that there are regions both above and below 8000 masl on the north side of the SCG that reveal thinning and thickening up to 30m for the period 1984-2017. This seems at odds with B22's statements that "… the distribution of dH on South Col Glacier is rather homogeneous and not different from the distribution of dH over ice-free areas or glacierized areas located within the same elevation range." Indeed, according to B22's Figure 1, dH over SCG is actually *at odds* with the *highly variable thickness change* over all other glacierized terrain at similar elevations. B22's Figure A1 obscures this by averaging over very large gains and losses to show a mean dH close to zero (minor comment -- what is the uncertainty shading supposed to represent in the right-hand panel of Fig. A1?). In addition, it seems unlikely that the south-facing SCG would experience little to no ablation while large parts of the north-facing Rongbuk Glacier would experience significant (up to 30m) ablation as indicated in Figure A1. Indeed, the upper branch of the Khumbu – only ~250 m from the SCG and with a similar (SW) aspect – did thin by tens of meters according to Figure 1.

We agree with B22's multiple mentions of the complications inherent with imagery interpretation for SCG and other high elevation regions and suggest that these might be hindering their Figure 1 results. We understand that The Pléiades DEM was generated from military-level satellite stereo images (0.5 m ground resolution). The only question is the accuracy of 1984 DEM which is not mentioned by B22. Our

understanding is that the DEM was generated from images collected by an airplane at 10,000 feet (3048 m) above the top of Mount Everest according to the article from which the 1984 images were obtained. An aircraft under these conditions inevitably has vibrations induced by airflows. Even though the Wild RC-10 camera was aimed straight down, the vibration of the airplane would always introduce a slight angle to the camera's nadir looking direction. We do not know how accurately the nadir looking direction was maintained since no information about the image acquisition was provided. Assuming only a 0.5 degree departure from the nadir direction, then the error of location on ground is over 26 m (tan(0.5) × 3048 = 26.6 m). For Mount Everest, the 26.6 m of displacement could result in a several meter-level error in elevation due to the drastic change of topography within a short distance.

(3) We find the information provided in B22's Figure A6 a bit perplexing. In particular, as stated by B22. … "the shaded hashed area represents glacierized area that might belong to SCG, but it is not possible to conclude solely from satellite imagery." Might this not be part of the areal loss assumed in P22 over the last three decades? Clearly the image recognition for this region is still in question.

(4) B22 present an interesting display of seasonal snow variability (their Figure 2). This is used to argue for it being unlikely that the SCG is snow-free during the monsoon, although they do not in fact include any images from the critical months of May and June, before arrival of the monsoon. In turn, they reason that this helps explain their conclusion that the SCG has not recently thinned, because P22 required snow-free conditions during the monsoon to drive the widespread melting needed to ablate the SCG. We certainly agree with B22, that the SCG is covered by snow during the latter portions of the monsoon in August and September, but question their assertion that snow cover is present in May through early to mid-July when insolation is at its annual maxim. Albedo data from the South Col AWS confirm a largely snow-free surface from November 2019 through mid-July 2020 (Bessin et al. 2021). In addition, the three years considered by B22 may not be representative of longer term conditions. A similar view of seasonal variability during several earlier image periods would have been interesting to include. To this end, we note the availability of twice-daily images from Mt Everest's Basecamp in Nepal (see below) should help shed ever more light on this issue, given that changes in snow-cover above 8000 masl are clearly visible. Details of these photographs are in Grey et al. (2022).

[Figure]

A related issue with B22's assessment of seasonal snow cover is their deployment of an empirical wind redistribution model. Several limitations of it are mentioned, but the most significant – and arguably so great that it should rule out its inclusion -- is that the SCG environment is so different from the (ice sheet) environment it has been tried and tested in. First, the SCG is in a very complex topographic setting and likely subject to very high small-scale wind variability due to shear-induced turbulence. This matters acutely when considering wind redistribution because maximum gusts set the upper limit on erosion potential. Second, the atmospheric pressure at the SCG is approximately one-third that of sea level. Even if air density is a parameter within their model, what evidence do they have that an empirical scheme developed and applied at much lower elevations/higher pressure behaves realistically in such a different environment? Taking the B22's own words (~L250 in the context of mass balance modelling) "*[models]… developed and tested in specific conditions, …[should not] be applied directly to other conditions, such as the very specific conditions of South Col glacier, without extensive validation.*"

The point about the monsoon possibly being a time when the SCG is snow-covered seems to be made most strongly with the Venus images and basic physical reasoning about the wind speeds and precipitation occurrence. The empirical model is so uncertain that we argue it detracts rather than adds to this argument. Perhaps instead B22 would consider replacing this with modelling of the SCG surface mass during the monsoon. Forcing their COSIPY and CROCUS model variants with the P22 precipitation (which they inhibit in their ice model runs) should give more useful insight into the extent of snow cover during this critical time of the year. All lines of

inquiry (Venus images, physical reasoning, and their empirical model) already identify the monsoon as a period of minimal wind deflation, so a more important question is to what extent monsoonal snowfall is melted or sublimated away – not least because all models (and both studies) agree on the very high importance of the latter.

In the context of the above, B22 assert (around L165) that P22's estimates of precipitation are highly uncertain because they tune them to match ablation over an arbitrary period. We agree that they are uncertain, but note that the long-period of integration (10 years) protects somewhat against sampling variability. We also note that P22's precipitation estimate (mean of 191 mm a$^{-1}$) was an order of magnitude greater than suggested by previous work (Salerno et al., 2015). There are reasonable explanations for that (including that the latter used poorly-shielded instrumentation to measure precipitation, hence a high risk of under-catch); but if B22 include our suggestion to model the SCG mass balance during the monsoon, we suggest that they keep this in mind: P22's precipitation estimates are unlikely to be biased low.

(5) We believe that the P22 core was drilled in a stagnating glacier that has a seasonally reconstituted accumulation zone comprised of continually avalanching snow and ice. In this context, we note that the arguments invoked by B22 about implied low ice velocity being evidence of no/limited ice melt (due to low mass turnover) are not relevant; they are just another way of describing their (alternative) hypothesis – that P22 drilled their ice core from the ablation area of glacier in balance. Under the P22 stagnation (and thinning) hypothesis, there is clearly no requirement for the ice flux to balance the implied ablation!

We also emphasize that our estimated ice loss was not just a guess, it was based on the identification of annual layers in the 10m ice core (verified by seasonality similar to our other Himalayan ice cores), radiocarbon dating of the near top and the bottom of this core, and depth/age data developed from the Rongbuk Glacier ice core which we (along with our Chinese colleagues) recovered ~5km north of South Col Glacier at 6518 masl in 2002 (Kaspari et al., 2009).

(6) B22 assert that there is no evidence for any substantial ice melt having occurred due to an absence of fluvial features on the ice or off-glacier. The authors of P22 discussed this via email with the B22 authors, but some relevant parts of that discussion were omitted in B22: (1) the gently sloping SCG surface (below the avalanched region) would promote evaporation of meltwater, not least because of the extreme vapour pressure gradient to be expected with a surface so much warmer than the atmosphere above. The SCG is also very small, so we should not expect "large supraglacial stream features" (visible in satellite imagery) to form. For example, P22 proposed a potential (ice) melt rate of ~16 mm/d (assuming 1.5 m w.e.

lost during a 90-day monsoon). If this depth (0.016 m) is multiplied by the 200,000 $m^2$ area of the SCG it means a volume of 3,200 $m^3$ evacuated per day, so a mean runoff of 0.04 $m^3$ $s^{-1}$. Given that much of this would be lost to evaporation, and possibly split between multiple streams, the potential for large supraglacial meltwater features seems limited. We also do not understand which "photographs of the glacier surrounds" B22 refer to when citing no "evidence of runoff, such as stones being embedded into re-frozen water" Such features would not be visible in satellite imagery; and re-frozen meltwater would in any case quickly sublimate at the SCG. We also explained that mountaineers described to the P22 team (prior to their 2019 expedition) that they could expect to observe meltwater at the SCG during the *pre-monsoon* (i.e., not even the period of maximum temperatures and insolation).

We also highlight that P22 suggest melting of an ice surface – if exposed – would occur during the monsoon, when equivalent temperature (proportional to the sum of the atmospheric sensible and latent heat content) is at a maximum, insolation is closest to its peak values, and when light winds would limit the potential for cooling of the surface via the turbulent heat fluxes. Unfortunately, very few people have ever seen the SCG during the monsoon – and that extends to the imagery shared by B22! However, we *can* make a space-for-time substitution. The authors of P22 have spent a combined total of almost one month at Camp II (6,464 masl) during the pre-monsoon (late April to late May) in 2019 and 2022. On both occasions, meltwater was abundant, with a significant supraglacial stream present on the northern margin of the Khumbu Glacier throughout (unfortunately we did not take pictures, but we estimate it several metres in width and tens of centimetres in depth). Following from the above, this is consistent with the much larger catchment area of the upper Khumbu compared to the SCG. In early May 2022 the team also observed a saturated snowpack at the base of the Lhotse Face (>100 m above Camp II; see the foreground in the image below taken during the 2022 expedition). This melting is occurring on a surface with a higher albedo than would be expected if ice were exposed at the SCG.

[Figure]

During the team's last days at Camp II in early May 2022, meltwater was particularly widespread (beyond the normal confines of the aforementioned stream), with it even being necessary to excavate channels to prevent the tent from being flooded (see image below). Just six days earlier Camp II was covered by ~10 cm of snow, and the maximum air temperature throughout this period of high melt remained well below freezing at -5C.

[Figure]

Critically, the potential melt energy is very similar between Camp II during late-April to late May, and the SCG during the monsoon (see figure below). Indeed, the incident (short- and long-wave) radiation (which, P22 suggest, drives the melting at the SCG) is almost identical between sites (right-hand histogram). Note that the difference in equivalent temperatures (left-hand histogram) becomes ever-less important as wind speeds drop. This point relates very clearly to those raised in the energy-balance section above. That is, this space-for-time substitution suggests that, if abundant melt is evident *above* Camp II *during the pre-monsoon*, it would be reasonable to expect a similar response at the SCG during the *monsoon,* given that the former seems to be a very appropriate (perhaps even conservative) analogue for the latter (given the lower albedo of the ice at the SCG).

[Figure]

[Figure]

Above:  Comparison of equivalent temperature (left) and incident radiation (incident shortwave radiation plus incident longwave radiation; right) at Camp II (red) during the pre-monsoon (last week in April to the last week in May, 2019 and 2022) and the SCG (blue) during the monsoon (July and August, 1991-2020). Note that the Camp II data were taken from the AWS at 6464 masl, and the SCG data were taken from the P22 ERA5 downscaled data (to the SCG AWS at 7,945 masl).

Taken together, point (6) indicates that B22 do not provide convincing evidence that substantial ice melt has not occurred at the SCG.

In closing, we highlight that P22 and now B22 have taken very different approaches to the study of the iconic SCG. They also reach different conclusions over whether (1) the SCG *could* thin rapidly (if ice were exposed), and (2) whether it *has* thinned rapidly.

We argue here that B22's findings – which challenged those of P22 on both counts -- are more uncertain than presented by the manuscript in its present form and should be re-examined before their paper is published.

**References**

Bessin, Z.; Dedieu, J.-P.; Arnaud, Y.; Wagnon, P.; Brun, F.; Esteves, M.; Perry, B.; Matthews, T. Processing of VENµS Images of High Mountains: A Case Study for Cryospheric and Hydro-Climatic Applications in the Everest Region (Nepal). *Remote Sensing.* 2022, *14*, 1098. https://doi.org/10.3390/rs14051098

Brun, F., King, O., Réveillet, M., Amory, C., Planchot, A., Berthier, E., Dehecq, A., Bolch, T., Fourteau, K., Brondex, J., Dumont, M., Mayer, C., and Wagnon, P.: Brief communication: Everest South Col Glacier did not thin during the last three decades, *The Cryosphere Discussions*. [preprint], https://doi.org/10.5194/tc-2022-166,  in review, 2022.

Grey, L., Johnson, A.V., Matthews, T., Perry, L., Elmore, A.C., Khadka, A., Shrestha, D., Tuladhar, S., Baidya, S.K., Aryal, D. and Gajurel, A.P. (2022), Mount Everest's photogenic weather during the post-monsoon. *Weather*, 77: 156-160. https://doi.org/10.1002/wea.4184

Kaspari, S., P. A. Mayewski, M. Handley, E. Osterberg, S. Kang, S. Sneed, S. Hou, and D. Qin, 2009, Recent increases in atmospheric concentrations of Bi, U, Cs, S and Ca from a 350-year Mount Everest ice core record, *Journal of Geophysical Research*, 114, D04302, doi:10.1029/2008JD011088.

Potocki, M., Mayewski, P.A., Matthews, T. *et al.* Mt. Everest's highest glacier is a sentinel for accelerating ice loss. *npj Climate Atmospheric Sciences* 5, 7 (2022). https://doi.org/10.1038/s41612-022-00230-0

Salerno, F., Guyennon, N., Thakuri, S., Viviano, G., Romano, E., Vuillermoz, E., Cristofanelli, P., Stocchi, P., Agrillo, G., Ma, Y., and Tartari, G.: Weak precipitation, warm winters and springs impact glaciers of south slopes of Mt. Everest (central Himalaya) in the last 2 decades (1994–2013), *The Cryosphere*, 9, 1229–1247, https://doi.org/10.5194/tc-9-1229-2015, 2015.

---

## Community Comment (CC2)

**Reply to reviewer Ann Rowan**

I am very grateful for the substantial efforts of reviewer Ann Rowan in attempting to reconcile disagreements across three sources (Potocki et al., 2022; Brun et al., 2022 – under review here; and Mayewski et el. 2022 – a comment posted in this forum on Brun et al. 2022). This is clearly a challenging task. In this reply to Ann Rowan (AR), I wish to make a few clarifications to aid with the rest of the review process.

**First**, AR states:

*"L28: worth noting here that the South Col AWS recorded only about five months of data (May-end summer 2019)…"*

However, the data (freely available here) extend from May 2019 *to August 2022*. The record (although interrupted in places) is therefore over three years in length.

**Second**, AR comments (in relation to the 'space for time substitution' in Mayewski et al. 2022) that it is 'not convincing' because:

*"the incident radiation would presumably be much lower than given here due to monsoon cloud cover"*

However, the incident radiation 'given here' (i.e., shown in the histogram of Mayewski et al. 2022) takes account of cloud cover. It uses the long-term (1991-2020) downscaled ERA5 insolation (which Potocki et al., 2022 showed was an excellent match to the observations). It therefore accounts for variable atmospheric transmissivity – including the effects of clouds. Note, too, that the extremely high insolation (close to top-of-atmosphere values) at the South Col AWS can be seen in Figure 4 of Matthews et al. (2020).

We make this clarification because it appears that AR was not aware of the provenance of the data used in the space-for-time comparison.

A more subjective point I note here is that AR stated that a large temperature difference between Camp II and the South Col means that the melting is unlikely to occur at the latter, even if it does at the former. However, as pointed out in Mayewski et al. 2022, this temperature difference counts very little if winds are very light (which they are in the monsoon: Matthews et al. 2020, Fig. 4). Indeed, in the extreme event that the air is still, the turbulent heat flux is zero, so the lower air temperatures do not act to cool the surface (note that the right-hand side of the histogram shown in Mayewski et al. shows incident radiation, so already includes the effect of lower air temperatures on longwave radiation).

***Third,*** AR states that:

*"Determining mass change over a representative timescale of several decades requires observations of longer-term change as provided by both papers."*

This seems to be a misunderstanding of the methods used by Potocki et al. (2022): their conclusions were reached with the help of a *70-year* record of (downscaled ERA-5) meteorological data, not just data from the 2019 season (which is the focus of Brun et al. 2022).

**Fourth**, AR mentions her own group's modelling work that shows annual accumulation of 7 m w.e. at the South Col.  According to my understanding, this result represents precipitation minus sublimation and any melt, and hence appears physically implausible: 7 m w.e. is ~13 times the annual mean precipitation (AMP) measured at Base Camp (5,315 m asl) and ~9 times the AMP at Phortse (3,810 m asl). Considering that AR's figure is net of any sublimation (if not also melt), and both that theory and our measurements highlight a decline in precipitation with altitude, I suggest that this (non-peer-reviewed) result is physically implausible and should be discounted from further discussion.

---

## Community Comment (CC4)

As a contribution to this discussion, I would like to submit some observations from remote sensing that may aid in interpretation of annual surface melting from modeling in Brun et al. (2002). In Scher et al. (2021) created a record of surface melting over glaciers in the Himalayas from a time series Sentinel-1 synthetic aperture radar (S1-SAR). Melt is detected where annual backscatter is reduced as liquid surface water obscures the radar scattering from the glacier interior, resulting in a marked reduction in backscatter. For the South Col glacier, we observe radar signatures that indicate surface melting is occurring in 2019 over areas of exposed ice in the southern extent of the glacier (Figure 1, attached). From time series S1-SAR, we observe continuous indications of surface melting from June 26, 2019, until October 6, 2019, with an approximately biweekly repeat observations during this period. Since seasonal snow over areas that are exposed on an interannual basis are not deep enough to contribute substantially to radar scattering, we infer that the melting signal originates from structural features (e.g., laying) in the glacier interior that result in enhanced backscatter during colder winter months. It is important to note that at C band frequencies backscatter is extremely sensitive to liquid water and it is difficult to differentiate very small amounts of surface melting from more extensive melting, and therefore our methodologies are not well suited to evaluate the amount of melting that may be occurring. For more details on our methodologies, please refer to Scher et al., (2021).

[Figure]

Figure 1 (a) A true-color map from Sentinel-2 during April 2021 of the South Col Glacier indicates areas of blue ice. Sentinel-1 time series synthetic aperture radar (SAR) from locations in (a) the accumulation zone, as designated by Brun et al., (2022) in Figure A6 indicate melting where observed backscatter is found to be ~3 db below (red lines) the winter mean (black-lines). For both the (c) exposed ice and the location of the (d) ice core site from Potocki et al. (2022) we find similar radar signatures that indicate surface melting.

---

## Community Comment (CC5)

This is a very valuable review. I commend the authors for attending to the debate in so much detail.

However, I wish to make a brief correction. It is stated that Potocki et al. (2022) did not assess the sensitivity of their SEB results to albedo. However, they did indeed perform this sensitivity assessment as part of a 'bounded' uncertainty quantification (Smith et al., 2018), whereby parameter values were perturbed to their plausible min/max values in order to estimate equivalent, plausible min/max rates of ablation. As part of that assessment (detailed in the SI of Potocki et al., 2022), the albedo was varied between 0.3 and 0.5. The higher albedo reduced the melt rate, but did not materially affect their SEB conclusions (i.e., the physical plausibility of substantial ice melt).

I also note that whilst the ice at the drill site was cleaner that the ice circled in the image below (shown in Matthews et al., 2022), the blue ice areas of Antarctica are clearly not a good surrogate for albedo at the South Col Glacier as a whole, so would encourage any such analogy to be made with caution. The ice circled is indeed likely to have been darkened through ablation (and may well have an albedo appreciably below 0.4). The assignment of the 0.4 value by Matthews at al. (2020) and Potocki et al. (2022) was an estimate, but an informed one based on visual site inspections/comparisons by the authors.

[Figure]

I also suggest cautious interpretation of the reviewers' 'spin-up' experiments. Presumably the reduced melt is because a perennial snowpack develops? This is questionable, because the precipitation data used by the authors of this review are tied to the surface energy

balance results of Potocki et al (precipitation was corrected by the authors to match ablation in the first decade of the simulation). With another model formulation that simulates less ablation, the precipitation would be reduced too (so a perennial snowpack would likely not develop in the spin-up period).

Importantly, this review, and the work of Brun et al., build on Potocki et al. to highlight the sensitivity of the South Col Glacier to SEB modelling uncertainty. This is very interesting in its own right. If Potocki et al.'s model formulation is close to being 'right', it suggests an extreme sensitivity to the maintenance of a protective snowpack at the South Col. This can remain true, even if Brun et al. are right in that the South Col Glacier did not actually thin as Potocki et al. suggested (i.e., because the snowpack *has* mostly been preserved).

If the conservative model configurations results of this review and Brun et al. are correct, however, then rapid thinning is unlikely *even if* ice becomes exposed for long periods of the year.

The resolution of this debate matters for our understanding of what may happen in the future at the South Col Glacier, and possibly for other ice masses at extreme elevation. Accordingly, I suggest that the discrepancy between results should be a call for urgent further research.

There are bright possibilities for that in the near future. First, the albedo at the South Col Glacier can be constrained using satellite measurements (Bessin et al., 2022). Second, high-resolution radar measurements can be used to identify melt events (Scher et al., 2021). Given that melt events **are** regularly detected at the South Col (Steiner, this discussion), we must reject the parameter values and model structures which generate *no* melt. Indeed, given the comment by Steiner in this forum showing melt events *throughout June-October*, I would also suggest that setups generating *minimal* melt rates (Brun et al., and Machguth and Mattea) – even when ice is exposed – are unlikely to be appropriate. Although Steiner's results cannot quantify the magnitude of melting, they show that it is *not* a rare occurrence. Indeed, if some melting is detected in June and October, melt rates in the peak Monsoon months of July and August – when moist enthalpy is at a maximum and wind speeds are at a minimum (Khadka et al. 2021) – would be considerably higher for an exposed ice surface. Quantifying *how much* higher – with perturbed parameter ensemble model runs constrained by satellite observations and the weather station data -- should be a high priority for future research.

**New references cited**

Bessin, Z., Dedieu, J.P., Arnaud, Y., Wagnon, P., Brun, F., Esteves, M., Perry, B. and Matthews, T., 2022. Processing of VENμS Images of High Mountains: A Case Study for Cryospheric and Hydro-Climatic Applications in the Everest Region (Nepal). *Remote Sensing*, *14*(5), p.1098.

Khadka, A., Matthews, T., Perry, L.B., Koch, I., Wagnon, P., Shrestha, D., Sherpa, T.C., Aryal, D., Tait, A., Sherpa, T.G. and Tuladhar, S., 2021. Weather on Mount Everest during the 2019 summer monsoon. *Weather*, *76*(6), pp.205-207.

Scher, C., Steiner, N.C. and McDonald, K.C., 2021. Mapping seasonal glacier melt across the Hindu Kush Himalaya with time series synthetic aperture radar (SAR). *The Cryosphere*, *15*(9), pp.4465-4482.

---

## Author Response (AR1)

**Reply to referee comments for paper TC-2022-166**

We would like to thank the editor and the reviewers for their detailed reviews of this paper, which helped a lot to improve it. We addressed all the reviewers' comments and please find below our point-by-point reply. We also want to acknowledge anyone who contributed to the open discussion. The discussion was rich and clearly contributed to increasing our knowledge about this South Col glacier and more broadly about all glaciers located at extremely high elevation. We also provide a point-by-point response to all comments, after the replies to reviewers.

**Summary of major changes**

- **DEM difference**: we provide more details about the processing of the aerial photographs (section 2), and hence about the quality of the elevation change map and robustness of our estimate. In the supplementary, we also added an alternative calculation of the elevation change uncertainty based on the Hugonnet et al. (2022) method, which led to smaller uncertainties. We also describe in more detail some patterns evident in the elevation change map which we interpret to show crevasse advection and serac falls.
- **Surface mass balance modeling**: we simplified section 3.3 to be more explicit about our main objective, i.e. showing that surface energy balance models can produce very different results in the same situation. We also removed the simulation using the Crocus model, as we feel that it diluted the message.
- **Wind erosion**: we changed the title of section 3.2, which is now "The potential of wind erosion", but kept the modeling section as wind erosion is a major process that needs to be highlighted. This process might explain why parts of the South Col Glacier are in ablation, despite a regional ELA being 2000 m below the glacier. We changed the terminology according to reviewer 3's suggestion.
- **Delineation of an accumulation and ablation area**: we removed fig. A6 as our delineations of an accumulation and ablation area were not solidly grounded enough. We largely rewrote section 4, and removed the considerations about ice fluxes that were rather speculative given the absence of data about the glacier thickness and velocity.
- We added two co-authors to the paper: Silvan Leinss and Romain Hugonnet who contributed to the Sentinel-1 analysis and the uncertainty calculation, respectively.

**Reply to referees comments**

The following pages contain a point-by-point reply to the comments provided by the four referees that reviewed our first submission (TC-2022-166).
Each of the referee's comment (RC) is numbered. If a comment contained several points, we numbered them, and addressed them individually in our author replies (ARC).

**Referee comment 1 - Ann Rowan**

[RC1-1] The brief communication by Brun et al. (2022) "Everest South Col Glacier did not thin during the last three decades" presents measurements of the mass change of Everest South Col Glacier between 1984 and 2017 from differencing of digital elevation models, and compares these to results from a set of sensitivity experiments using the COSIPY and CROCUS models. The manuscript is written in response to a paper by Potocki et al. (2022) which calculated mass loss from this glacier of 1.5 m w.e. a–1 from analysis of an ice core and the COSIPY model. The Brun et al. (2022) study finds that mass change for this glacier is within uncertainty of zero. Everest South Col Glacier is a small (0.2 km2) ice mass at 8,020 m a.s.l. on the southern side of Sagarmatha, located above the headwall of Khumbu Glacier and on the climbing route for this mountain from the south. While it is rare that so much effort is dedicated to determining the mass change of such a minor glacier, the location at extremely high elevation is used to justify the attempt with the implication that if glaciers are losing mass at present at the highest elevations in the Himalaya, then widespread mass loss is expected at all elevations.

In the first part of my review, I make several major comments and some minor (editorial) comments on the work by Brun et al. (2022) which I request that the authors address in revising their manuscript. In the second part of the review and because this work is a response to a previous study, I compared and evaluated the results of both papers. The authors of Potocki et al. (2022) wrote a response to Brun et al. (2022) in this discussion, led by the second author rather than the more junior first author, which I discuss in the second part of my review and refer to as Mayewski et al. TCD.

[ARC1-1] We would like to thank Ann Rowan for her thorough review, and the evaluation of both papers, Potocki et al. (2022) and Brun et al. (2022)

Review of Brun et al., 2022: Major comments
**[RC1-2]**
1. Glacier mass change data. Brun et al. first present an analysis of DEM data to demonstrate that no mass loss has occurred from Everest South Col Glacier between 1984 and 2017 CE. This method is well established and has been thoroughly tested for glaciers in this location, notably in the recent paper by King et al. (2020) of which one of the co-authors of Petocki et al. (2022), Alexander Tait, is also an author and therefore aware of the method. It would have been justified to end the paper after these analyses, as the results are convincing and of greater value than the modelling for the reasons outlined below.
These results could be described in more detail to make this section more accessible to a wider audience. Mayewski et al. TCD have interpreted areas of negative surface elevation change in the upper accumulation area of Khumbu Glacier as mass loss indicative of glacier wide mass balance rather than redistribution of mass within the glacier.

**[ARC1-2]** We appreciate the reviewer highlighting the previous successful use of optical stereo imagery to examine glacier change in the region. In this study, we examine imagery of higher resolution (0.5 m) than in many previous long-term studies in the region, to ensure we are able to generate accurate DEMs over glacier surfaces which may prove problematic to lower resolution sensors (e.g. ASTER). By doing so, we have built on previous work and have been able to examine glacier change in a more precise manner.

We agree with the reviewer that some areas of the dH data shown in Figure 1 have been misinterpreted as "ablation" (comment CC1-4) when they in fact show the advection of surface features of substantial relief (seracs and crevasse blocks) down-glacier by ice flow. Such features and associated patterns of elevation change are widespread on the Kangshung face directly north and east of the South Col Glacier and southwest of the South Col Glacier towards the base of the Lhotse face (Response Fig. 1).

[Figure]

**Resp. Fig. 1:** Examples of glacier surface conditions captured by aerial photographs (1984) and Pléiades imagery (2017) in the Western Cwm and associated changes in surface elevation. Top row: Crevassing of the Lhotse face in 1984 (a) and 2017 (b) and corresponding elevation change estimates (c) over the same period. The alternating positive and negative elevation difference pattern reflects the movement, opening and/or closure of crevasses. Bottom row: the expansion of the area of exposed bedrock around Camp II between 1984 (d) and 2017 (e) and associated elevation changes (f). Shadows prominent in panels a & d illustrate the winter time acquisition date of the aerial photographs, compared to the Spring (23rd March) acquisition of the Pléiades imagery (panels b & e).

We now detail these elevation change patterns at the end of section 2: "Several patterns of dH are evident over the Western Cwm and South Col Glacier surroundings which relate to both ice flow and surface mass balance processes (Fig. A1). Thinning and recession of the steep hanging glaciers on the north face of the Western Cwm is evident north of Camp II at

an elevation of 6500 m a.s.l. (Fig. A1 panels d to f). Slight (10 m or less) thinning is evident over the Khumbu Glacier up to an elevation of 7000 m a.s.l. Above this height, substantial elevation change is limited to areas where ice flow has driven crevasse field evolution between the two DEM dates, primarily on the Lhotse and Kangshung faces, to the east and southwest of South Col Glacier (Fig. A1 panels a to c). Over the South Col Glacier specifically, we find a mean elevation change of 0.01 ± 0.07 m a−1 for the period 1984-2017. The distribution of dH on South Col Glacier is rather homogeneous and not different from the distribution of dH over ice-free areas or over glacierized areas located within the same elevation range (Fig. A2)."

To better illustrate the characteristics of the elevation change (dH) data over the areas highlighted in **CC3-1**, we examined dH estimates over stable ground and the steepest sloping surfaces in the study area (>40˚), where lower resolution DEMs can indeed sometimes struggle to accurately reconstruct topography, and where any positional mismatches between DEMs would be most apparent (**Resp. Fig. 2**, panel A). Over surfaces which should be expected to have been stationary between the two DEM dates (off-glacier) *and* of a surface slope greater than 40˚, the mean elevation difference is -0.04 m (n = 352553, standard deviation 4.5 m). For surfaces of a slope greater than 50˚, the mean dH is 0.21 m (n = 96285, standard deviation 6.55 m), greater than 60˚ it is 0.55 m (n = 32082, standard deviation 9.34 m), and for slopes greater than 70˚ the mean is 0.31 m (n = 7381, standard deviation 13.2 m). Close agreement between the two DEMs over stationary surfaces, particularly over steep slopes, confirms the absence of positional errors between the two datasets.

[Figure]

**Resp. Fig. 2:** A) Distribution of elevation change values off-glacier as a function of the terrain slope, for slopes higher than 40°. B) Correlation of elevation change data over different spatial scales. C) Relationship between NMAD of dH and surface slope, derived from the Pléiades DEM. D) Relationship between NMAD of dH and terrain curvature, again derived from the Pléiades DEM.

Furthermore, considering the above and the comments of RC1-2 and CC3-1, we have undertaken additional work to more thoroughly examine the uncertainty associated with our elevation change data, specifically considering the relationship between error and terrain characteristics such as slope and curvature. To do so, we followed the methodology developed by Hugonnet et al. (2021) (and more thoroughly described in Hugonnet et al. (2022)), implemented in the xDEM Python package (https://xdem.readthedocs.io/en/latest/). TWe estimated and modeled the structure of error of our elevation change data by examining the spatial correlation of errors (scale of spatially consistent noise) and the heteroscedasticity of elevation changes (per-pixel variability in error), using stable terrain (off-glacier surfaces) as an error proxy. Then, we applied this model of error to spatially propagate elevation change errors to the mean elevation change in the area of South Col glacier.

The heteroscedasticity was estimated using both the NMAD and half of the 2.5-97.5 percentiles of elevation differences, keeping the latest one as most conservative (**Resp. Fig. 2** panels C and D). Indeed, in addition to the typical error variability found with slope and curvature (gradual increase), we also observe an unusual increase in error at moderate slopes (15-35°), better captured by the percentiles. We interpret this increase in error as being related to the moderate-slope location of the sampled pixels which, for the most part, is snow covered and thus has low contrast in the stereo-images leading to lesser precision in the DEMs. While, ideally, this variability could be modeled using a DEM quality variable (e.g., quality of stereo-correlation), none was generated in the present study. Thankfully, the dependency of snow cover to slope (which is approximately the same on stable and glacierized terrain) allows us to account for it through slope. To ensure the reliability of our approach, we verified on a Q-Q plot that our error model explains almost all of the departure from normality of the elevation change distribution, meaning that almost all of the variability in elevation change error is captured by our method.

The spatial variogram was estimated using a robust median estimator and modeled with a multi-range spherical function (**Resp. Fig. 2** panel B). We found that elevation change errors are correlated at 89% (in variance) until a range of 87 m, then correlated at 11% until 6.5 km, after which they are completely decorrelated. These findings match the scales over which stable ground elevation change data is visually correlated within the dH grid of Figure 1 in the main manuscript.

Eventually, we calculated an average per-pixel error of SCG (accounting for slope and curvature variability) of 5.10 m, slightly larger than the 3.88 m average dispersion on stable terrain, because SCG is located on bright, moderate slopes that have larger errors. This leads, after spatial integration, to an uncertainty in the mean elevation change of 1.77 m (at 1-sigma level) for South Col Glacier (see notebook with more details here: https://github.com/rhugonnet/uncertainty_analysis_SCG/blob/main/uncert_SCG_from_Hugonnet2022.ipynb

We note that this improved uncertainty estimate is slightly smaller than the one shown in B22 (2.2 m), which was thus conservative.

We also note that our photogrammetry-based elevation change assessment is now backed up by the analysis of in situ photographs taken by Horst Machguth and Enrico Mattea. (see RC4-2)

**[RC1-3]** 2. Surface energy-mass balance modelling. Calculating the surface energy balance of a glacier in this setting is extremely challenging, as extreme winds strongly affect the accumulation and removal of snow from the glacier surface, melt processes are dominated by sublimation, and the influence of the Indian Summer Monsoon on glacier mass balance is unknown at these elevations. I consider this glacier an unsuitable candidate for any surface energy-mass balance modelling study unless a model was developed specifically for this location and constrained by detailed and representative atmospheric and glaciological data (i.e., collected at the site of this glacier over several years, rather than using empirically derived values from other settings).

Brun et al. (2022) have taken a pragmatic approach by reproducing the COSIPY model parameterisation used by Potocki et al. (2022) in a sensitivity test that considers a graduated mass balance parameterisation of the same model and a comparison with results from a snow model, CROCUS. Their results demonstrate that the simulated glacier mass change is sensitive to the model time step used, and that there are large uncertainties associated with such calculations. The model results are useful as a comparison with the approach of Potocki et al. (2022), but I suggest that the modelling work from both papers is phrased more cautiously; as potentially useful to identify where the largest uncertainties arise in estimating the mass balance of South Col Glacier, but unlikely to accurately represent glacier change.
Brun et al. (2022) also used a snow deposition model to quantify the impacts of wind on snow accumulation at this glacier and determine when the ice surface is free of snow and hence may melt. These results are compared with satellite imagery and show good agreement. This model application is more valuable than COSIPY for investigating South Col Glacier, but still contains large uncertainties. As discussed below, these results illustrate the limitations of the interpretation of the ice core data by Potocki et al. (2022).

**[ARC1-3]** As noticed by Ann Rowan, modelling has been used in a pragmatic approach to 1. show that the simulated glacier mass change is sensitive to the model numerical implementation (use of Cosipy-grad and Crocus) and 2. highlight the importance of snow drift (snow deposition model). The objectives of both modelling approaches is neither to produce reliable estimates of glacier mass changes, nor to provide a perfect quantification of snow drift. Indeed, we fully agree that this glacier is an unsuitable candidate for any surface energy-mass balance modelling study unless a model was developed specifically for this location and constrained by detailed and representative atmospheric and glaciological data. This was clearly stated in our original manuscript in the last paragraph (lines 313-320 of the original manuscript):

"The surface mass balance processes happening in the extreme meteorological context of South Col Glacier are complex, and our study does not reach any definitive conclusion about the relative importance of each of these processes. The lack of direct observations hampers our ability to decipher the dominant glaciological processes, and thus to model the glacier recent and future evolution in a realistic way. Specifically, stake measurements would be needed to measure the surface mass balance and surface velocity in a direct way, ground

penetrating radar measurements would help constrain the ice thickness, and a number of subsurface temperature, snow-depth, snow transport or turbulent fluxes measurements would help constrain the processes. Without more data constrained knowledge, it appears currently impossible to conclude about the sensitivity of South Col Glacier to climate change, nor to predict its future evolution."

The aim of this modelling approach was more to highlight some specific processes not included in Potocki et al (2022), such as wind erosion, and to warn about the large sensitivity of the model results (melting here) to the numerical implementation. This rationale behind the use of these models was explained (lines 239-241, of the original manuscript):

"This numerical experiment demonstrates that the structure and physical implementations of a model can strongly affect the way the energy is spatially allocated and transported, leading to large variations in predicted melt despite solving, in principle, the same physical processes"

Yet, we acknowledge that the inclusion of Crocus might have blurred somewhat this message. Therefore, we decided to remove all sections and simulations with the Crocus model. We now focus our demonstration on showing that results obtained with Cosipy cannot be taken for granted as they prove to be fundamentally different when another equally-acceptable numerical implementation of the subsurface heat flux is tested. To clarify our objective, we added a new sentence at the beginning of section 3.3: "The challenge of modelling the surface mass balance":

"The purpose of the numerical simulations that we perform in this section is not to produce realistic estimates of the surface mass balance prevailing at SCG, but to show that various acceptable choices in the numerical treatment of the surface energy balance in Cosipy produce very different results in terms of melt. Thus, firm conclusions regarding melting at SCG should not be drawn based on such weakly determined Cosipy simulations."

**[RC1-4]**
3. Description of glacier geometry. I find it strange that Brun et al. (2022) assign an ablation area to South Col Glacier in Figure A6 and would expect emergence of ice to be minimal for this glacier as they predict. As discussed below, this interpretation seems rather strange based on the glacier's elevation relative to the local ELA. I suggest revising this figure and reframing the interpretation of the glacier as an accumulation area only.

[ARC1-4] We agree that assigning a geometry of an ablation area as done in Fig A6 is very speculative, and is based on many assumptions that are not possible to validate. In the revised manuscript, we removed this section concerning considerations of ice flow, emergence velocities and ablation/accumulation areas. We still believe that this glacier is very specific and cannot be considered as an accumulation area only, although its elevation is far above the local ELA. One compelling argument is the presence of exposed ice, which shows that ablation processes dominate, at least on some locations of the glacier. Indeed, wind plays an extremely important role for ablation processes (i.e., sublimation, snow drift and erosion). As a consequence, an ablation area is likely to exist, potentially in the lower part of this glacier, where wind is probably stronger (venturi effect due to the presence of South Col). However, the distribution could also be different, depending on the spatial variability of the wind velocity over this glacier, driving the relative importance of the accumulation versus ablation. Without further observations, nothing substantial can be said here and we prefer to remove all this section (lines 272 to 290 of the original manuscript), also to keep this paper as brief as possible.

Review of Brun et al., 2022: Minor comments

[RC1-5] Line 1 and 8: Is the glacier "iconic"? It's very high, but otherwise I suggest it is not widely known.
[ARC1-5] The adjective "iconic" has been removed line 1, but the word "icons" is kept line 8, because glaciers are often considered as icons of climate change in the media.

[RC1-6] L9: remove "large" as this is relative to the glacier in question; "...glaciers thin at rates often exceeding...".
[ARC1-6] Done

[RC1-7] L14 and elsewhere: check use of compound adjectives; hyphenation is not used with an adverb (ending in "-ly").
[ARC1-7] Done

[RC1-8] L24: "challenge of conducting scientific..."
[ARC1-8] Done

[RC1-9] L28: worth noting here that the South Col AWS recorded only about five months of data (May–end summer 2019). An earlier AWS at this location installed by the Ev-K2-CNR project measured three years of discontinuous data that did represent the entire annual cycle and could be of use if further field data are required.
[ARC1-9] Following this comment and Tom Matthews' clarification, we changed the sentence to provide information on the period of functioning of the South Col AWS: "...despite the installation of an Automatic Weather Station (AWS), which was running between May 2019 and August 2022, with some gaps, on a rock outcrop close to the South Col (Matthews et al., 2020)."
Thanks for the information concerning Ev-K2-CNR station.

Comparison of the results and conclusions of Brun et al. (2022) with those of Potocki et al. (2022), in consideration of the response by Potocki et al. in TCD.

**[RC1-10]** A key question addressed by both papers is; what is the duration of snow cover on the glacier surface? This would indicate when the bare ice surface is exposed to incoming solar radiation and ice melt could occur. However, the occurrence of seasonal melt does not imply net annual mass loss. Determining mass change over a representative timescale of several decades requires observations of longer-term change as provided by both papers.
In the case of both papers, I consider that the COSIPY model is unsuitable for application to South Col Glacier and the associated uncertainties render the results insignificant. My group's ongoing work applied COSIPY to Khumbu Glacier including the area occupied by South Col Glacier (https://doi.org/10.5194/egusphere-egu21-8663). COSIPY was forced by downscaled CORDEX RCM outputs and constrained by AWS data including the five months of data from the Nat Geo South Col AWS (Matthews et al., 2020). In each simulation, the net annual mass balance at the location of South Col Glacier was strongly positive (>7 m w.e. a–1). We can debate the strengths and limitation of any of these model parameterisations but any existing glacier surface energy-mass balance model is unlikely to be suitable for South Col Glacier due to the significant differences in the processes that control mass balance at 8,000 m a.s.l. compared to glaciers for which these models were developed at lower elevations where the mass balance is better understood by established glaciological theory. The different datasets used by each study (e.g., DEMs of difference/an ice core) are more important indicators of glacier mass change.
**[ARC1-10]** Thanks for this comment and contribution to the discussion. We fully agree with this comment, and please refer to ARC1-3 for a detailed reply.

**[RC1-11]** More important than debating the parameterisation of models that are likely not meaningful, we should consider the glaciological context of South Col Glacier. The elevation of the glacier is about 2,000 m above the equilibrium line altitude (ELA) for this region, determined for Khumbu Glacier as about 6,000–6,400 m a.s.l. (Rowan et al., 2015; 2021). While glaciers usually melt during the ablation season due to warm air temperatures and high incoming solar radiation, this does not equate to mass loss year-on-year. It is difficult to see why a glacier 2,000 m above the local ELA would have a net annual negative mass balance. Mayewski et al. TCD refute the suggestion by Brun et al. (2022) that their core is collected from the glacier ablation area. As the entire glacier is located well above the local ELA, the entire glacier should be "accumulation area" and therefore as a small cold-based glacier, the mass of South Col Glacier is likely to remain stable over decadal timescales.
**[ARC1-11]** Thanks for this comment, we agree with it, except maybe that the entire glacier should be an accumulation area. See ARC1-4 where we debate the fact that there is potentially an accumulation and an ablation area, on South Col glacier.

**[RC1-12]** Mayewski et al. TCD refer to the glacier as have a stagnant area. This term is used to describe the tongues of debris-covered glaciers such as Khumbu Glacier where the velocity of ice flow has declined rapidly as the glacier has lost mass in recent decades. The term is not accurately applied by Mayewski et al. TCD in context of South Col Glacier, which has not undergone a change in glacier dynamics but instead has a typical (slow, deformation only) flow regime as a cold-based glacier.
**[ARC1-12]** Thanks for this discussion, no reply needed here.

**[RC1-13]** As referenced by Mayewski et al. TCD, Figure 1 of Brun et al. (2022) shows areas of negative surface elevation change in the upper accumulation area of Khumbu Glacier. These bands are often interspersed with bands of positive surface elevation change, and I would interpret that they are evidence of large avalanches onto and within the glacier, and opening or closing of crevasses close to the bergschrund. Again, it is the net annual mass balance over a representative period of years (i.e., the integration of these features across the entire glacier) that tells us if the glacier is losing mass. These features in the DEMs of difference are not evidence of glacier mass loss but represent mass redistribution within the glacier. The mass gain of >7m a–1 predicted by our COSIPY simulations of South Col Glacier indicate the source of these avalanches—75% of accumulation to Khumbu Glacier and neighbouring valley glaciers occurs by avalanching of snow from the steep slopes, in some cases initiated by wind erosion of snow at the ridge crests (Benn and Lehmkuhl, 2000).

**[ARC1-13]** We agree with the reviewer's suggestion that the pattern of dH data evident in the upper reaches of the Khumbu Glacier (localised, alternating bands of positive and negative elevation change) are a result of crevassing due to ice flow over this particularly steep area of the Western Cwm. We also agree that these features are not necessarily good indicators of the overall mass budget of the glacier. Please see also ARC1-2 for a complete reply.

**[RC1-14]** Potocki et al. (2022) interpret their ice core as representing the accumulation area of the glacier and the age of the ice collected near the glacier surface (0.1–0.7 m core depth) is about 2,000 years old ("1966 ± 179 years ago"). This period is then multiplied by the annual layer thickness for the entire core (27 mm w.e. a–1) to estimate mass loss (apparently without any correction from water equivalent to ice thickness accounting for ice density?). This calculation assumes that the age of the ice at the glacier surface is the same as at the depth measured in the core and that the annual layer thickness is consistent throughout. From their Supplementary Information, it appears that annual layers were only measured in a 0.1 m section of the core at about 6 m depth. The representativeness of these values is determined by comparison with a core from East Rongbuk Glacier at 6,518 m (Kaspari et al., 2009). However, the annual layer thickness at South Col Glacier could be much thinner if wind erosion is accounted for. The snow deposition model results from Brun et al. (2022) suggest that at South Col Glacier nearly all precipitation can be eroded from the glacier surface by wind, which would not be the case at East Rongbuk Glacier or Khumbu Glacier where the majority of accumulation is sourced from avalanching. It is therefore possible that since the Sol Col Glacier last expanded and formed the moraines identified by Petocki et al. (2022) that most or all of the annual snow accumulation is scoured off by wind and that the exposed ice surface represents the last period when the glacier expanded. These moraines are undated at South Col Glacier, but there are three possible equivalent ice-marginal moraine ages at Khumbu Glacier dated to 1.3 ± 0.1 ka, 0.9 ± 02 ka and 0.6 ± 0.16 ka (Hornsey et al., 2022).

**[ARC1-14]** Thanks for this information and discussion, no reply needed here

**[RC1-15]** The 'space-for-time substitution' suggested by Mayewski et al. TCD reasons that because snow melts at Camp 2 in April–May then ice must melt at South Col Glacier in July–August. Their photographs show melt water on the surface of the accumulation area of Khumbu Glacier at Camp 2 (6,464 m a.sl.) on a patch of rock debris. The low albedo of the debris combined with high incoming solar radiation is likely to promote snow melt, but again is not evidence for net annual glacier mass change. The substitution reasoning is not convincing; South Col is up to 20 degrees colder based on the ERA data presented by Mayewski et al. TCD and the incident radiation would presumably be much lower than given

here due to monsoon cloud cover. The uncertainties in this estimate of seasonal melt seem similar or greater than those in the COSIPY experiments.

**[ARC1-15]** Thanks for this information and discussion, no reply needed here

**[RC1-16]** In summary, there are limitations to both the DEM differencing and ice core methods. I am convinced that both papers present their results accurately and have not made errors in their data processing. However, the DEM differencing presented by Brun et al. (2022) quantifies mass change across the glacier over a representative period of several decades. The ice core collected by Potocki et al. (2022) represents only one point on the glacier and is open to an alternative interpretation in context of the erosion of snow by wind from the glacier surface. I suggest that the model results are discounted as indicative of glacier mass balance due to the limitations of simulating this extreme environment without a dedicated model driven by spatially and temporally representative measurements from South Col Glacier. The remaining question is if there is value or feasibility in collecting direct glaciological measurements from South Col Glacier. Installing and managing equipment over a sufficient timescale (>5 years) would be very challenging and expensive. We therefore need to rely on high-quality remotely sensed observations, including those presented by Brun et al. (2022), which in the last 15 years or so have greatly improved understanding of recent glacier change in the Himalaya.

**[ARC1-16]** We agree that the DEM differencing method has limitations, and we have made a concerted effort to more clearly describe and quantify the potential errors and uncertainties associated with this method in the revised manuscript (see ARC1-2). This method has been widely used and improved by the remote sensing cryospheric community in the last decades, and a thorough quantification of the uncertainties has been developed by Hugonnet et al. (2022), and applied in our present study. We are therefore confident that our elevation change results reflect the fact that South Col Glacier did not thin over the last 3 decades. This is indeed the main conclusion of our study, and in turn the title of our brief communication. This conclusion is in contradiction with results from Potocki et al. (2022), obtained from an ice core analysis, and we tried to understand why.

As already mentioned in ARC1-3, the goal of the numerical simulations of surface mass balance that have been performed in our study is not to assess whether significant melt actually occurs at SCG, but to demonstrate that a firm answer to this question is beyond the reach of available numerical models which are not designed for such specific environments. Yet, we acknowledge that this message was not that clear due to our initial choice to include simulations performed with both a modified version of Cosipy and Crocus models. Therefore, all references to Crocus have been withdrawn from the new version of the manuscript. We now focus our demonstration on the sensitivity of Cosipy results in terms of melt to choices in the numerical implementation of physical processes. More specifically, we show that an alternative numerical treatment of the subsurface heat flux in the surface energy balance module of Cosipy that is more conformal to the physics and consistent with implementations adopted in other skin-layer models (e.g., Covi et al., 2022 and other references to DBAM) turns out to produce fundamentally different results. We believe that this is enough to show that uncertainties on the results produced with Cosipy at SCG are so large that no conclusion can be drawn from them. We think that this demonstration is relevant to support our conclusion and, therefore, that this modelling section is worth keeping in our brief communication.

References

J. Höhle and M. Höhle, "Accuracy assessment of digital elevation models by means of robust statistical methods," ISPRS J. Photogramm. Remote Sens., vol. 64, no. 4, pp. 398–406, Jul. 2009.

Hugonnet, R., McNabb, R., Berthier, E., Menounos, B., Nuth, C., Girod, L., Farinotti, D., Huss, M., Dussaillant, I., Brun, F., and Kääb, A.: Accelerated global glacier mass loss in the early twenty-first century, Nature, 592, 726–731, https://doi.org/10.1038/s41586-021-03436-z, 2021.

Hugonnet, R., Brun, F., Berthier, E., Dehecq, A., Mannerfelt, S., Eckert, N., Farinotti, D.: Uncertainty analysis of digital elevation models by spatial inference from stable terrain, IEEE Journal of Selected Topics in Applied Earth Observations and Remote Sensing, 1–17, doi.org/10.1109/JSTARS.2022.3188922, 2022.

Covi, F., Hock, R., & Reijmer, C. (2022). Challenges in modeling the energy balance and melt in the percolation zone of the Greenland ice sheet. *Journal of Glaciology,* 1-15. doi:10.1017/jog.2022.54

References

Benn DI, Lehmkuhl F. 2000. Mass balance and equilibrium-line altitudes of glaciers in high-mountain environments. Quaternary International 65–66 : 15–29. DOI: 10.1016/S1040-6182(99)00034-8

Brun, F., King, O., Réveillet, M., Amory, C., Planchot, A., Berthier, E., Dehecq, A., Bolch, T., Fourteau, K., Brondex, J., Dumont, M., Mayer, C., and Wagnon, P. 2022. Brief communication: Everest South Col Glacier did not thin during the last three decades, The Cryosphere Discussions. [preprint], https://doi.org/10.5194/tc-2022-166.

Hornsey J, Rowan AV, Kirkbride MP, Livingstone SJ, Fabel D, Rodes A, Quincey DJ, Hubbard B, Jomelli V. 2022. Be-10 Dating of Ice-Marginal Moraines in the Khumbu Valley, Nepal, Central Himalaya, Reveals the Response of Monsoon-Influenced Glaciers to Holocene Climate Change. Journal of Geophysical Research: Earth Surface 127 DOI: 10.1029/2022JF006645

Kaspari S, Hooke RLeB, Mayewski PA, Kang S, Hou S, Qin D. 2008. Snow accumulation rate on Qomolangma (Mount Everest), Himalaya: synchroneity with sites across the Tibetan Plateau on 50–100 year timescales. Journal of Glaciology 54 : 343–352. DOI: 10.3189/002214308784886126

King O, Bhattacharya A, Ghuffar S, Tait A, Guilford S, Elmore AC, Bolch T. 2020. Six Decades of Glacier Mass Changes around Mt. Everest Are Revealed by Historical and Contemporary Images. One Earth 3: 608–620. DOI: 10.1016/j.oneear.2020.10.019

Matthews, T., Perry, L. B., Koch, I., Aryal, D., Khadka, A., Shrestha, D., Abernathy, K., Elmore, A. C., Seimon, A., Tait, A., Elvin, S.,Tuladhar, S., Baidya, S. K., Potocki, M., Birkel, S. D., Kang, S., Sherpa, T. C., Gajurel, A., and Mayewski, P. A.: Going to Extremes: Installing the World's Highest Weather Stations on Mount Everest, Bulletin of the American Meteorological Society, 101, E1870–E1890, https://doi.org/10.1175/BAMS-D-19- 0198.1, 2020.

Potocki, M., Mayewski, P.A., Matthews, T. et al. Mt. Everest's highest glacier is a sentinel for accelerating ice loss. npj Climate Atmospheric Sciences 5, 7 (2022). https://doi.org/10.1038/s41612-022-00230-0

Rowan AV, Egholm DL, Quincey DJ, Glasser NF. 2015. Modelling the feedbacks between mass balance, ice flow and debris transport to predict the response to climate change of debris-covered glaciers in the Himalaya. Earth and Planetary Science Letters 430 : 427–438. DOI: 10.1016/j.epsl.2015.09.004

Rowan AV, Egholm DL, Quincey DJ, Hubbard B, King O, Miles ES, Miles KE, Hornsey J. 2021. The Role of Differential Ablation and Dynamic Detachment in Driving Accelerating Mass Loss From a Debris-Covered Himalayan Glacier. Journal of Geophysical Research: Earth Surface 126 DOI: 10.1029/2020JF005761

**Referee comment 2:**

[RC2-1] This brief communication by Brun et al. is an important and timely response to the recent paper by Potocki et al. (2022) reporting dramatic ice loss from the Everest South Col Glacier. From differences in photogrammetric DEMs rather than dating an ice core, Brun et al. concluded that the 1984-2017 surface elevation change did not statistically differ from zero. Both studies then attempt surface mass balance modelling to interpret their results. The authors of Potocki et al. (2022) have already responded in the discussion. Implications of their comment that Brun et al. have given the wrong elevation for the ice core should be considered. Rather than the comment that the modelling by Brun et al. challenges the possibility of such large thinning rates, however, I would say that they have demonstrated that uncertainty precludes firm conclusions from modelling in this case.

[ARC2-1] We want to acknowledge the anonymous reviewer for this assessment of our study, and for considering that our contribution is an important response to Potocki et al. (2022). We want to stress that the modelling used in Brun et al. (2022) does not aim at interpreting the results from DEM differencing (i.e. no thinning between 1984 and 2017) but rather illustrate the sensitivity of the models (Cosipy in our case) to arbitrary choices in the numerical and physical treatment of the surface energy balance. Our results show that uncertainties are large (for melt for instance) and consequently, no conclusion concerning surface processes can be drawn so far. See reply ARC1-3 for an exhaustive reply.

In figures 1 and A6 of Brun et al. (2022), the location reported for the ice core extracted for the Potocki et al. (2022) study is correct, with a horizontal approximate accuracy of +/- 15 m originating both from the coordinate precision provided in Potocki et al. (2022) and the uncertainty in the Pléiades geolocation. The elevation difference comes from the fact that in Brun et al. (2022), the elevation corresponds to height relative to the ellipsoid WGS84, and not the geoid. This was specified in the Figure A6 caption, but not in Fig 1 caption. Now all elevations are expressed as heights above the geoid, as suggested by reviewer 4 (see ARC4-14), and the contour lines in figure 1 have been relocated accordingly.

Minor corrections

[RC2-2] Abstract - No need to be so cautious: "This is in contradiction"
[ARC2-2] Done

[RC2-3] line 202 - "melt that immediately refreezes within the same time step could occur"
[ARC2-3] This sentence has been removed from the manuscript.

[RC2-4] line 276 – "we suggest that the core"
[ARC2-4] Done

[RC2-5] line 370 - "$E_p$ is" or "$E_p$ is given by"
[ARC2-5] Done, we replaced "writes" by "is given by"

[RC2-6] line 382 - "$u_{*t}$"

**[ARC2-6]** Done: *t has been put in indice

**[RC2-7]** line 395 - "(one hour)"
**[ARC2-7]** Done

**[RC2-8]** line 399 - "the former thicknesses of each layer"
**[ARC2-8]** This is line 385: thickness is now plural, as suggested

**[RC2-9]** line 436 - "is the ice thickness"
**[ARC2-9]** Fixed

**[RC2-10]** Figure 1 - The inset showing the location of Mt Everest is not referred to and is not necessary.
**[ARC2-10]** We prefer to keep the inset to locate Mount Everest. The inset is now referred to in the caption.

**Referee comment 3:**

Review comments on "Brief Communication: Everest South Col Glacier did not thin during the last three decades" by Fanny Brun et al.

1. General comments:

**[RC3-1]** This paper reports the surface elevation change of a small (0.2 km2) Himalayan glacier located at a high elevation (~8000 m a.s.l.) for a period from 1984 to 2017. The analysis was performed by comparing two DEMs constructed from aerial photographs taken in 1984 and satellite images acquired in 2017. The motivation of the study is a recent publication (Potocki et al., 2022), which estimated an ice thinning rate of ~2 m a−1 based on the analysis of an ice core drilled from this glacier and surface mass balance modeling. In contrast to the rapid thinning rate reported by Potocki et al., the DEM differencing showed little change in the surface elevation. To explain the two inconsistent results, numerical experiments on wind erosion of snow and surface mass balance were performed, as well as inspections of glacier surface conditions with satellite images. Based on the series of analyses, the authors concluded that ablation due to melting was overestimated by the numerical experiment by Porocki et al. (2022).

Considering the importance of glacier changes in the Himalayas as well as the unique location of the studied glacier, the estimate of ~2 m of ice loss every year at 8000 m a.s.l. has a large impact on the research community and society. Therefore, I appreciate the authors' effort to inspect the glacier change with a different approach. I think the DEM analysis is reliable enough to exclude the possibility of such rapid thinning. Therefore, I support the swift publication of this manuscript on Cryosphere.

**[ARC3-1]** We appreciate the positive evaluation of our paper, and want to thank the reviewer

2. Concerns

**[RC3-2]** (1) Numerical modeling

**[RC3-2a]**It is a good idea to report the result of DEM differencing as a short article. However, the manuscript is not really a "Brief Communication". The effort of the authors is acknowledged, but in my opinion, this glacier is not suitable for numerical experiments using a model developed somewhere else.
**[ARC3-2a]** We agree with the last sentence and please look at ARC1-3 for a detailed reply. We want to keep the communication as brief as possible, and some sections have been shortened such as all considerations concerning glacier flow, and ablation/accumulation areas. We also removed all simulations with Crocus and the referring text as well, to clarify the message and to keep the mass/energy balance modeling part as simple as possible.

**[RC3-2b]**Moreover, the importance of snow erosion is clear on such a location even without numerical simulations. The satellite images tell us a lot more than the erosion model.

**[ARC3-2b]** We agree with the reviewer but still think that the deposition model and the Venµs images analysis complement each other well. Moreover, the model allows a rough quantification of the deposition efficiency at this high altitude site, which is valuable for our analysis. Regarding snow erosion, we thus prefer to keep both numerical simulations and satellite image analysis.

**[RC3-2c]** My suggestion to the authors is to keep the modeling part as simple as possible. For example, experiments with shorter spatial and temporal resolutions of the COSIPY mass balance model nicely showed that heat conduction into the ice was possibly missed in Potocki et al. (2022). However, I am worried about the use of Crocus because the model is not validated in the extreme environment of the studied glacier. Why not simply compare the two COSIPY models to discuss possible shortcomings?

**[ARC3-2c]** We agree with the reviewer's analysis and decided to remove all sections and simulations with the Crocus model. Indeed, Crocus is not evaluated in the extreme environment of the studied glacier, and we do not have any way to evaluate it. As a consequence, it does not provide relevant additional information compared to the sensitivity tests performed with Cosipy. Removing the Crocus section allows to keep the modeling as simple as possible, and to reduce the length of the paper to respect a brief communication format.

**[RC3-3]** (2) Retention, refreezing and superimposed ice

I am wondering if the authors consider retention of meltwater in a firn layer or ice crucks, and subsequent refreezing and superimposed ice formation. I believe these are important processes related to melt in cold environments. Isn't it likely that melt happens, but it refreezes and does not leave the glacier?

**[ARC3-3]** We agree with the reviewer that refreezing, in a firn layer or deeper in the glacier, is likely to play a role in the case of surface melting in such a cold environment. Looking at refreezing is definitely an important point to accurately assess the glacier mass balance  but it would require some specific model. Indeed, a precise quantification of the refreezing process is beyond the capacity of the simple modeling approach applied in this study, and would require a dedicated glacial hydrology model. This is beyond the scope of our brief communication.
Moreover, in our study, we intentionally applied the COSIPY model (with different numerical implementations) in an extreme and unrealistic case when ice is always exposed at the glacier surface, to maximize the mass loss in order to test whether Potocki et al. (2022) results are reliable. In such a case of an icy surface, the COSIPY model predicts that the totality of the melt water is evacuated from the glacier, and no refreezing occurs. This unrealistic configuration is useful to test whether Potocki et al. (2022) melt rates are robust but does not allow us to quantify refreezing or superimposed ice formation.
Anyway, our purpose is not to provide an accurate modeling of the SCG surface mass balance, but rather to highlight that the default COSIPY outputs lack robustness when applied to the SCG (see ARC1-3). We do not think that the inclusion of refreezing processes is necessary to achieve this goal.

A new sentence has been added in the revised manuscript to address this issue: "Additionally, even though there were some melt on this glacier, it is likely that a large amount of this meltwater would refreeze at the surface in case of the presence of snow or firn, or would form superimposed ice. While COSIPY accounts for refreezing in snow, in the case of pure ice all the melted water percolates and finishes as runoff , limiting its applicability in such a cold context."

**[RC3-4]** (3) Setting an "ablation area" (Line 273–289, Fig. A6)

The authors set a boundary of ablation and accumulation areas to assess the importance of glacier flow in the ice thickness change. However, it is odd to set such an imaginary boundary because the idea of accumulation and ablation zones does not work on such a small glacier. Further, the assumption of uniform emergence velocity (or thickening due to vertical straining) over the "ablation area" is not realistic. My suggestion is to estimate the velocity and its gradient from the ice thickness and temperature to confirm 2 m of thickening due to vertical straining is not possible at the coring site.

**[ARC3-4]** We agree with this comment and decided to remove all this section regarding ice flow, ablation and accumulation area (see ARC1-4 for a complete reply). We do not know how feasible it is to estimate the velocity and its gradient from the ice thickness and temperature since we do not have measurements neither of ice thickness nor of ice temperature.

3. Specific comments

**[RC3-5]** Line 19: "estimated that contemporary thinning rates — or ablation rates," >> This is confusing. What was estimated by Potocki et al. (2 m a−1) is "negative surface mass balance", I think.
**[ARC3-5]** Agreed. We changed "ablation rates" into "negative surface mass balance"

**[RC3-6]** Line 27: "Automatic Weather Station" >> automatic weather station
**[ARC3-6]** Done

**[RC3-7]** Line 32: "1.5 m a−1" >> Here and in other places, please make it clear if it is water equivalent, snow depth, or ice equivalent.
**[ARC3-7]** Fixed

**[RC3-8]** Line 76-77: This is already mentioned in Line 25.
**[ARC3-8]** Agreed. We removed the L25.

**[RC3-9]** Line 106: The terminology is not clear to me because: (1) erosion occurs after snow deposits on the surface and (2) precipitation includes snow drifting away before deposition. Why not like this?

- Precipitation: all snow falling on the glacier surface
- Deposition: snow attached to the glacier surface, a part of the precipitation
- Erosion: snow removed from the glacier surface after the deposition
- Accumulation: deposition minus erosion

**[ARC3-9]** We indeed use a different terminology in our paper, considering that deposition = precipitation - erosion compared to the reviewer's definition where our deposition = their accumulation, and we do not consider snowdrift. The semantics on this subject are not yet consensual and still vary greatly from one paper to another. Our main concern is that processes are defined without ambiguity for the reader. We agree to follow the terminology suggested by the reviewer, which may provide a clearer description of the accumulation/ablation processes. The manuscript has been adapted accordingly, and we added the following text at the beginning of section 3.2: "Hereafter we define accumulation as the snow that is deposited to the glacier surface (i.e. a fraction of precipitation) minus the erosion (i.e. the snow that is removed from the glacier surface after the deposition). The accumulation can thus be negative when erosion exceeds deposition."

**[RC3-10]** Line 110-111: "… the most similar …" >> Are you talking about the inland of the Antarctic ice sheet? Isn't it much drier than the studied glacier? I do not think high elevations in the Himalayan mountains and Antarctica are so similar.

**[ARC3-10]** We agree that such comparison is subjective, and not very meaningful given that Antarctica has very different and diverse weather conditions, and that we do not have any quantitative comparison. We removed this sentence then.

**[RC3-11]** Line 114: "offline nature" >> What do you mean? The erosion model is decoupled from the climate model?

**[ARC3-11]** The snow erosion module in MAR is implemented directly in the surface scheme of the model, which is coupled to the atmospheric module. In our paper, instead of using a climate model, we extracted the physics from the surface scheme of MAR to compute erosion rates using reanalysis data as inputs, with no interactions with the atmosphere. In that sense it is an off-line approach. However, for clarification we rephrased as follows:e:

"Here we develop a simplified analytical approach expressed in a 1D vertical framework, in which the erosion model is only forced by the meteorological variables with no further interactions between the surface and the atmosphere."

**[RC3-12]** Line 116: "as a function of surface snow density only" >> Wind speed?

**[ARC3-12]** *Yes erosion rates are computed as a function of wind speed too. This is now specified in the text.*

**[RC3-13]** Line 127-128: Not clear what "uncorrected precipitation" and "tuned estimates" are. Can you clarify the sentence?

**[ARC3-13]** The sentence has been clarified as follow:

"However, instead of using artificially reduced precipitation (averaging at 66.9 mm a$^{-1}$) to implicitly account for wind erosion which is missing in Potocki et al. (2022), we prescribe uncorrected precipitation rates (averaging at 191 mm a$^{-1}$) as we intend to explicitly model wind erosion."

**[RC3-14]** Line 138: "falling snow is not eroded" >> It sounds odd because erosion occurs for deposited snow, but not for falling snow.

**[ARC3-14]** This is a good point, the sentence is ambiguous. By following the terminology suggested in comment RC3-9, it will naturally become clearer. We rephrased the entire sentence as "A large proportion of deposited precipitation is not eroded, and the accumulation efficiency gradually increases."

**[RC3-15]** Line 142: "191 mm w.e." >> Is this what you wrote in Line 128? If yes, please avoid repetition. Please also be consistent with the unit.

**[ARC3-15]** The bracket has been removed to avoid repetition, but we included the adjective "uncorrected" to avoid any confusion.

"… the annual uncorrected precipitation ranges from 147 to 259 mm w.e., and only 0 to 51% …"

**[RC3-16]** Line 149: "The wind erosion model is simple and has large limitations."?

**[ARC3-16]** rephrased accordingly, thanks

**[RC3-17]** Line 150: "act as a negative feedback" >> It sounds strange to me that density increases as a function of erosion, because Equation A7 is not like that. Maybe, "regulate"?

**[ARC3-17]** Equation A7 is only called if erosion occurs: densification occurs when the model erodes, and an increase in density decreases the erosion rate. We followed your suggestion.

**[RC3-18]** Line 151: "snowfalls disappear" >> It sounds odd if you mean snow disappears from the glacier surface. "snow on the glacier disappears"?

**[ARC3-18]** Thanks, we accepted your phrasing

**[RC3-19]** Line 155: "predicted" >> "reproduced"?

**[ARC3-19]** Accepted

**[RC3-20]** Line 160: "eroded or re-mobilized after deposition" >> This is the correct use of "deposition", but it is wrong according to the definition by the authors.

**[ARC3-20]** We have adapted the terminology so this sentence is now right. Thanks.

**[RC3-21]** Line 162-164: Please revise this sentence because (1) it is self-evident that "deposition efficiency is not constant", and (2) it does not imply "erosion is a major ablation process", and (3) the last clause "that is not constant in time" is redundant.

**[ARC3-21]** About (1), the temporal variability of erosion is mentioned in comparison to Potocki et al. who implicitly take into account the effect of erosion by reducing precipitation by a constant factor over time (see our response ARC3-13). About (2), we rely on the fact that a large part or all of deposited snow is eroded (which corresponds to periods of low to negative values of the deposition efficiency in Fig 3b in the original manuscript) to suggest that erosion is a major ablation process over the corresponding specific time periods. About (3), you're right, this is redundant. We rephrased accordingly. The new sentence now reads:

"Second, the fact that the deposition efficiency is not constant in time, together with the low to negative values of the deposition efficiency at the end of the monsoon, suggest that (1) the temporal variability of accumulation cannot be properly resolved by reducing precipitation by a constant factor over time, and that (2) wind erosion is a major ablation process in this area."

**[RC3-22]** Line 167: "thus integrate the surface energy balance over a much longer period" >> What do you mean?

**[ARC3-22]** This sentence was not clear, so we removed it.

**[RC3-23]** Line 202: "15 min" >> Isn't it 1 min as stated in Line 186?
**[ARC3-23]** This sentence was removed.

**[RC3-24]** Line 287: "velocity deformation" >> "velocity due to ice deformation"?
**[ARC3-24]** Yes, you are right and it is changed accordingly. Anyway, this section has been removed.

**[RC3-25]** Line 293: "continental type" >> This sounds odd. I think ice flows slowly because the glacier is small.
**[ARC3-25]** This section has been removed.

**[RC3-26]** Line 295: "incoming precipitation depositions" >> Is this term usual in glaciology? I have never seen it before.
**[ARC3-26]** Not very usual in glaciology indeed. This has been changed into:
"A large fraction of precipitation (> 60 %) is eroded, limiting the accumulation"

**[RC3-27]** Line 319: "impossible" >> Maybe "very difficult"?
**[ARC3-27]** Changed accordingly

**[RC3-28]** Figure 4 caption Line 3: "predicted" >> "estimated" or "simulated"?
**[ARC3-28]** "Predicted" replaced by "estimated"

**Referee comment 4 - Horst Machguth and Enrico Mattea**

Review of "Brief communication: Everest South Col Glacier did not thin during the last three decades" by Brun et al.

Horst Machguth and Enrico Mattea, Department of Geoscience, University of Fribourg, Switzerland

**1. Introduction**

**[RC4-1]** The study of Brun et al. argues that South Col Glacier has not changed substantially since 1984. Their finding contradicts Potocki et al. (2022) – in the following Potocki et al. – who claim that the glacier has thinned at their drill site (8020 m a.sl.) by about 55 m. The timing of the thinning is unclear but it is suggested by Potocki et al. that the climate at that elevation warmed substantially in the 1950s and even more substantially in the later 1990s. We divide out review in a general section which refers to both studies, and a section of detailed comments focusing on Brun et al.

**[ARC4-1]** We thank Horst Machguth and Enrico Mattea for their detailed review of our communication and for bringing new material contributing to the overall discussion

**2. General comments on South Col Glacier changes**

**[RC4-2]** Brun et al. contradict that South Col Glacier has thinned dramatically. They do so by comparing two digital elevation models representing different points in time. The evidence provided by Brun et al. appears sound and corresponds to state-of-the-art. Nevertheless, we chose an independent way of assessing whether the glacier has changed or not. To do so, we compared historical photos of South Col Glacier to recent images. We obtained images taken during the 1956 Swiss Everest/Lhotse expedition; the second ever expedition to summit Everest and the first to summit Lhotse. The images are publicly accessible at https://alpinfo.ch/en/portrait/historical-notes/expeditions/everest-lhotse- 1956/ . We have also considered photos from the two Swiss 1952 Everest expeditions (also publicly available via the above link), but found that images from 1956 are optimal for the comparison. We compare two historical images from 1956 to recent images. Figure 1 shows the perspective from somewhere in the vicinity of the South Col AWS, looking slightly down on the plateau of South Col Glacier. Figure 2 shows the view towards the tongue of South Col Glacier, looking upward from what was maybe Camp V or VIa during the 1956 expedition.

[Figure]

[Figure]

Fig. 1: Comparison of a historical image of South Col Glacier, recorded in May 1956 (left) to an image taken in May 2022 (right). For scale an approximate elevation difference has been drawn based on the Pléiades DEM by Brun et al. Note that the two images were taken from two different viewpoints, with the 1956 picture taken from a point somewhat more east and closer to South Col. Image courtesy of the Swiss Foundation of Alpine Research (1956) and Tim Mosedale (2022, https://timmosedale.co.uk/).

The comparison in Fig. 1 shows, if at all, small changes in South Col glacier. There is no support for the claim of Potocki et al. that the glacier has thinned in excess of 50 m. The near-absence of changes is most obvious at the glacier margins. Admittedly, changes are more difficult to assess in the glacier centre where Potocki et al. have drilled. Nevertheless, the glacier appears also to have changed little at the drill site. If the glacier would have thinned 55 m in the centre, then one would expect an even more pronounced change at its tongue. Such a thinning would be obvious as the ice cliff is in close proximity to the drill site (less than 200 m apart). However, recent imagery shows that the glacier tongue is at the same location as it was in 1956 (Fig. 2) and the ice thickness appears unchanged. The tongue appears similarly active as in the 1950s (see for example an excellent 2022 overview of South Col Glacier: https://www.mountainpanoramas.com/___p/___p.html?panoid=2022_M1&labels=on).

[Figure]

Fig. 2: Tongue of South Col Glacier in 1956 and 2008. Left: detail of a historical image of South Col Glacier, recorded in May 1956 by the Swiss Everest/Lhotse expedition. Right: detail of an image taken in May 2008. Note that the two images were taken from somewhat different viewpoints, with the 1956 image taken from a higher elevation. Image courtesy of the Swiss Foundation of Alpine Research (1956) and https://spaceref.com/science-and-exploration/scott-parazynski-everest-photo- update-4-june-2008 (2008).

We investigated whether glacier changes are detectable near the actual South Col where the glacier surface appears very flat and the ice looks rather thin (directly east (right) of the "Everest South Col" marker in the linked panoramic photograph). Interestingly, also there the situation in 1956 appears similar to more recent photos (not shown). We note that there are many good images available for South Col, for different points in time, providing excellent possibilities to investigate whether glacier changes took place or not.

[ARC4-2] We thank Horst Machguth and Enrico Mattea for backing up very convincingly our DEM analysis using high quality photography. They indeed confirm the lack of thinning (and frontal retreat) of South Col Glacier. We refer to the material published in this review by adding: "This observation can be extended back in the past, as the comparison of photographs from the Swiss expedition to Everest in 1956 with photographs from 2022 shows that there are no visible changes in South Col Glacier (Machguth and Mattea, 2022)."

3. General comments on the surface mass balance modelling

We argue that uncertainties in available model input parameters are too large to make any reliable statement on the mass and energy balance at South Col Glacier, based on model

simulations. These uncertainties also affect the model comparison presented in Brun et al. We detail our argumentation on the example of ice albedo at South Col and by running a third surface mass and energy balance model for South Col Glacier.

**[RC4-3] The problematic of parameter uncertainties:** Both Brun et al. and Potocki et al. use an ice albedo value of 0.4. The value has been measured at Base Camp, at approximately 5400 m a.sl. (Matthews et al., 2020). The ice at the surface of South Col Glacier is referred to as blue ice (Brun et al.). Albedo values for Antarctic blue ice are substantially higher, typically in the range of 0.6 to 0.65. (e.g. Reijmer et al. 2001; Genthon et al. 2007; Smedley et al. 2020). While Antarctica's blue ice areas might not be representative for South Col blue ice, it is also questionable to use an albedo value measured on glacier ice 2500 m lower than South Col. At Base Camp melt processes and surface ice conditions differ substantially from South Col. Also, the glacier ice has been formed under different conditions (cold on South Col vs. possibly temperate at Khumbu glacier), further affecting ice albedo.

Both Potocki et al. and Matthews et al. (2020) do not perform sensitivity tests with ice albedo, regardless of the extreme importance their studies assign to short wave radiation. In the case of South Col, critical uncertainties are not limited to surface albedo alone. A parameter sensitivity study is mandatory as soon as some parameters have relevant uncertainties. In the case of South Col, even a simple sensitivity study, as demonstrated below, might show that the range in possible outcomes of model simulations is simply too large for model results to be deemed reliable.

A thorough sensitivity analysis is also missing in Brun et al. While COSIPY is used in two different constellations of model numerics, CROCUS is not subject to any sensitivity assessment. Sensitivity to uncertainties in model input has neither been evaluated for the COSIPY variants nor for CROCUS. We understand that this is beyond the scope of the study by Brun et al. They also clearly state that uncertainties in any simulation for South Col are too large, given our current knowledge of meteorological conditions and mass balance. Nevertheless, we gained the impression that Brun et al. somewhat consider CROCUS the benchmark for other models. While this could be true, it would require demonstrating that CROCUS is more robust to changes in model numerics and comparing sensitivity of all models to input parameter perturbations. In this sense, we would like to ask Brun et al. to check their manuscript for any explicit or implicit "model hierarchy" and to further emphasize the problematic of poorly constrained model input and the absence of a parameter sensitivity study.

**[ARC4-3]** We totally agree with Horst Machguth and Enrico Mattea that, because in-situ atmospheric and glaciological data at SCG are lacking, many important model parameters, including albedo, and forcings are poorly constrained. It follows that any modeling study aiming at producing realistic estimates of surface mass balance at SCG should include a parameter sensitivity study. However, as the reviewers noticed, this is not our goal here. Instead, we wanted to highlight the lack of robustness of any modeling results in such specific conditions. We acknowledge that the inclusion of Crocus in the original version of the manuscript might have blurred this message. We do not consider Crocus as the benchmark for other models. Indeed, neither Crocus nor any other available mass-balance models are validated in the extreme environment of the studied glacier, and such a validation is simply out of reach due to the lack of data. Therefore, we have decided to withdraw Crocus simulations in the new version of the manuscript. We now focus on demonstrating the lack of reliability of results produced by Cosipy for this glacier. Indeed, these results turn out to be thoroughly different if another equally-reasonable numerical treatment of the subsurface heat flux in the surface energy balance is implemented instead of the original one. We think that this is enough to show that uncertainties on obtained results are so strong that no conclusion can be drawn from them. In such a context and to address our objective, we believe that a parameter sensitivity study is useless and is even meaningless. See also ARC1-3 for a detailed reply concerning the objectives of the mass-energy balance modeling

**[RC4-4] Simulating South Col Glacier surface mass and energy balance using EBFM:** The energy balance and firn model (EBFM, van Pelt et al., 2012) was developed following Klok and Oerlemans (2002) and the subsurface model SOMARS by Greuell and Konzelmann (1994). The model has recently been modified for use on Abramov Glacier, Kyrgyzstan (Kronenberg et al., 2022) and for Colle Gnifetti, Swiss Alps (Mattea et al., 2021). Here we deploy the version by Mattea et al. (2021) for a series of South Col model sensitivity experiments.

The model uses a skin layer formulation, calculating surface energy fluxes from meteorological variables. The surface energy balance equation is solved for surface temperature and mass fluxes, including melt and sublimation rates. Surface albedo is bounded by constant values for fresh snow, firn, and ice (respectively $\alpha_{fresh}$, $\alpha_{firn}$ and $\alpha_{ice}$); it evolves as an exponentially decaying function of time since the last significant snowfall (defined by a minimum precipitation rate $P_{min}$).

At each time-step, the computed surface boundary conditions drive a Lagrangian simulation of the glacier subsurface: it consists of a stack of NL layers able to move freely along the depth axis, following the addition or removal of mass at the surface. A new layer is added at the top whenever snowfall and riming push the topmost layer thickness beyond threshold $z_s$.

Notable omissions in the EBFM include penetration of short-wave radiation and wind erosion of snow (less significant for the simulation of an ice surface). As in COSIPY and Crocus, terrain reflections and topographic shading are also ignored; they are expected to play a minor role in the overall energy balance (e.g. Mattea et al., 2021).

We use the same downscaled ERA5 data of Potocki et al. and Brun et al. to force the model. Table 1 reports the main results of our sensitivity runs.

Table 1: summary of sensitivity EBFM runs. Shaded column headings indicate model parameters, the other columns are model results. Melt M is shown in bold.

| Id | Period | Spinup | NL | $P_{min}$ | $z_s$ | $\alpha_{fresh}$ | $\alpha_{firn}$ | $\alpha_{ice}$ | M | S | D | E | C | $\alpha_{mean}$ | $Q_g$ |
|---|---|---|---|---|---|---|---|---|---|---|---|---|---|---|---|
| 1 | 2000-2019 hourly | None | 50 | 2.5e-8 | 0.10 | 0.83 | 0.52 | 0.39 | **0.0007** | 0.241 | 0.012 | 0.000 | 0.000 | 0.80 | 0.25 |
| 2 | 2000-2019 hourly | None | 50 | 2.5e-6 | 0.10 | 0.83 | 0.52 | 0.39 | **0.7750** | 0.428 | 0.004 | 0.062 | 0.000 | 0.47 | 3.2 |
| 3 | 2000-2019 hourly | None | 50 | 2.5e-6 | 0.01 | 0.83 | 0.52 | 0.39 | **0.5948** | 0.489 | 0.002 | 0.039 | 0.000 | 0.45 | 17 |
| 4 | 2000-2019 hourly | None | 2000 | 2.5e-6 | 0.01 | 0.83 | 0.52 | 0.39 | **0.5201** | 0.487 | 0.002 | 0.036 | 0.000 | 0.46 | 18 |
| 5 | 2000-2019 hourly | None | 500 | 2.5e-6 | 0.01 | 0.83 | 0.52 | 0.39 | **0.5529** | 0.493 | 0.002 | 0.038 | 0.000 | 0.46 | 20 |
| 6 | 2019 minutely | None | 500 | 2.5e-6 | 0.01 | 0.83 | 0.52 | 0.39 | **0.3539** | 0.327 | 0.005 | 0.036 | 0.000 | 0.63 | 6.9 |
| 7 | 2019 hourly | None | 500 | 2.5e-6 | 0.01 | 0.83 | 0.52 | 0.39 | **0.3520** | 0.342 | 0.005 | 0.032 | 0.000 | 0.63 | 11 |
| 8 | 2000-2019 hourly | None | 500 | 2.5e-6 | 0.01 | 0.83 | 0.60 | 0.60 | **0.0490** | 0.355 | 0.007 | 0.007 | 0.000 | 0.65 | 7.8 |
| 9 | 1950-1999 hourly | None | 500 | 2.5e-6 | 0.01 | 0.83 | 0.60 | 0.60 | **0.0282** | 0.323 | 0.006 | 0.005 | 0.000 | 0.65 | 6.8 |
| 10 | 2000-2019 hourly | 50 yr, run 9 | 500 | 2.5e-6 | 0.01 | 0.83 | 0.60 | 0.60 | **0.0178** | 0.351 | 0.007 | 0.004 | 0.000 | 0.65 | 7.8 |
| 11 | 1950-1999 hourly | None | 500 | 2.5e-6 | 0.01 | 0.83 | 0.52 | 0.39 | **0.3486** | 0.439 | 0.003 | 0.026 | 0.000 | 0.48 | 13 |
| 12 | 2000-2019 hourly | 50 yr, run 11 | 500 | 2.5e-6 | 0.01 | 0.83 | 0.52 | 0.39 | **0.4609** | 0.480 | 0.003 | 0.032 | 0.000 | 0.46 | 14 |

1. M, S, D, E and C are annual means of melt, sublimation, deposition, evaporation and condensation. αmean is the mean surface albedo. Qg is the mean annual subsurface heat flux.
2. Pmin isinmw.e.s-1;zs inm;M,S,D,E,Cinmw.e.yr-1;Qg inWm-2;allotherparametersare dimensionless.
3. Changes in model setup from one model run to the next are highlighted.

The ERA5 meteorological series contains extremely frequent, small precipitation events, which constantly reset surface albedo to the fresh snow value (αmean in run 1). As such, in runs 2 through 12 we increase Pmin by 100x, to effectively disable the albedo from being restored in most cases.

With a layer thickness limited at 10 cm and a time-step of 1 hour (run 2), the EBFM calculates mean annual melt amounts of 0.78 m w.e. over 2000-2019, which corresponds to half of the values reported by Potocki et al.; notably, mean albedo over the modeled period is 0.47, which is about 25 % lower than Antarctic blue ice albedo.

Even before altering physical parameters, we note that the numerical setup has a significant impact on model results: forcing the use of 10x thinner grid layers (run 3), computed melt amounts drop by about 30 %. (Mean subsurface heat flux also has a five-fold increase, but the value remains reasonably low – unlike what is reported for COSIPY by Brun et al.). The maximum depth of simulation also affects the results somewhat (runs 3, 4, 5): annual melt amounts increase for shallower grids, from 0.52 m w.e. (at 20 m maximum depth), to 0.55 and 0.59 (respectively at 5 m and 50 cm). Unlike the COSIPY result by Brun et al., a finer time resolution of 1 minute (with linearly interpolated climate variables) does not reduce melt at all in the EBFM (runs 6 and 7).

Most importantly, in run 8 we test a standard value of 0.60 for the albedo of blue ice. This simple change reduces melt rates by 90-95 % compared to the glacier-ice default value of 0.39 (run 5). A further reduction by more than 60 % occurs when running the model after a spin-up period of 50 years (run 10). The latter observation also holds true for simulations with the default (lower) albedo values (run 12).

Such a high sensitivity to the albedo parameters indicates a very high degree of uncertainty in the simulated energy balance at South Col, and raises serious concerns on the applicability of any albedo values not measured in situ.

Computed sublimation rates in the EBFM are in all cases comparable (20-50 cm w.e. yr-1 ) to the results of Potocki et al. and Brun et al. Still, parameters involved in the calculation of turbulent fluxes (such as surface roughness lengths) are known to be poorly constrained, especially in high accumulation areas (e.g. Mattea et al., 2021). Therefore, if sublimation plays a major role in the surface mass fluxes at South Col, its modeling uncertainties are likely also significant for the overall error budget, and should be investigated.

In conclusion, also a skin-layer, or skin-temperature model appears to be able to predict, in its basic configuration, no melt for the South Col Glacier. Relatively small perturbations in model parameters, however, are sufficient to change model output substantially, reaching from almost zero up to ~50% of the melt simulated by Potocki et al.

**[ARC4-4]** We sincerely thank Horst Machguth and Enrico Mattea for performing new simulations using EBFM at South Col glacier, with the same forcing data as Potocki et al. (2022) or Brun et al. (2022), and testing different numerical setup or model parameters. Their results in table 1 nicely illustrate that model outputs (mainly melt) are highly sensitive to the model numerical setup or to key parameters such as the albedo. We totally agree with their conclusions and the concerns they raise on the applicability of any mass-energy balance model in this extreme environment. We indeed reach the same conclusions with the COSIPY model (see ARC1-3 for a complete reply). As stated in ARC4-3, we did not perform any parameter sensitivity study in our communication. Since the model is not suitable in such an environment, this is finally meaningless. But we expect to have a large sensitivity to albedo, and to a lesser extent to roughness lengths, as shown for EBFM. We added a sentence about the sensitivity to ice albedo value:

"We also rise some awareness about the parametrization of albedo, and the fact that the blue ice of South Col Glacier might have an albedo larger than 0.4, as observed for Antarctic blue ice that has an albedo of 0.5 to 0.6 (e.g., Smedley et al., 2020). A higher ice albedo would dramatically reduce the melt totals, as suggested by the sensitivity tests of Potocki et al. (2022). and Machguth and Mattea (2022)."

4. Detailed comments on the manuscript by Brun et al.
**[RC4-5]** Line 26: We suggest spelling out JJAS where it is first mentioned.
**[ARC4-5]** Done

**[RC4-6]** Line 36: This communication is not brief. Depending on the editorial guidelines, it could also be published as a normal paper.
**[ARC4-6]** If the Editor agrees, we prefer to keep this contribution as a brief communication, because this study has been mainly conducted to reply to Potocki et al. (2022) and consequently, it would lose its interest as a normal paper. Moreover the lack of in-situ data prevents from performing a reliable analysis of mass-energy balance, which would be valuable in a normal paper. To keep this communication short, we have removed or shortened some sections: all Crocus simulations and all text referring to glacier flow and emergence velocities.

**[RC4-7]** Lines 54-57: Could you quantify "as the range of elevation change values were higher here"? How are "minor data voids" defined?

**[ARC4-7]** The standard deviation of dH values above 6800 m is higher than below 6800 m due to the presence of high magnitude (±~ 35 m) elevation differences associated with previously described [ARC1-2] crevasse block movement on the Lhotse and Kangshung face.

To fill small data voids we computed a smoothed version of the dH grid where the value of each cell was derived as the mean of a surrounding 5 x 5 cell (10 x 10 m) window. Data voids in the original dH grid smaller than this window size could then be filled with the 'mean' values.

We have slightly reworked the text in Section 2 of the manuscript to better describe these parts of the dH data processing:

"Following DEM differencing, surface elevation change data (dH) were filtered to remove outliers, with values outside the range of five times the standard deviation of dH estimates within 50 m elevation bands removed below 6800 m a.s.l. Above 6800 m, or from the base of the much steeper Lhotse face, we applied a threshold of three times the standard deviation of dH estimates, as the range of elevation change values here include high magnitude elevation changes (± ~35 m) associated with crevasse field evolution captured by both DEMs. To fill small data voids we computed a smoothed version of the dH grid where the value of each cell was derived as the mean of a surrounding 5 x 5 cell (10 x 10 m) window. Data voids in the original dH grid smaller than this window size could then be filled with the 'mean' values."

Note that these steps are mostly relevant for glaciers other than SCG, as no pixel was excluded from SCG, whereas on average of 7.2 % of the pixel were excluded for the other glaciers, due to their steeper slopes and more variable dH.

**[RC4-8]** Lines 156/157: The article by Brun et al. criticises most results from Potocki et al. However, here one of their results is cited as if correct, to support the argumentation by Brun et al. For the purpose of assessing the study, we suggest, to argue based on independent results also where the results from Potocki et al. fit the own argumentation.
**[ARC4-8]** Agreed. This sentence has been removed

**[RC4-9]** Line 177: Citation: It appears that this is basic knowledge generally understood. We would like to ask the authors to cite a more original reference.
**[ARC4-9]** Tubini et al. (2021) has been replaced by Anderson (1976)

**[RC4-10]** Lines 262/263: Please add at least one more original citation or remove the citation here. It is long known how ice dynamics transport ice from the accumulation to the ablation area.
**[ARC4-10]** The citation has been removed

**[RC4-11]** Lines 278-289: We do not fully understand the idea behind this calculation. The authors first state that the glacier is in balance, as shown by the DEM differencing. Then the ablation values of Potocki et al. are used to estimate at which speed the ice would need to flow so both conditions are fulfilled, that is (i) glacier is in equilibrium and (ii) ablation is ~2 m a-1. But Potocki et al. do not claim that the glacier is in equilibrium, hence we do not understand what is supposed to be shown here.

**[ARC4-11]** The idea here was to exclude the possibility that the 1.5 m w.e./yr melt suggested in Potocki et al. (2022) is compatible with a mean dh=0, i.e. the melt would be compensated by emergence. Anyway this section has been removed (see ARC1-4 for a complete reply)

**[RC4-12]** Line 290-293: The argumentation could be clearer. The motivation of the previous paragraph is already unclear to us (see above). Then, without an introduction or explanation, another method, based on other assumptions, is used to estimate flow velocity in South Col glacier.
**[ARC4-12]** The section has been removed - see ARC4-11 and ARC1-4 for a complete reply)

**[RC4-13]** Line 293: Is the term "continental" the right term here? It is a monsoon influenced glacier, likely cold and frozen to the bed. While indeed arid (albeit not only because of low precipitation but also because of strong wind erosion), annual fluctuations in air temperature, characteristic for continental glaciers, are not particularly large (e.g. Suppl. Fig. 8 in Potocki et al.).
**[ARC4-13]** Agreed. The entire section has been removed and the term continental abandoned.

**[RC4-14]** Figures 1 and A6: Mayewski et al. state in their comment on Brun et al. that the latter placed the drill site at the wrong location. It appears to us that the drill location as visualized by Brun et al. is correct, the only difference being that Brun et al. express elevation in meters above the ellipsoid while Potocki et al. use elevation in meter above the geoid. Nevertheless, it appears that in Fig. 1 the contour lines are in meters above the ellipsoid (the drill site is slightly below 8000 m) while the elevation of Lhotse peak is given in meters above the geoid. The caption of the figure does not indicate which elevation datum was used. As readers might be familiar with elevations of South Col, Lhotse Peak and Everest, and these well-known numbers are in meters above sea level, we suggest that elevations on maps are expressed in relation to the same datum.

**[ARC4-14]** We appreciate the reviewer highlighting this point relating to the use of geoid heights versus ellipsoid heights, which is an oversight on our part. We confirm that our DEMs were generated to represent the height above the ellipsoid rather than the height above geoid, hence the difference in elevation at the ice core drill site. We have corrected the Pleiades DEM using the EGM2008 geoid model and replotted the contours used in Fig. 1 to ensure consistency with other landmark heights in the area. At the point of the drill site (27.977211, 86.929861, taken from Potocki et al. 2022), the Pleiades DEM estimates the elevation to be 8003 m. Note that there is a +/- 10 m uncertainty in the absolute elevation from Pléiades DEM, which does not affect the DEM difference. We have altered the Figure caption to state that indicated elevations are height above the geoid.

[Figure]

**Figure 1.** Surface elevation change over the Western Cwm (a) between 1984 and 2017, and over the South Col Glacier (b). The location of the ice core and AWS from Potocki et al. (2022) are shown with blue dots. Background is a shaded relief from the Pléiades DEM. The conditions at the surface of the South Col Glacier on 23 March 2017 are captured by a Pléiades orthoimage in panel c (Pléiades, copyright CNES 2017, Distribution Airbus DS). The inset of panel a shows the location of Mount Everest in the broader context of High Mountain Asia. All indicated elevations are expressed as height above the geoid.

Acknowledgements
We thank Tim Mosedale (https://timmosedale.co.uk/) and the Swiss Foundation for Alpine Research (https://alpinfo.ch) for granting permission to use their photos of South Col Glacier in the context of this review.

References
Genthon, C., Lardeux, P. & Krinner, G. 2007. The surface accumulation and ablation of a coastal blue- ice area near Cap Prudhomme, Terre Adélie, Antarctica. Journal of Glaciology, 53, 635–645.

Greuell, W. & Konzelmann, T. 1994. Numerical modelling of the energy balance and the englacial temperature of the Greenland ice sheet. Calculations for the ETH-Camp location (West Greenland, 1155 m a.s.l.). Global Planet. Change, 9, 91–114.

Klok, E.J. & Oerlemans, J. 2002. Model study of the spatial distribution of the energy and mass balance of Morteratschgletscher, Switzerland. J. Glaciol., 48, 505–518.

Kronenberg, M., van Pelt, W., Machguth, H., Fiddes, J., Hoelzle, M. & Pertziger, F. 2022. Long-term firn and mass balance modelling for Abramov glacier, Pamir Alay. The Cryosphere Discussion.

Mattea, E., Machguth, H., Kronenberg, M., van Pelt, W., Bassi, M. & Hoelzle, M. 2021. Firn changes at Colle Gnifetti revealed with a high-resolution process-based physical model approach. The Cryosphere, 15, 3181–3205.

Matthews, T., Perry, L.B., Koch, I., Aryal, D., Khadka, A., Shrestha, D., Abernathy, K., et al. 2020. Going to extremes: installing the world's highest weather stations on Mount Everest. Bulletin of the American Meteorological Society, 101, E1870–E1890.

Potocki, M., Mayewski, P.A., Matthews, T., Perry, L.B., Schwikowski, M., Tait, A.M., Korotkikh, E., et al. 2022. Mt. Everest's highest glacier is a sentinel for accelerating ice loss. npj Climate and Atmospheric Science, 5.

Reijmer, C.H., Bintanja, R. & Greuell, W. 2001. Surface albedo measurements over snow and blue ice in thematic mapper bands 2 and 4 in Dronning Maud Land, Antarctica. Journal of Geophysical Research: Atmospheres, 106, 9661–9672.

Smedley, A.R.D., Evatt, G.W., Mallinson, A. & Harvey, E. 2020. Solar radiative transfer in Antarctic blue ice: spectral considerations, subsurface enhancement, inclusions, and meteorites. The Cryosphere, 14, 789–809.

van Pelt, W.J.J., Oerlemans, J., Reijmer, C.H., Pohjola, V.A., Pettersson, R. & van Angelen, J.H. 2012. Simulating melt, runoff and refreezing on Nordenskiöldbreen, Svalbard, using a coupled snow and energy balance model. The Cryosphere, 6, 641–659.

**Reply to community comments**

The following pages contain a point-by-point reply to the comments provided by the community.
Each of the community's comment (CC) is numbered. If a comment contained several points, we numbered them, and addressed them individually in our author replies (ACC).

**Community comment 1**

Response to Brun et al. re Potocki et al. (2022)*
* Mariusz Potocki1, Paul Andrew Mayewski1, Tom Matthews2, L. Baker Perry3, Margit Schwikowski4, Alexander M. Tait5, Elena Korotkikh1, Heather Cliffiord1, Shichang Kang6,7, Tenzing Chogyal Sherpa8, Praveen Kumar Singh9, Inka Koch10, and Sean Birkel1 with additional input from Song Shu3
1Climate Change Institute, University of Maine, Orono, ME, USA.
2Department of Geography, King's College London, London, UK.
3Department of Geography and Planning, Appalachian State University, Boone, NC, USA.
4Laboratory of Environmental Chemistry, Paul Scherrer Institut, Villigen, Switzerland.
5National Geographic Society, 1145 17th St., Washington, D.C., USA.
6State Key Laboratory of Cryospheric Sciences, Northwest Institute of Eco-Environment and Resources, Chinese Academy of Sciences (CAS), Lanzhou, China.
7University of CAS, Beijing, China.
8International Centre for Integrated Mountain Development, Kathmandu, Nepal. 9Centre of Excellence in Disaster Mitigation and Management, Indian Institute of
Technology Roorkee, Roorkee, Uttarakhand, India.
10Department of Geosciences, University of Tübingen, Tübingen, Germany.
Correspondence: mariusz.potocki@maine.edu; paul.mayewski@maine.edu;
tom.matthews@kcl.ac.uk

[CC1-1] In our paper (Potocki et al., 2022 – hereafter 'P22') there were two major findings: (1) that extremely rapid ice loss is possible once a protective snowpack is ablated away; and (2) this appears to have happened at the South Col Glacier (SCG) as evidenced by the presence of surface ice on the SCG dated at ~2000 years ago, indicating the loss of a significant portion of effectively what we consider to be at least currently a "stagnating glacier".
Brun et al. (hereafter B22) challenge both findings of P22. We welcome their paper and agree that more research, despite the extreme conditions involved in undertaking this research, is needed. Notably additional ice coring, mass balance stakes and an ice radar survey to more fully decipher ice dynamics and mass balance. However, we question the evidence used by B22 to underpin their conclusions. Our queries are outlined below – dealing first with finding (1) of P22 ("could the SCG thin so rapidly") and then finding (2) ("did the SCG recently thin"). Note that (1) is related to the surface mass/energy balance modelling of B22; (2) is concerned mostly with their DEM analysis. Accordingly, we organise our comment under the headings: "Energy Balance Modelling" and "DEM Analysis". Note that there is naturally some overlap between these sections.

**[ACC1-1]** We appreciate the detailed comment and discussion offered by Paul Andrew Mayewski. We agree with the description offered here that the Potocki et al. (2022) put forward two main findings, that are both challenged based on (i) our analysis of the COSIPY's implementation and (ii) our DEM analysis.

**[CC1-2]** Energy Balance Modelling
B22 conclude that substantial ice melt may not be physically plausible (even under extreme insolation). That is, they challenge the evidence that the SCG could thin at the rate proposed by P22. Their reasoning is that P22's conclusion is not robust to all modelling assumptions. Notably, if the conductive heat flux is calculated by the COSIPY model using a finer resolution near the surface (the 'grad' experiment), or using a different model (CROCUS) substantial ice ablation cannot occur even if a snowpack is removed. This is a very interesting result, but is it physically plausible? Can such high conductive heat fluxes be maintained if the sub-surface was warming so much? Might increasing the resolution only near the surface (and not at all depths) create a 'cold' bias? Also, the temperature profile used to initialize COSIPY might be inappropriate for the B22 experiment: it was the outcome of a spin-up under the P22 setup -- i.e., with a much-reduced conductive heat flux. If spun up with the P22 grad method, the sub-surface temperature profile would very likely be different. In other words, there's a physical inconsistency between the spin up and the B22 experiments.
In the context of the above, we (P22) note here that both the COSIPY grad and CROCUS model variants are completely untested in an environment like the SCG. Indeed, CROCUS, as B22 explains, is primarily a snowpack model. By contrast, P22 used the 'default' COSIPY model code (i.e., with the conductive heat flux computed using a less-fine resolution near the surface). The setup used in P22 has been tested and shown to perform well against observations, including on Zhadang Glacier (at >5,600 m) on the Tibetan Plateau (a cold and dry environment not too dissimilar from the SCG). Note that the agreement between COSIPY grad and CROCUS and divergence from COSIPY P22 noted by P22 may just highlight a shared weakness of those model configurations; it is not reassurance of their physical realism. It would also be very helpful if B22 explained which (if any) other COSIPY parameters were changed in their study, and if so, how results varied across those ensemble runs. That is, even if their model variants produce physically plausible results, B22 will only have demonstrated that the P22 conclusions are not robust to all (reasonable) modelling assumptions, but no real insight into just how non-robust they are. This is particularly important because P22 performed an uncertainty assessment, perturbing model parameters across a broad range of plausible values. P22's conclusion about the plausibility of substantial ice melt was robust across all scenarios considered.
Taken together then, we caution that B22 give the impression that their conclusions about the plausibility of substantial ice loss should be given as much weighting as those of P22. However, without further work by the B22 authors, this is not the case: (i) the ability of their models to capture the surface energetics at the SCG remains to be determined; and (ii) their uncertainty assessment is too narrow to give an adequate perspective on the robustness of P22's conclusions.

**[ACC1-2]** We think the goal of our modeling has been somewhat misunderstood. As detailed in the response to the other reviewers (ARC1-3 and ARC4-3), our goal is not to produce a better estimation for the melting of SCG, but to highlight a structural limitation of COSIPY that impedes its current application to SCG.

The difference between the COSIPY_grad and the original COSIPY configuration is not that the spatial resolution was refined near the surface. Rather, we modified COSIPY so that the computation of the sub-surface heat flux (abusively called ground heat flux in the COSIPY model description) used for the surface energy balance is performed as close as possible to the surface. This choice is not arbitrary as it follows from physics (e.g. Eq 4 of Sauter et al., 2020) and is the standard implementation in other skin-layer models (e.g. Covi et al., 2022). Note also the computation of the sub-surface heat flux appears as well for the computation of the temperature evolution of the first sub-surface layer (as a source of energy entering the ice from the surface). The COSIPY_grad version is thus internally consistent from this point of view.

Since (i) the default choice of COSIPY to compute the ground heat flux for the surface energy balance using the temperature in the first 10cm is not imposed by physics and (ii) that this choice has a large influence on surface melting, we conclude that no firm conclusions can currently be drawn from the use of COSIPY at SCG. Note that the sensitivity to the model structure and implementation is not specific to COSIPY, as shown by Machguth and Mattea [RC4-4].

We agree that the inclusion of Crocus simulations in our study blurred the message by giving the impression that the agreement between Crocus and COSIPY_grad should be understood as a proof of their correctness. We have removed all references to Crocus in the new version of our study and clarified the aim of the surface mass balance modeling section.

**[CC1-3]**2. DEM Analysis
B22 argue that, based on the difference in DEMs constructed for 1984 and 2017, the SCG has not thinned. We welcome this analysis, but have several observations that challenge their findings and overall interpretations:
**[CC1-3a]** (1) We note that B22's Figure 1 has our ice core at the wrong elevation. The ice core was not collected below 8000m but rather at 8020 m, which would place it within B22's so-called 'accumulation' zone.
**[ACC1-3a]** Please refer to ARC4-6 regarding the elevation of the ice core in Figure 1.

**[CC1-3b]** In addition, it is not clear why in their Figure A6, B22 define a particular line as the transition between accumulation and ablation based on a single day's image, implying this as the equilibrium line. Such a differentiation would be very sensitive to the timing of image acquisition and should instead be based on a much longer record of mass balance.
**[ACC1-3b]** Please refer to ARC1-4 regarding Fig A6 and the delineation of ablation and accumulation zones

**[CC1-3c]** As briefly acknowledged as a possibility (around L300 of B22), the steep snow slope designated as an accumulation area on Figure A6 is likely comprised of avalanche material (as evidenced below by the tongues of avalanched material) and is, therefore, not a standard snow accumulation region. The SCG surface downslope from the avalanche tongues is clearly exposed ice with patches of seasonal snow cover. Whether or not the SCG currently even has an accumulation area (at best very small) there is ice core and modeling evidence for current thinning/ablation up to at least the elevation (band) of 8020 m, which would be dominated by ablation indicating that the SCG currently has a negative mass balance. At lower elevation

bands (than the ice core location) surface ablation/thinning rates of SCG is likely even stronger. The presence of clear banding and the identification of "annual layers" in the SCG ice core suggest that avalanching has not been the accumulation source for the SCG in the past, therefore something has happened, notably the transition to a stagnating glacier with its upper reaches comprised of avalanched snow.

**[ACC1-3b]** Our study indeed clearly challenges the statement that SCG currently has negative mass balances (since 1984) and Reviewer 4 provided additional evidence that there was no thinning of SCG since 1956 (see RC4.2). Please refer to ARC1-2 for the accuracy of our DEM differencing analysis. Our mass-energy balance modeling tests with different numerical implementations also suggest that there is no clear modeling evidence of large melt at SCG (see ARC1-3). We do not completely understand why the presence of banding would be an argument against avalanche dominated accumulation, because the deposition area of avalanches usually spreads out over a horizontal area, but we agree that the presence of exposed ice shows that parts of SCG are dominated by ablation processes. As the glacier has not been thinning for the last thirty to sixty years, it implies that ablation is fully balanced by ice fluxes in its current state. Given that the glacier is small and cold, ice fluxes have to be very small, and thus ablation is likely equally very small.

However, our DEM difference covers only for the most recent period, and thinning might very well have occurred between before 1984 (or 1956 according to RC4-2). We clarified this point in the revised manuscript (section 4): "Note that our present study focuses on the period 1984-2017 (or 1956-2022; Machguth and Mattea, 2022) where no thinning is observed, but we cannot exclude any thickening and/or thinning episodes anterior to this period, potentially explaining why Potocki et al. (2022) observed ice as old as 1966 years at the surface of their ice core."

[Figure]

**[CC1-4a]** (2) B22's Figure 1 shows that there are regions both above and below 8000 masl on the north side of the SCG that reveal thinning and thickening up to 30m for the period 1984-2017. This seems at odds with B22's statements that "... the distribution of dH on South Col Glacier is rather homogeneous and not different from the distribution of dH over ice-free areas or glacierized areas located within the same elevation range." Indeed, according to B22's Figure 1, dH over SCG is actually *at odds* with the *highly variable thickness change* over all other glacierized terrain at similar elevations. B22's Figure A1 obscures this by averaging over very large gains and losses to show a mean dH close to zero (minor comment -- what is the uncertainty shading supposed to represent in the right-hand panel of Fig. A1?). In addition, it seems unlikely that the south-facing SCG would experience little to no ablation while large parts of the north-facing Rongbuk Glacier would experience significant (up to 30m) ablation as indicated in Figure A1. Indeed, the upper branch of the Khumbu – only ~250 m from the SCG and with a similar (SW) aspect – did thin by tens of meters according to Figure 1.

[ACC1-4a] As noted in [ARC1-2], we believe that the points raised in [CC1-4a] relating to the patterns of elevation change evident in the Western Cwm amount to misinterpretation of the dH grid we present in Fig.1:

- CC1-4a states 'there are regions both above and below 8000 masl on the north side of the SCG that reveal thinning and thickening up to 30 m for the period 1984-2017'. This pattern of elevation change is also present to the southeast and southwest of the South Col Glacier and is related to the advection of seracs and crevasses (which may have opened or closed over the study period) down slope by the flow of the Khumbu and Kangshung Glaciers. This pattern was also acknowledged by reviewer 1 (Ann Rowan) and further discussed in ARC1-2. Resp. Fig. 1 (ARC1-2) provides an example of the evolution of the crevasse field on the Lhotse face above Khumbu Glacier and the associated elevation differences caused by this process.

- The authors of CC1-4a state 'B22's Figure A1 obscures this by averaging over very large gains and losses to show a mean dH close to zero'. This comment is not applicable to panel A of Figure A1, which shows the distribution of *all* dH values (not average values) over the surface of the South Col Glacier and over stable, off-glacier terrain, between which there is very little difference. Panel B of Figure A1 provides an illustration of the variance of the dH data through the elevation range covered by the DEMs in the form of the Normalised Median Absolute Deviation (NMAD). This plot indeed shows that NMAD is highest between ~7200-7800 m, where elevation change patterns associated with advection of seracs and crevasses are common. The magnitude of NMAD over the elevation range covered by South Col Glacier is much lower (1.92 m), indicating elevation change of lower variance is predominant at this height. Finally, it must be noted that the averaging of dh values over elevation bands is not used to "obscure" any relevant signal as suggested by this comment, but in order to analyze the elevation changes in a way that is not affected by ice flow.

- The authors of CC1-4a suggest that 'it seems unlikely that the south-facing SCG would experience little to no ablation while large parts of the north-facing Rongbuk Glacier would experience significant (up to 30m) ablation as indicated in Figure A1'. We want to clearly state that our dH data do *not* cover the Rongbuk Glacier, which is on the northern flank of Mt. Everest, ~4 km from the South Col Glacier. The area of the dH grid highlighted here covers the upper reaches of the Kangshung face on the eastern side of Mt. Everest, which again is characterised by a large number of seracs and crevasses blocks, the movement of which has been captured in our pair of DEMs. The ~30 m elevation differences here are likely related to this ice flow, rather than a surface mass balance process.

We recognize that the initial version of this manuscript did not describe these patterns in a way that can be understood by a wider audience and therefore made several changes to section 2 to improve this, as summarized in ARC1-2.

**[CC1-4b]** We agree with B22's multiple mentions of the complications inherent with imagery interpretation for SCG and other high elevation regions and suggest that these might be hindering their Figure 1 results. We understand that The Pléiades DEM was generated from military-level satellite stereo images (0.5 m ground resolution). The only question is the accuracy of 1984 DEM which is not mentioned by B22. Our understanding is that the DEM was generated from images collected by an airplane at 10,000 feet (3048 m) above the top of Mount Everest according to the article from which the 1984 images were obtained. An aircraft

under these conditions inevitably has vibrations induced by airflows. Even though the Wild RC-10 camera was aimed straight down, the vibration of the airplane would always introduce a slight angle to the camera's nadir looking direction. We do not know how accurately the nadir looking direction was maintained since no information about the image acquisition was provided. Assuming only a 0.5 degree departure from the nadir direction, then the error of location on ground is over 26 m (tan(0.5) × 3048 = 26.6 m). For Mount Everest, the 26.6 m of displacement could result in a several meter-level error in elevation due to the drastic change of topography within a short distance.

**[ACC1-4b]** Whilst we appreciate the concerns raised in CC1-4b regarding the accuracy of the derived DEMs, we find the comments aimed at the 1984 imagery somewhat unjustified.

The accuracy of the 1984 DEM is primarily dependent on the source of the Ground Control Points (GCPs) used to fix the location of the trio of images we used during DEM extraction. Here we use the Pléiades DEM and orthoimagery as the source of GCPs for the 1984 DEM given the lack of suitable field-based GCPs, for obvious logistical reasons. Response Figure 3 (below) shows the distribution of tie points (used to relate features visible in overlapping images to one another), 90 of which were placed over the overlapping area of the three aerial photographs, and GCPs used in the generation of the 1984 DEM (10 of which were used to relate the overlapping images to ground coordinates). The 1984 aerial photography and Pléiades imagery are of comparable resolution (0.5 m) and so the same features used as GCPs were easily identifiable in each set of images.

[Figure]

**Resp.Fig.3.** Location of Tie Points and Ground Control Points used during the extraction of the 1984 DEM from three overlapping aerial photographs. Ground Control Points were identified over features which we could confidently assume did not move between the dates of the 1984 aerial photograph acquisition and the 2017 Pléiades scene (i.e. off-glacier terrain).

Photogrammetric software such as PCI Geomatica (now CATALYST) considers the user-provided tie points and ground control points in a 'bundle adjustment', which calculates the position and orientation of the camera and the 3D position of points across each scene. The residuals from this process (Response Table 1) provide a summary of the agreement between the calculated positioning of points in the images and the provided ground control data. The RMS of the GCP residuals (in x, y and z-direction) are all low, which shows that our processing has closely matched the position of features identified in both the 1984 imagery and the reference Pléiades scene. These errors are significantly lower than those hypothesised in CC1-4b.

Res. Table 1. Residuals (in metres) of tie points and ground control points used during the 1984 DEM extraction process.

| | RMS (X, Y, Z) | X Bias (StDev) | Y Bias (StDev) | Z Bias (StDev) |
|---|---|---|---|---|
| **Tie Points (90)** | 0.410, 0.397, 0.290 | -0.002 (0.411) | 0.002 (0.398) | 0.003 (0.291) |
| **Ground Control Points (10)** | 0.763, 0.941, 1.927 | 0.099 (0.757) | 0.054 (0.939) | 0.357 (1.893) |

Still, as an additional step to ensure the precise location of the 1984 DEM in relation to the Pléiades DEM, we undertook a subsequent coregistration procedure (described in the manuscript) to remove any remnant minor shifts. The magnitude of these shifts were calculated using off-glacier (stable) terrain dH statistics (see Nuth and Kääb, 2011) and were all less than 2m. This is a common procedure involved in geodetic glacier mass loss studies.

Finally, the reconstruction of surface topography through photogrammetry actually benefits from variations in stereo angle (i.e. departure from nadir pointing), where more than one 'perspective' of a feature can improve its reconstruction in a DEM. For instance, the Pléiades satellites can acquire three images in sequence (forward, near-nadir and backward) with variable stereo angles (sometimes >10°) which in combination provide superior coverage of DEMs over glacier surfaces (Deschamps-Berger et al., 2020).

Deschamps-Berger, C., Gascoin, S., Berthier, E., Deems, J., Gutmann, E., Dehecq, A., Shean, D., and Dumont, M.: Snow depth mapping from stereo satellite imagery in

mountainous terrain: evaluation using airborne laser-scanning data, The Cryosphere, 14, 2925–2940, https://doi.org/10.5194/tc-14-2925-2020, 2020.

**[CC1-5]** (3) We find the information provided in B22's Figure A6 a bit perplexing. In particular, as stated by B22. ... "the shaded hashed area represents glacierized area that might belong to SCG, but it is not possible to conclude solely from satellite imagery." Might this not be part of the areal loss assumed in P22 over the last three decades? Clearly the image recognition for this region is still in question.

**[ACC1-5]** Figure A6 has been removed (see ARC1-4) and Reviewer 4 showed there was no frontal retreat since 1956 (RC4.2). We agree that our delineation of South Col Glacier remains disputable, especially in the southern edge of the glacier where the Pléiades imagery does not allow to clearly identify a glacier front. However, we remain confident that our outline of South Col Glacier covers most parts of the glacier.

**[CC1-6]** (4) B22 present an interesting display of seasonal snow variability (their Figure 2). This is used to argue for it being unlikely that the SCG is snow-free during the monsoon, although they do not in fact include any images from the critical months of May and June, before arrival of the monsoon. In turn, they reason that this helps explain their conclusion that the SCG has not recently thinned, because P22 required snow-free conditions during the monsoon to drive the widespread melting needed to ablate the SCG. We certainly agree with B22, that the SCG is covered by snow during the latter portions of the monsoon in August and September, but question their assertion that snow cover is present in May through early to mid-July when insolation is at its annual maxim. Albedo data from the South Col AWS confirm a largely snow-free surface from November 2019 through mid-July 2020 (Bessin et al. 2021). In addition, the three years considered by B22 may not be representative of longer term conditions. A similar view of seasonal variability during several earlier image periods would have been interesting to include. To this end, we note the availability of twice-daily images from Mt Everest's Basecamp in Nepal (see below) should help shed ever more light on this issue, given that changes in snow-cover above 8000 masl are clearly visible. Details of these photographs are in Grey et al. (2022).

[Figure]

**[ACC1-6]** This is strange that P. Mayewski and others claim that Brun et al. (2022) do not include any Venµs images in May and June although in Figure 2, there are images from 27 April 2019, 25 May 2019, 14 June 2019 and 2 July 2019. The whole image dataset (267 images in total, including images in May and June) from 27 November 2017 to 30 October 2020 is available to everybody in a Zenodo repository (Brun, 2022, cited line 89 of the original MS, and referenced line 462 of the original MS). We thus invite the authors of CC1-6 to look at all images available on-line.

Actually, Potocki et al. (2022) try to explain their ~55 m thinning over the last ~25 years with a change in surface state, from snow/firn covered surface to ice-exposed surface, implying a step change in melting due to the enhanced absorption of incoming solar radiation. Independently from the fact that the melt rates simulated by Potocki et al. (2022) over ice exposed surface are questionable (see ARC1-4 for the debate concerning the sensitivity of COSIPY to different numerical implementations), the series of Venµs images (Brun, 2022) clearly shows that SCG is entirely or partly snow-covered during long periods of the year, including periods when solar radiation is not maximum but still intense (the second half of the monsoon). P22 simulate annual melt rates that are on average 653 mm w.e. for the August and September months for the period 1950-2019 (**Resp. Fig. 4**). On average for the whole period, these two months contribute to 43% of the annual melt in P22. We agree that Venµs satellite has been operating only for 3 years (2017-2020) and this period might not be representative of longer term conditions, but we observe the same pattern of snow seasonality during these three years. Additionally, Sentinel-2 (2016 - present; https://drive.google.com/file/d/1ygZhRU5cHFK4CildxRojMNUrxGiYYIJB/view) and Landsat 5 images (1988 - 2011; https://drive.google.com/file/d/14EdWEJXWNKA2BlyXg0xmSqXIsFDK1n3k/view1) show the same temporal pattern for an extended period, and suggest that July is also snow covered most of the time. The Venµs images additionally show that P22 cosipy simulations do not reproduce the seasonal cycle of snow presence.

[Figure]

**Resp.Fig.4.**: monthly melt rates (mm w.e.) for the ice simulation of P22 for the period 1950-2020

We do not see how twice-daily images taken from Everest Base Camp in 2021 can shed light of the long-term seasonality of the snow-cover over SCG because the glacier is not visible from base camp, is not comparable with the southern steep and rocky flanks of Mt Everest circled in red in the Figure shown in CC1-6, and the dataset is even shorter than that of Venµs images.

**[CC1-7a]** A related issue with B22's assessment of seasonal snow cover is their deployment of an empirical wind redistribution model. Several limitations of it are mentioned, but the most significant – and arguably so great that it should rule out its inclusion -- is that the SCG environment is so different from the (ice sheet) environment it has been tried and tested in. First, the SCG is in a very complex topographic setting and likely subject to very high small-scale wind variability due to shear-induced turbulence. This matters acutely when considering wind redistribution because maximum gusts set the upper limit on erosion potential. Second, the atmospheric pressure at the SCG is approximately one-third that of sea level. Even if air density is a parameter within their model, what evidence do they have that an empirical scheme developed and applied at much lower elevations/higher pressure behaves realistically in such a different environment? Taking the B22's own words (~L250 in the context of mass balance modelling) "[models]... developed and tested in specific conditions, ...[should not] be applied directly to other conditions, such as the very specific conditions of South Col glacier, without extensive validation."

**[ACC1-7a]** Here we agree that the sentence quoted by CC17-a regarding the applicability of models was insensitive and we rephrased it in the revised manuscript as follows: "We thus highlight here that COSIPY does not appear to be suitable to model mass balance in the specific conditions of South Col glacier without extensive validation". However, we feel that the rest of the comment is a bit unfair, regarding the fact that model development is often an uncertain, but necessary, step towards a better understanding of processes, especially in the absence of field observations. The model presented in our manuscript is grounded on physics and we never claimed it to be accurate. We just think that it provides additional evidence, together with Venus images, that wind erosion is a non-negligible process. It stresses the fact that modeling surface mass balance without explicitly accounting for wind erosion, as done in any COSIPY simulation, will necessarily lead to results that are questionable. We modified the title of section 3.2, which now reads "The potential of wind erosion".

**[CC1-7b]** The point about the monsoon possibly being a time when the SCG is snow-covered seems to be made most strongly with the Venus images and basic physical reasoning about the wind speeds and precipitation occurrence. The empirical model is so uncertain that we argue it detracts rather than adds to this argument. Perhaps instead B22 would consider replacing this with modelling of the SCG surface mass during the monsoon. Forcing their COSIPY and CROCUS model variants with the P22 precipitation (which they inhibit in their ice model runs) should give more useful insight into the extent of snow cover during this critical time of the year. All lines of inquiry (Venus images, physical reasoning, and their empirical model) already identify the monsoon as a period of minimal wind deflation, so a more important question is to what extent monsoonal snowfall is melted or sublimated away – not least because all models (and both studies) agree on the very high importance of the latter.

**[ACC1-7a]** None of the current surface energy balance models (COSIPY in the configuration grad or P22 and Crocus) is able to represent the observed dynamics of snow cover during monsoon for South Col Glacier. There are many reasons behind this, but two important ones could be the large uncertainty on precipitation (see also **ACC1-7c** below) and the albedo parameterization. We do not discuss this in detail here, but it is clear in all simulations that sublimation is not playing a major role during monsoon, and especially during July-August, because of the very high relative humidity and low wind speed. Res. Fig. 4 shows the monthly values of sublimation for the snow and ice scenario of P22, which reaches a minimum during July-August.

[Figure]

**Resp.Fig.5**: monthly sublimation rates (mm w.e.) for the ice and snow simulations of P22 for the period 1950-2020

**[CC1-7c]** In the context of the above, B22 assert (around L165) that P22's estimates of precipitation are highly uncertain because they tune them to match ablation over an arbitrary period. We agree that they are uncertain, but note that the long-period of integration (10 years) protects somewhat against sampling variability. We also note that P22's precipitation estimate (mean of 191 mm a-1) was an order of magnitude greater than suggested by previous work (Salerno et al., 2015). There are reasonable explanations for that (including that the latter used poorly-shielded instrumentation to measure precipitation, hence a high risk of under-catch); but if B22 include our suggestion to model the SCG mass balance during the monsoon, we suggest that they keep this in mind: P22's precipitation estimates are unlikely to be biased low.

**[ACC1-7c]** We completely agree that precipitation estimates are extremely uncertain, however we would like to stress that the precipitation gradient is extrapolated over more than 3000 m of elevation difference in P22, which is very large given the variability of precipitation gradients in mountainous environments. Additionally, we want to caution about the precipitation calibration strategy applied in P22. First, as shown in B22, the surface mass balance model of P22 is very uncertain. Precipitation estimates calibrated with this model thus inherit these uncertainties. In P22, the annual precipitation estimate is balanced by annual sublimation, which is very sensitive to the choice of roughness length. Second, this calibration strategy does not account for wind erosion, which is likely a non-negligible phenomenon. Third, the tuned precipitation estimate is very sensitive to the modelled surface mass balance, and thus to the state of the glacier surface. In particular, it seems that the South Col Glacier

was already presenting an ice surface during the 1953 British Expedition to Everest. This is shown by Alfred Gregory's photographs, which we cannot reproduce here for copyright reasons, but that are available in the book *Alfred Gregory: Photographs from Everest to Africa*, and online here: http://www.alfredgregoryphotographs.com/Ev14.html. Exposed ice would lead to a completely different surface mass balance in COSIPY simulations and hence a completely different precipitation tuning factor. As a consequence, it seems impossible to know whether the P22 precipitation estimates are biased high or low.

Gregory, A. *Alfred Gregory: Photographs from Everest to Africa*. Penguin Random House, 2008. https://books.google.fr/books?id=R8J8GQAACAAJ.

**[CC1-8]** (5) We believe that the P22 core was drilled in a stagnating glacier that has a seasonally reconstituted accumulation zone comprised of continually avalanching snow and ice. In this context, we note that the arguments invoked by B22 about implied low ice velocity being evidence of no/limited ice melt (due to low mass turnover) are not relevant; they are just another way of describing their (alternative) hypothesis – that P22 drilled their ice core from the ablation area of glacier in balance. Under the P22 stagnation (and thinning) hypothesis, there is clearly no requirement for the ice flux to balance the implied ablation!

We also emphasize that our estimated ice loss was not just a guess, it was based on the identification of annual layers in the 10m ice core (verified by seasonality similar to our other Himalayan ice cores), radiocarbon dating of the near top and the bottom of this core, and depth/age data developed from the Rongbuk Glacier ice core which we (along with our Chinese colleagues) recovered ~5km north of South Col Glacier at 6518 masl in 2002 (Kaspari et al., 2009).

**[ACC1-8]** All considerations about ablation/accumulation zones and ice flow have been removed (see ARC1-4). Actually, the point was to say that given that there is no thinning (as evidenced by Brun et al. (2022) DEM differencing or picture comparison provided by Reviewer 4 - see RC4-2), ablation rates as high as 1.5 m w.e./yr simulated by Potocki et al. (2022) are only possible if they are compensated by an emergence velocity of the same magnitude. Otherwise, if not compensated, it would result in thinning. We argue that such a large emergence velocity is very unrealistic for this nearly stagnating glacier.

We do not question the dating of the ice core, which we are convinced was done in a rigorous way, but to our knowledge, the dating of the bottom of the core is available neither in P22, nor on CCI repository (https://www.icecoredata.org/cci/Others.html). We only question the interpretation of this dating, especially in terms of thinning. The only ways to reconcile our observation of no thinning (which was also done in a rigorous way following state-of-the-art methodologies) and the ice core dating are 1) that ice melt was compensated by emergence velocity, or 2) that the ice core is in an area of near balance or slight ablation over the past 2000 years. Explanation 1 is very unrealistic for this small glacier while explanation 2 is coherent with the seasonal presence of blue ice at the surface, the windy conditions at the col and the wind erosion model we proposed.

**[CC1-9]** (6) B22 assert that there is no evidence for any substantial ice melt having occurred due to an absence of fluvial features on the ice or off-glacier. The authors of P22 discussed this via email with the B22 authors, but some relevant parts of that discussion were omitted in B22: (1) the gently sloping SCG surface (below the avalanched region) would promote

evaporation of meltwater, not least because of the extreme vapour pressure gradient to be expected with a surface so much warmer than the atmosphere above. The SCG is also very small, so we should not expect "large supraglacial stream features" (visible in satellite imagery) to form. For example, P22 proposed a potential (ice) melt rate of ~16 mm/d (assuming 1.5 m w.e. lost during a 90-day monsoon). If this depth (0.016 m) is multiplied by the 200,000 m2 area of the SCG it means a volume of 3,200 m3 evacuated per day, so a mean runoff of 0.04 m3 s-1. Given that much of this would be lost to evaporation, and possibly split between multiple streams, the potential for large supraglacial meltwater features seems limited. We also do not understand which "photographs of the glacier surrounds" B22 refer to when citing no "evidence of runoff, such as stones being embedded into re-frozen water" Such features would not be visible in satellite imagery; and re-frozen meltwater would in any case quickly sublimate at the SCG. We also explained that mountaineers described to the P22 team (prior to their 2019 expedition) that they could expect to observe meltwater at the SCG during the pre-monsoon (i.e., not even the period of maximum temperatures and insolation).

[ACC1-9a] Thanks for the considerations concerning the fluvial surface features. These considerations have been removed in the revised MS. Nevertheless, P. Mayewski and others say that, in case of 16 mm w.e./day of melt, most of the water would be lost by evaporation. This is actually not physically plausible, because this process would require a tremendous amount of turbulent latent heat flux. Indeed, to evaporate 16 mm w.e./day i.e. 16/86 400 = 0.0002 kg/m2/sec, this would require a mean daily latent heat flux of 0.0002 * 2.5 10^6 = 460 W/m2, which is unrealistic especially because the wind is light in the monsoon. The latent heat flux is at least one order of magnitude lower than this value (as stated by Tom Matthews as well - see CC2-1b), so most of the melt water cannot be lost by evaporation.

[CC1-9b] We also highlight that P22 suggest melting of an ice surface – if exposed – would occur during the monsoon, when equivalent temperature (proportional to the sum of the atmospheric sensible and latent heat content) is at a maximum, insolation is closest to its peak values, and when light winds would limit the potential for cooling of the surface via the turbulent heat fluxes. Unfortunately, very few people have ever seen the SCG during the monsoon – and that extends to the imagery shared by B22! However, we can make a space-for-time substitution. The authors of P22 have spent a combined total of almost one month at Camp II (6,464 masl) during the pre- monsoon (late April to late May) in 2019 and 2022. On both occasions, meltwater was abundant, with a significant supraglacial stream present on the northern margin of the Khumbu Glacier throughout (unfortunately we did not take pictures, but we estimate it several metres in width and tens of centimetres in depth). Following from the above, this is consistent with the much larger catchment area of the upper Khumbu compared to the SCG. In early May 2022 the team also observed a saturated snowpack at the base of the Lhotse Face (>100 m above Camp II; see the foreground in the image below taken during the 2022 expedition). This melting is occurring on a surface with a higher albedo than would be expected if ice were exposed at the SCG.

[Figure]

During the team's last days at Camp II in early May 2022, meltwater was particularly widespread (beyond the normal confines of the aforementioned stream), with it even being necessary to excavate channels to prevent the tent from being flooded (see image below). Just six days earlier Camp II was covered by ~10 cm of snow, and the maximum air temperature throughout this period of high melt remained well below freezing at -5C.

[Figure]

Critically, the potential melt energy is very similar between Camp II during late-April to late May, and the SCG during the monsoon (see figure below). Indeed, the incident (short- and long-wave) radiation (which, P22 suggest, drives the melting at the SCG) is almost identical between sites (right-hand histogram). Note that the difference in equivalent temperatures (left-hand histogram) becomes ever-less important as wind speeds drop. This point relates very clearly to those raised in the energy-balance section above. That is, this space-for-time substitution suggests that, if abundant melt is evident *above* Camp II *during the pre-monsoon*, it would be reasonable to expect a similar response at the SCG during the *monsoon,* given that the former seems to be a very appropriate (perhaps even conservative) analogue for the latter (given the lower albedo of the ice at the SCG).

[Figure]

[Figure]

Above: Comparison of equivalent temperature (left) and incident radiation (incident shortwave radiation plus incident longwave radiation; right) at Camp II (red) during the pre-monsoon (last week in April to the last week in May, 2019 and 2022) and the SCG (blue) during the monsoon (July and August, 1991-2020). Note that the Camp II data were taken from the AWS at 6464 masl, and the SCG data were taken from the P22 ERA5 downscaled data (to the SCG AWS at 7,945 masl).

**[CC1-9b]** See RC1-15 and ACC2-1b for replies to the space-for-time substitution experiment. Actually, we welcome the observations made in CC1-9 about ice melt at Camp II, which provide clear validation of our dH data. Panels d-f of Resp. Fig. 1 (ARC1-2) show the changes around Camp II between 1984 and 2018, as captured by the aerial photographs, Pléiades scenes and subsequent DEM differencing. The majority of the ice loss here has occurred from the lowermost parts of the steep hanging glaciers above the Camp II, which have receded to expose a greater area of bedrock directly north of the moraine on which the camp is placed. The portion of Khumbu Glacier proximal to Camp II has experienced some slight surface lowering over the study period (~10 m, or 30 cm/year), with any higher magnitude changes in elevation again restricted to steeper areas of the Western Cwm affected by extensive crevassing.

**[CC1-10]** Taken together, point (6) indicates that B22 do not provide convincing evidence that substantial ice melt has not occurred at the SCG.

In closing, we highlight that P22 and now B22 have taken very different approaches to the study of the iconic SCG. They also reach different conclusions over whether (1) the SCG *could* thin rapidly (if ice were exposed), and (2) whether it *has* thinned rapidly.

We argue here that B22's findings – which challenged those of P22 on both counts -- are more uncertain than presented by the manuscript in its present form and should be re- examined before their paper is published.

[ACC1-10] Thanks for the interesting discussion P. Mayewski and others provided. We agree that Potocki et al. (2022) and Brun et al. (2022) have taken different approaches and reached contradictory conclusions regarding the recent thinning of SCG. We have tried to address all comments as well as possible. The main finding of Brun et al. (2022) is based on DEM differencing showing no thinning since 1984. And we hope that our tests concerning the sensitivity of Cosipy outputs to the different numerical implementations at SCG have raised some awareness about the quantification of melt in such a high elevation site.

References:

Bessin, Z.; Dedieu, J.-P.; Arnaud, Y.; Wagnon, P.; Brun, F.; Esteves, M.; Perry, B.; Matthews, T.
Processing of VENµS Images of High Mountains: A Case Study for Cryospheric and Hydro-Climatic Applications in the Everest Region (Nepal). Remote Sensing. 2022, 14, 1098. https://doi.org/10.3390/rs14051098

Brun, F., King, O., Réveillet, M., Amory, C., Planchot, A., Berthier, E., Dehecq, A., Bolch, T., Fourteau, K., Brondex, J., Dumont, M., Mayer, C., and Wagnon, P.: Brief communication: Everest South Col Glacier did not thin during the last three decades, The Cryosphere Discussions. [preprint], https://doi.org/10.5194/tc-2022-166, in review, 2022.

Grey, L., Johnson, A.V., Matthews, T., Perry, L., Elmore, A.C., Khadka, A., Shrestha, D., Tuladhar, S., Baidya, S.K., Aryal, D. and Gajurel, A.P. (2022), Mount Everest's photogenic 160. https://doi.org/10.1002/wea.4184

Kaspari, S., P. A. Mayewski, M. Handley, E. Osterberg, S. Kang, S. Sneed, S. Hou, and D. Qin, 2009, Recent increases in atmospheric concentrations of Bi, U, Cs, S and Ca from a 350-year Mount Everest ice core record, Journal of Geophysical Research, 114, D04302, doi:10.1029/2008JD011088.

Potocki, M., Mayewski, P.A., Matthews, T. et al. Mt. Everest's highest glacier is a sentinel for accelerating ice loss. npj Climate Atmospheric Sciences 5, 7 (2022). https://doi.org/10.1038/s41612-022-00230-0

Salerno, F., Guyennon, N., Thakuri, S., Viviano, G., Romano, E., Vuillermoz, E., Cristofanelli, P., Stocchi, P., Agrillo, G., Ma, Y., and Tartari, G.: Weak precipitation, warm winters and springs impact glaciers of south slopes of Mt. Everest (central Himalaya) in the last 2 decades (1994–2013), The Cryosphere, 9, 1229–1247, https://doi.org/10.5194/tc-9-1229-2015, 2015.

**Community comment 2 - Tom Matthews**

Reply to reviewer Ann Rowan

**[CC2-1]** I am very grateful for the substantial efforts of reviewer Ann Rowan in attempting to reconcile disagreements across three sources (Potocki et al., 2022; Brun et al., 2022 – under review here; and Mayewski et el. 2022 – a comment posted in this forum on Brun et al. 2022). This is clearly a challenging task. In this reply to Ann Rowan (AR), I wish to make a few clarifications to aid with the rest of the review process.

**[CC2-1a]**First, AR states: "L28: worth noting here that the South Col AWS recorded only about five months of data (May-end summer 2019)..."
However, the data (freely available here) extend from May 2019 to August 2022. The record (although interrupted in places) is therefore over three years in length.
**[ACC2-1a]** Thanks for this clarification and the whole observation period is now mentioned in the revised text (see ARC1-9)

**[CC2-1b]**Second, AR comments (in relation to the 'space for time substitution' in Mayewski et al. 2022) that it is 'not convincing' because: "the incident radiation would presumably be much lower than given here due to monsoon cloud cover" However, the incident radiation 'given here' (i.e., shown in the histogram of Mayewski et al. 2022) takes account of cloud cover. It uses the long-term (1991-2020) downscaled ERA5 insolation (which Potocki et al., 2022 showed was an excellent match to the observations). It therefore accounts for variable atmospheric transmissivity – including the effects of clouds. Note, too, that the extremely high insolation (close to top-of-atmosphere values) at the South Col AWS can be seen in Figure 4 of Matthews et al. (2020).
We make this clarification because it appears that AR was not aware of the provenance of the data used in the space-for-time comparison. A more subjective point I note here is that AR stated that a large temperature difference between Camp II and the South Col means that the melting is unlikely to occur at the latter, even if it does at the former. However, as pointed out in Mayewski et al. 2022, this temperature difference counts very little if winds are very light (which they are in the monsoon: Matthews et al. 2020, Fig. 4). Indeed, in the extreme event that the air is still, the turbulent heat flux is zero, so the lower air temperatures do not act to cool the surface (note that the right-hand side of the histogram shown in Mayewski et al. shows incident radiation, so already includes the effect of lower air temperatures on longwave radiation).
**[ACC2-1b]** Thanks for the clarification. This comment is for Ann Rowann. Nevertheless, we believe that this space-for-time comparison can only remain very speculative, as it is less solidly grounded than the modeling work of P22. We fully agree that radiative fluxes matter a lot when it comes to surface energy balance, however they are not the only components to take into account. As shown in our study, heat diffusion in the snow and ice plays an important role and cannot be neglected. In P22 and in COSIPY_grad simulations all the surface fluxes are similar, and still the predicted melt is completely different. We also refer to RC4-4, who shows a large variety of melt estimates for exactly the same meteorological forcings as P22. In other words, the space-for-time substitution would apply only if we would study an infinitely thin layer of ice. We also note that the temperature distributions are very different, with roughly 15 K of difference, which leaves much more room for night time cooling at SCG through sensible heat flux, despite constantly overcast conditions. We note a 1500 m elevation

difference between camp II and the South Col Glacier, which seems very large. For our alpine standards, it would mean comparing the conditions at Jungfaujoch with the tongue of Aletsch Glacier, or comparing the col du midi with the tongue of Mer de Glace.

**[CC2-1c]** Third, AR states that: "Determining mass change over a representative timescale of several decades requires observations of longer-term change as provided by both papers." This seems to be a misunderstanding of the methods used by Potocki et al. (2022): their conclusions were reached with the help of a 70-year record of (downscaled ERA-5) meteorological data, not just data from the 2019 season (which is the focus of Brun et al. 2022).

**[ACC2-1c]** Thanks for the clarification. We want to recall that COSIPY simulations performed in Brun et al. (2022) do not aim at quantifying the surface processes occurring at the glacier surface, but to warn about the extreme sensitivity of the model outputs to physical and numerical implementations. As a consequence, it does matter whether one year (2019 in Brun et al) or more are presented here. (see ARC1-3, and ARC4-3).

**[CC2-1d]** Fourth, AR mentions her own group's modelling work that shows annual accumulation of 7 m w.e. at the South Col. According to my understanding, this result represents precipitation minus sublimation and any melt, and hence appears physically implausible: 7 m w.e. is ~13 times the annual mean precipitation (AMP) measured at Base Camp (5,315 m asl) and ~9 times the AMP at Phortse (3,810 m asl). Considering that AR's figure is net of any sublimation (if not also melt), and both that theory and our measurements highlight a decline in precipitation with altitude, I suggest that this (non-peer-reviewed) result is physically implausible and should be discounted from further discussion.

**[ACC2-1d]** This discussion regarding precipitation amounts is very interesting and of particular importance for surface mass balance modelling. But we must keep in mind that the important variable is not really how much precipitation (snowfall in the case of South Col Glacier) falls at the glacier surface, but how much stays on the ground and for how long. Disentangling the true amount of precipitation from the net accumulated snow (i.e. the point surface mass balance) on the surface is impossible at this extremely windy site, and is beyond our capacities of observations or modelling. Some authors like Salerno et al. (2015) propose a decline in precipitation with altitude above 5000 m asl., but this altitudinal gradient is probably highly uncertain and anyway disturbed by topography. Moreover, the net accumulated snow on the surface is also very dependent on the interplay between wind and topography, with some locations prone to over-accumulation and others to strong erosion. In conclusion, observing one order or more of net accumulation magnitude from one site to another, only a few hundreds of meters away of each other, is common in high altitude windy sites (e.g., Vincent et al., 2017)

Salerno, F., Guyennon, N., Thakuri, S., Viviano, G., Romano, E., Vuillermoz, E., Cristofanelli, P., Stocchi, P., Agrillo, G., Ma, Y., and Tartari, G.: Weak precipitation, warm winters and springs impact glaciers of south slopes of Mt. Everest (central Himalaya) in the last 2 decades (1994–2013), The Cryosphere, 9, 1229–1247, https://doi.org/10.5194/tc-9-1229-2015, 2015.

Vincent, C., E. Le Meur, D. Six, M. Funk, M. Hoelzle, and S. Preunkert (2007), Very high-elevation Mont Blanc glaciated areas not affected by the 20th century climate change,J. Geophys. Res.,112, D09120, doi:10.1029/2006JD007407

**Community comment 3 - Tom Matthews**

**[CC3-1]** As a co-author on Potocki et al. (2022) and on the comment by Mayewski et al. (this Discussion), I would like to re-iterate the query expressed in the latter for Brun et al. to communicate/explore uncertainty in the 1984 DEM (and hence the DEM of difference) in more detail. For example, does the uncertainty analysis presented by the authors fully consider positional errors in the 1984 image (see comments about the camera's departure from nadir)? Is the uncertainty (in dH) independent of surface slope? Might it be, for example, that the very large dH in the (steep) upper Khumbu and Rongbuk glacier sections reflects such positional errors, rather than a redistribution of mass within the glacier. This request for more attention on uncertainty quantification was perhaps covered by reviewer Ann Rowan who advocated for greater clarity of the methods used, but has not been mentioned again by the other reviews. Given the importance of the DEM analysis -- to reach an important conclusion about an such an "iconic" location -- I hope that Brun et al. will attend to this request in their revision.

**[ACC3-1]** We refer to ACC1-4b for more information about the processing of the 1984 DEM and the hypothesized 'positional errors' mentioned in CC1-4b and CC3-1 here. Please refer to ARC1-2 regarding uncertainty in the DEM difference analysis.

**Community comment 4 - Nicholas Steiner**

**[CC4-1]** As a contribution to this discussion, I would like to submit some observations from remote sensing that may aid in interpretation of annual surface melting from modeling in Brun et al. (2002). In Scher et al. (2021) we created a record of surface melting over glaciers in the Himalayas from a time series Sentinel-1 synthetic aperture radar (S1-SAR). Melt is detected where annual backscatter is reduced as liquid surface water obscures the radar scattering from the glacier interior, resulting in a marked reduction in backscatter. For the South Col glacier, we observe radar signatures that indicate surface melting is occurring in 2019 over areas of exposed ice in the southern extent of the glacier (Figure 1, attached). From time series S1-SAR, we observe continuous indications of surface melting from June 26, 2019, until October 6, 2019, with approximately biweekly repeat observations during this period. Since seasonal snow over areas that are exposed on an interannual basis are not deep enough to contribute substantially to radar scattering, we infer that the melting signal originates from structural features (e.g., laying) in the glacier interior that result in enhanced backscatter during colder winter months. It is important to note that at C band frequencies backscatter is extremely sensitive to liquid water and it is difficult to differentiate very small amounts of surface melting from more extensive melting, and therefore our methodologies are not well suited to evaluate the amount of melting that may be occurring. For more details on our methodologies, please refer to Scher et al., (2021).

[Figure]

Figure 1 (a) A true-color map from Sentinel-2 during April 2021 of the South Col Glacier indicates areas of blue ice. Sentinel-1 time series synthetic aperture radar (SAR) from locations in (a) the accumulation zone, as designated by Brun et al., (2022) in Figure A6 indicate melting where observed backscatter is found to be ~3 db below (red lines) the winter mean (black-lines). For both the (c) exposed ice and the location of the (d) ice core site from Potocki et al. (2022) we find similar radar signatures that indicate surface melting.

Scher, C., Steiner, N. C., & McDonald, K. C. (2021). Mapping seasonal glacier melt across the Hindu Kush Himalaya with time series synthetic aperture radar (SAR). *The Cryosphere*, *15*(9), 4465-4482.

 **[ACC4-1]** We would like to thank Nicholas Steiner who contributed to the general discussion. However, we want to raise some awareness about the direct interpretation of the results from Sentinel-1 (S1) backscatter data. We also note that the two dates provided by N. Steiner (2019-06-26 and 2019-10-06) correspond to acquisitions from orbit 121 descending where South Col is completely in radar layover. Therefore, we performed additional analysis of S1 data, which we show below.

For the region of interest S1 radar data (IW, GRD) from two orbits are available (012 ascending and 121 descending). Due to the very steep topography, South Col is located completely in radar layover for orbit 121. Therefore, we analyzed data from orbit 012 where South Col is partially visible. For orbit 012, the approximate incidence angle relative to the ellipsoid is theta = 34°, the acquisition time: 12:13h UTC = 17:58 NPT local time. The data was preprocessed and downloaded from Google Earth Engine as orthorectified 8 bit grayscale data with 10m pixel spacing, VV polarization, and with sigma_0 ranging from -22 to +5 dB. Google orthorecitifies the data using the SRTM 30m DEM (https://developers.google.com/earth-engine/guides/sentinel1#sentinel-1-preprocessing).

Due to the steep topography, the SRTM can contain significant artifacts likely due to layover or phase unwrapping errors of the SRTM raw data. A comparison with the ALOS World 3D - 30 m DEM (AW3D30) shows that at South Col the SRTM is 30 to 90 meters higher than the AW3D30 while at other locations the difference is around zero (-+10m). The height errors of the SRTM cause horizontal shifts in the orthorectification in the direction of ground range (line-of-sight projected to the ground). At 34° incidence angle, a height error of Delta_h = +90 m corresponds to an horizontal shift of Delta_x = 90m/tan(theta) = 133 m to the west, therefore it is difficult to precisely geolocate S1 backscatter data as suggested by the figure provided by N. SteinerAs reprocessing the entire S1 data is beyond the scope, we rely on visual inspection and detection of features to accurately determine points for further analysis of the radar backscatter signal. Resp fig. 6 shows where in the radar image we identified the ice covered part of South Col (blue) and the ice free part (red).

[Figure]

**Resp. Fig. 7:** Location of points of interest on the swisstopo map (top panel) and on the S1 mean backscatter image (bottom)

From Venus and S2 data, we found that South Col Glacier is largely snow covered from June/July - Dec in 2016 - 2022. For an analysis of the backscatter time series we selected several  points and averaged the backscatter intensity within a window of 90x90 meters: We selected two points in the ice-covered part of South Col (North and South of P.  8029), one point in the ice free / rock covered part of South Col, and one point on the West Cwm Glacier at an altitude of 6600m where the glacier is relatively flat and not strongly affected by ice avalanches. For all of these four points we plotted backscatter time series (Resp. Fig. 7).

[Figure]

**Resp. Fig. 8:** Time series of backscatter intensity for the locations identified in Resp. Fig. 7

We found that all points show seasonal variations in backscatter, very likely due to the presence of liquid water (we could exclude other reasons for backscatter reduction like refractive-index matching, smoothing of the surface, increase moisture in the atmosphere which all have weaker effects than 2-3 dB). There is a strong temporal correlation of the backscatter drop at West Cwm Glacier and at South Col, which we interpret as a strong indicator for the existence of wet snow/liquid water at South Col. However, the backscatter drops 12 - 15 dB on West Cwm Glacier where the snow is not only wet but also producing surface runoff. At an elevation of about 7000 m, S1 backscatter imagery indicates a transition from a strong reduction of backscatter in summer to a weaker, but still significant reduction, indicating a different kind/depth of snow or different liquid water content. Similarly, at South Col (~8040m), the backscatter drops only by 2 - 5 dB which is still significant compared to other effects affecting the backscatter. A backscatter drop is also observed for the ice-free/rock covered area at South Col, likely because the rock is covered by snow in summer.

From the S1 data, we conclude that wet snow/liquid water occurs at South Col, but we cannot quantify the amount of runoff.

**Community comment 5 - Tom Matthews**

[CC5-1] This is a very valuable review. I commend the authors for attending to the debate in so much detail.

However, I wish to make a brief correction. It is stated that Potocki et al. (2022) did not assess the sensitivity of their SEB results to albedo. However, they did indeed perform this sensitivity assessment as part of a 'bounded' uncertainty quantification (Smith et al., 2018), whereby parameter values were perturbed to their plausible min/max values in order to estimate equivalent, plausible min/max rates of ablation. As part of that assessment (detailed in the SI of Potocki et al., 2022), the albedo was varied between 0.3 and 0.5. The higher albedo reduced the melt rate, but did not materially affect their SEB conclusions (i.e., the physical plausibility of substantial ice melt).

I also note that whilst the ice at the drill site was cleaner that the ice circled in the image below (shown in Matthews et al., 2022), the blue ice areas of Antarctica are clearly not a good surrogate for albedo at the South Col Glacier as a whole, so would encourage any such analogy to be made with caution. The ice circled is indeed likely to have been darkened through ablation (and may well have an albedo appreciably below 0.4). The assignment of the 0.4 value by Matthews at al. (2020) and Potocki et al. (2022) was an estimate, but an informed one based on visual site inspections/comparisons by the authors.

[Figure]

I also suggest cautious interpretation of the reviewers' 'spin-up' experiments. Presumably the reduced melt is because a perennial snowpack develops? This is questionable, because the precipitation data used by the authors of this review are tied to the surface energy balance results of Potocki et al (precipitation was corrected by the authors to match ablation in the first decade of the simulation). With another model formulation that simulates less ablation, the

precipitation would be reduced too (so a perennial snowpack would likely not develop in the spin-up period).

[ACC5-1] Thanks for the clarifications, no reply needed from our side here. And any modeling approach is presently very limited by the absence of in-situ data, such as albedo values at least. The point raised by the reviewers about SCG albedo being potentially higher than 0.4, and potentially reaching 0.6 is probably relevant for most parts of SCG not visible on the picture shown by Tom Matthews. When looking at pictures of SCG on the internet, we were actually surprised by how difficult it is to tell if the surface is snow or ice (see for example: https://www.mountainpanoramas.com/___p/___p.html?panoid=2022_M1&labels=on), suggesting that exposed ice has actually a very high albedo.

[CC5-2] Importantly, this review, and the work of Brun et al., build on Potocki et al. to highlight the sensitivity of the South Col Glacier to SEB modelling uncertainty. This is very interesting in its own right. If Potocki et al.'s model formulation is close to being 'right', it suggests an extreme sensitivity to the maintenance of a protective snowpack at the South Col. This can remain true, even if Brun et al. are right in that the South Col Glacier did not actually thin as Potocki et al. suggested (i.e., because the snowpack has mostly been preserved).
If the conservative model configurations results of this review and Brun et al. are correct, however, then rapid thinning is unlikely even if ice becomes exposed for long periods of the year.
The resolution of this debate matters for our understanding of what may happen in the future at the South Col Glacier, and possibly for other ice masses at extreme elevation. Accordingly, I suggest that the discrepancy between results should be a call for urgent further research.
There are bright possibilities for that in the near future. First, the albedo at the South Col Glacier can be constrained using satellite measurements (Bessin et al., 2022). Second, high-resolution radar measurements can be used to identify melt events (Scher et al., 2021). Given that melt events are regularly detected at the South Col (Steiner, this discussion), we must reject the parameter values and model structures which generate no melt. Indeed, given the comment by Steiner in this forum showing melt events throughout June-October, I would also suggest that setups generating minimal melt rates (Brun et al., and Machguth and Mattea) – even when ice is exposed – are unlikely to be appropriate. Although Steiner's results cannot quantify the magnitude of melting, they show that it is not a rare occurrence. Indeed, if some melting is detected in June and October, melt rates in the peak Monsoon months of July and August – when moist enthalpy is at a maximum and wind speeds are at a minimum (Khadka et al. 2021) – would be considerably higher for an exposed ice surface. Quantifying how much higher – with perturbed parameter ensemble model runs constrained by satellite observations and the weather station data -- should be a high priority for future research.

**[ACC5-2]** We agree with Tom Matthews' comments for future research about this SCG. We want to add two important concerns that Tom Matthews did not mention: 1. wind related processes (erosion, sublimation, drift) are likely the most important processes to look at, if a dedicated study is undertaken on SCG, 2. regardless of the amount of melting at the surface, a thorough quantification of how much meltwater refreezes at the surface is necessary. We note that ice and snow albedo quantification is not possible from Venus images, as it does not have the 1.6 µm SWIR band needed to compute albedo (Bessin et al., 2022). However, Sentinel-2 images could be suitable (Naegeli et al., 2017).

New references cited

Bessin, Z., Dedieu, J.P., Arnaud, Y., Wagnon, P., Brun, F., Esteves, M., Perry, B. and Matthews, T., 2022. Processing of VENµS Images of High Mountains: A Case Study for Cryospheric and Hydro-Climatic Applications in the Everest Region (Nepal). Remote Sensing, 14(5), p.1098.

Khadka, A., Matthews, T., Perry, L.B., Koch, I., Wagnon, P., Shrestha, D., Sherpa, T.C., Aryal, D.,Tait, A., Sherpa, T.G. and Tuladhar, S., 2021. Weather on Mount Everest during the 2019 summer monsoon. Weather, 76(6), pp.205-207.

Scher, C., Steiner, N.C. and McDonald, K.C., 2021. Mapping seasonal glacier melt across the Hindu-Kush Himalaya with time series synthetic aperture radar (SAR). *The Cryosphere*, *15*(9), pp.4465-4482.

---

## Referee Report (RR1)

**Review of revised manuscript "Brief communication: Everest South Col Glacier did not thin during the last three decades" by Brun et al.**

**Horst Machguth, Enrico Mattea and Marcus Gastaldello,** Department of Geoscience, University of Fribourg, Switzerland

**1. Introduction**

We thank Brun et al. for the revised version of the manuscript. We believe the revised manuscript reads well and concise. Basically, we only have a few minor comments. However, Marcus Gastaldello, an MSc student currently simulating firn processes at Colle Gnifetti (Switzerland/Italy) with COSIPY, came across a number of issues in the code of COSIPY. These issues have recently been communicated to the COSIPY developers. The model version used in the revised manuscript (and in the paper by Potocki et al.), however, might be affected by these issues. We highlight these issues below for Brun et al. being able to assess their relevance to the simulations.

**2. Known issues with COSIPY**

The model employs an L-BFGS-B or SLSQP algorithm to iteratively solve the surface temperature for each time-step, by minimising flux residuals in the surface energy balance. This solver is constrained with an upper bound of the melting point of ice and uses the result of the previous time-step as an initial guess. Unfortunately, there appears to be a susceptibility for the algorithms to prematurely terminate on the upper bound of 273.16 K, before the actual convergence of the energy fluxes - particularly if the previous surface temperature value was at this temperature. Whilst this does not produce additional melt, since there is no positive excess energy, the reported surface mass and energy fluxes are incorrect. In addition, there is a missing pair of parentheses in the calculation of the ground heat flux: this issue was amended by the model developers in early August 2022, but was likely still present in the calculations of the paper by Potocki et al.

Thermal diffusion through the sub-surface layers is determined by resolving the Fourier heat equation with a second order, central difference scheme. However, the scheme uses a fixed/Dirichlet boundary condition on the basal node that effectively constrains the thermal regime to the user-defined, initial basal temperature, set in the constants file.

A volumetric approach is used to determine the composition of sub-surface layers, representing them in terms of a fractional proportion of ice, water and air. Within the refreezing module, energy is not properly conserved during latent heat release. The calculation of the internal energy increase of firn layers only accounts for the fractional mass of the converted water, as opposed to the whole layer. This results in a substantial under-estimation of the layers subsequent temperature increase. Furthermore, water is distributed to layers via a bucket approach constrained by their irreducible water content, prior to refreezing. This significantly restricts the true refreezing potential of sub-surface layers as it should be constrained by the cold content and volumetric limits of the layer.

We emphasize that the most critical of these issues, the erroneous calculation of refreezing in firn and snow, might be irrelevant to the ice-only simulations by Brun et al. and Potocki et al.

**1. Detailed comments on the revised manuscript by *Brun et al.**

Line 47: Correct us if we are wrong, but does the resolution depend on the location on the image (higher resolution for areas closer to the camera, coarser for areas further away)? This would be relevant in the extreme topography of the Everest range. Hence, if we do not misunderstand, 0.5 m is an average value? We suggest mentioning that 0.5 m is an average value.

Lines 79-85: South Col Glacier is the main focus of the paper, so maybe its dH result should be described before the dH result of the rest of the Everest region? (Although that provides a nice contextualization). Up to the authors.

Line 94: Suggest "in the thickness" instead of "of the thickness"

Line 109: Remove the number and leave "Fourteen".

Lines 119/120: Reword: " except at the on the lower cliff "

Lines 156/158: suggest removing "at" after "averaging"

Line 220: "abusively" - while we agree with the concept, is this proper wording?

Lines 228-229: Could the Authors provide some more details on how the model initialization works when precipitation is switched off? Does the grid after spin-up consist entirely of impermeable glacier ice, during the whole simulation of 2019?

Line 280: "local mass balance rates approaching 2 m a-1," Shouldn't it read -2 m a-1?

Line 282: Suggest writing "~2000".

Lines 288: As above.

Line 293: Suggest adding the uncertainty.

Line 298: suggest "mean density" instead of "density"

Line 307: typo in "bergschrung", should read "bergschrund"

Line 308: suggest "horizontal velocity" instead of "velocity" (possibly remove "horizontal" at the next line if it sounds too repetitive)

Line 333: suggest "or" instead of "nor".

Figure 3: Suggest including "accumulation efficiency" in the legends of subplot b and c.

Figure A4 still mentions a now-removed panel about Crocus.

---

## Author Response (AR2)

Dear Editor and dear reviewers,

We thank all the reviewers and the Editor for this second round of reviews. Below we provide a point-by-point response to all the comments. The original comments are in the blue colored cells of the table and our responses are in the white cells. The changes in the text are highlighted in orange font. We did some additional minor editorial changes to improve the clarity of the text, and added a figure (detailed in response to **R3C2** below).

Former lines 187 to 197 now reads as:

"From the analyses of the satellite images and the modeling of wind erosion, we conclude that large parts of the fallen snow are likely eroded or re-mobilized after deposition, adding a degree of complexity in the precipitation estimates. Potocki et al. (2022) used a stationary scaling factor to compute effective precipitation, whereas our analysis suggests that this should instead exhibit seasonality. The higher effective precipitation in the monsoon (from reduced wind speeds) that we find here would make it easier to re-establish a snowpack over the glacier -- something that Potocki et al. (2022) inferred was unlikely to occur in their 'ice' experiment. Hence, the South Col Glacier may not have thinned as Potocki et al. (2022) concluded because the high ice melt rates required do not have a chance to occur as the glacier surface remains snow covered throughout the monsoon."

We hope that we addressed all comments in a satisfying way and that the changes we suggest are acceptable at this stage of the revision process. We can make larger changes in the structure, and move some figures from the appendix back to the main text if needed.

All the best,
Fanny and co-authors

Handling Editor (Thomas Mölg)

**[HEC1]**Dear authors,

After taking over this manuscript and studying the MS records, I am convinced that your article will be a valuable addition to the literature. Its evolvement represents an openness in the very best scientific sense, which is one of the essential quality flags of scientific research.

I would like to thank all involved parties: the authors of this manuscript as well as the authors of the Potocki et al. (2022) study; the reviewers; the colleagues who contributed comments; and the former editor from who I took over.

Thanks for your positive appreciation of our work. We now also thank all the participants to the lively discussion and added the following sentence in the acknowledgement section:
"We thank the editors (Francesca Pellicciotti and Thomas Mölg), the five reviewers (including Ann Rowan, Horst Machguth and Martin Lüthi), Enrico Mattea, Marcus Gastaldello, Nicolas Steiner and the authors of Potocki et al. for insightful and constructive comments and vivid discussions, which improved the quality of this manuscript."

Please consider some closing, minor remarks from the reviewers for preparing the final version of this manuscript (in particular, I would ask to think about whether the reference to "Antarctic blue ice" is necessary; one could simply call the albedo of the study glacier "blue ice-type albedo" and mention that a somewhat higher value is likely).

We removed the reference to Antarctic blue ice, and the sentence now reads as:
"Indeed, the ice of South Col Glacier appears very blue and might have a larger albedo than 0.4, as it is observed for blue ice-type albedo that can reach 0.5 to 0.6 (Smedley et al., 2020)."

Also, and as discussed with you, we would welcome a formal change of this manuscript, which means it should not be a "brief communication" (it is too lengthy in the meantime). I invite you to think about the following and change the manuscript type:
- Take brief comm. out of the title. If you want to indicate that this is kind of a comment on a different paper, you could start your title with "An alternative perspective on ..." and make it clear in the abstract what study you are referring to.

We agree that our manuscript does not compile with the "Brief communication" format. We changed the title, which is now: "Everest South Col Glacier did not thin during the period 1984-2017"

- You could make the abstract a bit longer, if you want.

We expanded the abstract to include an explicit reference to Potocki et al. (2022). It now reads as:
"The South Col Glacier is a small body of ice and snow (approx. 0.2 km$^2$), located at the very high elevation of 8 000 m a.s.l. on the southern ridge of Mt. Everest. A recent study by Potocki et al. (2022) proposed that South Col Glacier is rapidly losing mass. This is in contradiction with our comparison of two digital elevation models derived from aerial photographs taken in December 1984 and a stereo Pléiades satellite acquisition from March 2017, from which we estimate a mean elevation change of 0.01 ± 0.05 m a$^{-1}$. To reconcile these results, we investigate some aspects of the surface energy and mass balance of South Col Glacier. From satellite images and a simple model of snow compaction and erosion, we show that wind erosion has a major impact on the surface mass balance, due to the strong seasonality in precipitation and wind, and cannot be neglected. Additionally, we show that the melt amount predicted by a surface energy and mass balance model is very sensitive to the model structure and implementation. Contrary to previous findings, melt is likely not a dominant ablation process on this glacier which remains mostly snow-covered during the monsoon."

- In my opinion, you do not have to change the structure. You could add one sentence in the introduction, saying that this research evolved from a comment-type manuscript and, therefore, does not follow the common structure.

Kind regards,
Thomas Mölg

Handling Editor &
Co-Editor-In-Chief TC

Thanks for this suggestion, we changed the beginning of the last section of the introduction which is now:

"In this article, we explore [..]. The structure of this article is rather uncommon, as it evolved from a comment-type manuscript instead of a stand-alone research."

**Reviewer 1**

**[R1C1]** This submission led to an unusually lively discussion, which might even become a case study of how interactive journals can operate. The authors have responded robustly to review and community comments, and have revised their manuscript accordingly. I recommend that a final paper can now be published after copy editing for minor writing errors (a notable one is that "abusively" on line 220 should be "incorrectly").

We thank the reviewer for their positive comment and modified L220 accordingly

**Reviewer 2 (Martin Lüthi)**

**[R2C1]** Dear collegues

The scientific editor asked me for an independent assessment of this publication. I have not followed the debate and have not read all the arguments in the long discussion on TCD.

Overall, I find this manuscript very convincing. For the purpose of the determination whether this glacier has undergone important geometry changes, Chapter 2 would be sufficient. Careful analysis of different DEMs that are referenced on surrounding bedrock show little to no change. This is a simple story, entirely conclusive and convicing.

As an outsider (but with quite some experience working on and modeling high-altitude glaciers), I do not see any reason to debate this conclusion. If the methodology has been correctly applied (which I cannot judge without doing it myself), I think the results are solid and convincing.

The arguments given in the Potocki paper are, on the other hand, less convicing and likely incorrect. It seems that their dating of the ice core relies on a method with very high error bars, while they could not even detect the expected tritium peak in 1966 or volcanic SO4 layers that are commonly used to establish an ice core chronology.

Mass balance modeling in such an extreme geometry cannot work. Extreme wind drift changes accumulation/ablation rates by huge amounts within 10s of meters, and surface geometry is mostly shaped by winds and sublimation, and not by the amount of theoretically deposited

snow. The strongest proof is Fig 1 in Potocki: bare rock and thick ice next to each other prove that any mass balance model is bound to fail. In my eyes, this glacier should be considered a huge, persistent cornice, almost entirely shaped by the action of extreme winds.

Nevertheless, the author do an admirable effort to discuss and even to model many processes that are difficult in extended flat areas (such as an ice sheet), and nearly impossible in conditions like on a mountain saddle located in the free atmosphere and mostly above the weather systems.

Given my hand-wavy assessment above, and conviced that any thickness change of 10s of meters would be easily visible on photographs (which apparently is not the case), I am conviced that this manuscript is a solid piece of work. Therefore I recommend that it be published after correcting a few typos mentioned below.

Sincerely,

Martin Lüthi

We thank the reviewer (Martin Lüthi) for his positive assessment of our manuscript.

109 fourteen 14

307 bergschrund (not ng)

Modified accordingly

Fig 4: Temperature in degrees C might make this more readable

Changed accordingly

Reviewer 3 (Horst Machguth, Enrico Mattea and Marcus Gastaldello)

[R3C1] 1. Introduction
We thank Brun et al. for the revised version of the manuscript. We believe the revised manuscript reads well and concise. Basically, we only have a few minor comments. However, Marcus Gastaldello, an MSc student currently simulating firn processes at Colle Gnifetti (Switzerland/Italy) with COSIPY, came across a number of issues in the code of COSIPY. These issues have recently been communicated to the COSIPY developers. The model version used in the revised manuscript (and in the paper by Potocki et al.), however, might be affected by these issues. We highlight these issues below for Brun et al. being able to assess their relevance to the simulations.

We thank Horst Machguth, Enrico Mattea and Marcus Gastaldello for their positive comments and for their interest in COSIPY. We provide a point-by-point response to these comments

below.

**[R3C2]** 2. Known issues with COSIPY
The model employs an L-BFGS-B or SLSQP algorithm to iteratively solve the surface temperature for each time-step, by minimising flux residuals in the surface energy balance. This solver is constrained with an upper bound of the melting point of ice and uses the result of the previous time-step as an initial guess. Unfortunately, there appears to be a susceptibility for the algorithms to prematurely terminate on the upper bound of 273.16 K, before the actual convergence of the energy fluxes - particularly if the previous surface temperature value was at this temperature. Whilst this does not produce additional melt, since there is no positive excess energy, the reported surface mass and energy fluxes are incorrect. In addition, there is a missing pair of parentheses in the calculation of the ground heat flux: this issue was amended by the model developers in early August 2022, but was likely still present in the calculations of the paper by Potocki et al.

Thanks for reporting these issues. We are actually aware of both issues that we also reported to the developers of COSIPY. Our version of the model includes both bug fixes. We did a number of tests and found that they have no impact on the predicted melt rates in the case of our simulations.

We suspect additional changes in the code, as we did not manage to perfectly match the results from Potocki et al. (2022), as shown in Figure R1. We propose to add this figure to the appendix of our manuscript, as it shows the sensitivity of the predicted melt to the choice of the interpolation depth at which the subsurface/ground heat flux is calculated. It also shows some problems in COSIPY when this depth reaches the typical size of a near surface layer in COSIPY.

[Figure]

*Fig. R1: Sensitivity of the annual melt for the year 2019 to the choice of the temperature interpolation depth (zlt2) in COSIPY. Note that the parameter zlt1 is set as zlt2/2. The red symbols correspond to simulations run at a minute time step and the grey ones correspond to simulations run at an hour time step. P22 is the original simulation from Potocki et al. (2022).*

**[R3C3]** Thermal diffusion through the sub-surface layers is determined by resolving the Fourier heat equation with a second order, central difference scheme. However, the scheme uses a fixed/Dirichlet boundary condition on the basal node that effectively constrains the thermal regime to the user-defined, initial basal temperature, set in the constants file.

We are aware of this choice made by the developers of COSIPY. The boundary condition has no influence on the amount of surface melt predicted by COSIPY for the short runs (one year) we tested. However, we suspect that for longer runs the remeshing routine might introduce some unexpected behavior that likely breaks the energy conservation. In the figure R2, we show the mean temperature of the domain and the temperature of the bottom node of COSIPY simulations similar to the 'ice case' of Potocki et al. (2022). We do not fully understand how these issues are handled in the code, but they might create extreme diffusive fluxes at the boundary (especially at the ice/rock interface).

[Figure]

*Fig. R2: Annual air temperature and temperature of the ice column and ice bottom sampled every 5 days at 6:00.*

**[R3C4]** A volumetric approach is used to determine the composition of sub-surface layers, representing them in terms of a fractional proportion of ice, water and air. Within the refreezing module, energy is not properly conserved during latent heat release. The calculation of the internal energy increase of firn layers only accounts for the fractional mass of the converted water, as opposed to the whole layer. This results in a substantial under-estimation of the layers subsequent temperature increase. Furthermore, water is distributed to layers via a bucket approach constrained by their irreducible water content, prior to refreezing. This significantly restricts the true refreezing potential of sub-surface layers as it should be constrained by the cold content and volumetric limits of the layer.
We emphasize that the most critical of these issues, the erroneous calculation of refreezing in firn and snow, might be irrelevant to the ice-only simulations by Brun et al. and Potocki et al.

Thanks for highlighting these issues, we were not aware of them. We do not think that they impact our ice only simulations, but we will keep them in mind for applicability to other contexts.

**[R3C5]** 1. Detailed comments on the revised manuscript by Brun et al.

See the point-by-point responses below

Line 47: Correct us if we are wrong, but does the resolution depend on the location on the image (higher resolution for areas closer to the camera, coarser for areas further away)? This would be relevant in the extreme topography of the Everest range. Hence, if we do not misunderstand, 0.5 m is an average value? We suggest mentioning that 0.5 m is an average value.

You are correct, the resolution depends on the distance between the camera and the target. The 20 Dec. 1984 Washburn's Learjet flew at about 12 000 m a.s.l. The focal length of the Wild RC-10 camera is six inches (approximately 152 mm). The diapositives were scanned with a 1693 dpi scanner (in other words one pixel is $1.3 \cdot 10^{-5}$ m). This means that for a point located 3 000 m below the plane (at Everest summit), the pixel size is approximately 25 cm, and it is 35 cm for a pixel located 4 000 m below the plane (i.e. at the elevation of South Col Glacier) and would be 50 cm for a pixel located at 6 000 m.
We added "mean" to specify this point.

Lines 79-85: South Col Glacier is the main focus of the paper, so maybe its dH result should be described before the dH result of the rest of the Everest region? (Although that provides a nice contextualization). Up to the authors.

We prefer to keep the order as is to highlight that other glaciers in the Everest region are thinning, even at elevations higher than 6500 m.

Line 94: Suggest "in the thickness" instead of "of the thickness"

Modified accordingly

Line 109: Remove the number and leave "Fourteen".

Done, thanks for catching the typo.

Lines 119/120: Reword: " except at the on the lower cliff "

Done, thanks for catching the typo.

Lines 156/158: suggest removing "at" after "averaging"

Done

Line 220: "abusively" - while we agree with the concept, is this proper wording?

As suggested by reviewer 1, we replaced "abusively" with "incorrectly"

Lines 228-229: Could the Authors provide some more details on how the model initialization works when precipitation is switched off? Does the grid after spin-up consist entirely of impermeable glacier ice, during the whole simulation of 2019?

We start from an 'ice case' simulation, where the exposed surface consists of ice most of the time. In the restart file we used, the density is at 917 kg/m3 for all layers, confirming the presence of ice over the whole profile. We added this precision in the text:
"At the start of the simulation the snow thickness is zero, and the domain consists only of impermeable ice."

Line 280: "local mass balance rates approaching 2 m a-1," Shouldn't it read -2 m a-1?

Thanks for pointing this out. It should read -2 m a-1.

Line 282: Suggest writing "~2000".
Lines 288: As above.

Done

Line 293: Suggest adding the uncertainty.

Done

Line 298: suggest "mean density" instead of "density"

Done

Line 307: typo in "bergschrung", should read "bergschrund"

Done

Line 308: suggest "horizontal velocity" instead of "velocity" (possibly remove "horizontal" at the next line if it sounds too repetitive)

Modified accordingly

Line 333: suggest "or" instead of "nor".

Done

Figure 3: Suggest including "accumulation efficiency" in the legends of subplot b and c.

Done

Figure A4 still mentions a now-removed panel about Crocus.

Thanks for pointing this out, it is now removed.

Reviewer 4 (Ann Rowan)

We thank the reviewer 4 (Ann Rowan) for suggesting to accept our manuscript as is.

---

## Author Response (AR3)

Dear Thomas,

Thank you again for editing our article and for providing some final comments and suggestions.

We have modified the text as suggested, and decided to keep most of the figures in the appendix. Additionally, we added the University of Gratz affiliation for two co-authors (Owen king and Tobias Bolch).

All the best,

Fanny